# Neuropathic pain caused by miswiring and abnormal end organ targeting

Vijayan Gangadharan[1,2], Hongwei Zheng[3], Francisco J. Taberner[1,4], Jonathan Landry[5], Timo A. Nees[1], Jelena Pistolic[5], Nitin Agarwal[1], Deepitha Männich[1], Vladimir Benes[5], Moritz Helmstaedter[2], Björn Ommer[6], Stefan G. Lechner[1], Thomas Kuner[3] & Rohini Kuner[1✉]

Nerve injury leads to chronic pain and exaggerated sensitivity to gentle touch (allodynia) as well as a loss of sensation in the areas in which injured and non-injured nerves come together[1–3]. The mechanisms that disambiguate these mixed and paradoxical symptoms are unknown. Here we longitudinally and non-invasively imaged genetically labelled populations of fibres that sense noxious stimuli (nociceptors) and gentle touch (low-threshold afferents) peripherally in the skin for longer than 10 months after nerve injury, while simultaneously tracking pain-related behaviour in the same mice. Fully denervated areas of skin initially lost sensation, gradually recovered normal sensitivity and developed marked allodynia and aversion to gentle touch several months after injury. This reinnervation-induced neuropathic pain involved nociceptors that sprouted into denervated territories precisely reproducing the initial pattern of innervation, were guided by blood vessels and showed irregular terminal connectivity in the skin and lowered activation thresholds mimicking low-threshold afferents. By contrast, low-threshold afferents—which normally mediate touch sensation as well as allodynia in intact nerve territories after injury[4–7]—did not reinnervate, leading to an aberrant innervation of tactile end organs such as Meissner corpuscles with nociceptors alone. Genetic ablation of nociceptors fully abrogated reinnervation allodynia. Our results thus reveal the emergence of a form of chronic neuropathic pain that is driven by structural plasticity, abnormal terminal connectivity and malfunction of nociceptors during reinnervation, and provide a mechanistic framework for the paradoxical sensory manifestations that are observed clinically and can impose a heavy burden on patients.

Recent years have witnessed major breakthroughs in our understanding of central-nervous-system-based mechanisms of neuropathic pain, with key contributions attributed to spinal glial activation[8–11] and the cellular circuitry and signalling that mediate the disinhibition of nociception by light touch[7,12–15]. Whether peripheral nerve alterations are causally linked to the maintenance of chronic neuropathic pain or only required as an initial trigger for central plasticity processes remains unclear[3,16]. Chronic pain has seldom been studied in association with nerve regeneration after injury. After physical trauma, damaged peripheral nerve fibres have a limited capacity to gradually regenerate in permissive environments[17–20]. If regeneration is hindered by physical obstacles or a non-conducive local environment, neighbouring undamaged nerve fibres can invade denervated regions through collateral sprouting[17–20]. However, whether this leads to the recovery of normal sensation and disappearance of pain[21], or to the exacerbation of neuropathic pain through hyper-innervation[22] or miswiring of collaterals[18], is controversial and unresolved. Unequivocally disambiguating the role of peripheral nerve regeneration in neuropathic pain has been hindered

so far by the correlative and population-based nature of post-mortem tissue analyses in preclinical and clinical studies, which do not take into account the dynamic nature of changes, inter-individual variability of innervation patterns and connectivity with terminal end organs; these studies also lack causal analyses, thus failing to establish whether peripheral reorganization is a cause or a consequence of neuropathic pain.

## In vivo imaging of sensory innervation

In this study, we now disambiguate the role of peripheral regeneration in chronic pain by overcoming these multiple caveats through longitudinal imaging of specific, defined subsets of sensory afferents. We dynamically visualized in living mice the degeneration, regeneration, collateral sprouting and end organ connectivity of peripheral tactile fibres and nociceptors in the skin before and for 10 months to a year after nerve injury, and combined this with coordinated behavioural assessments of sensory function. Using multiphoton excitation fluorescence imaging, which permits studying deep-seated structures in a minimally invasive

[1]Institute of Pharmacology, Heidelberg University, Heidelberg, Germany. [2]Max Planck Institute for Brain Research, Frankfurt am Main, Germany. [3]Department of Functional Neuroanatomy, Institute for Anatomy and Cell Biology, Heidelberg University, Heidelberg, Germany. [4]Instituto de Neurociencias de Alicante, Universidad Miguel Hernández–CSIC, San Juan de Alicante, Spain. [5]Genomics Core Facility, European Molecular Biology Laboratory, Heidelberg, Germany. [6]Interdisciplinary Center for Scientific Computing, Heidelberg University, Heidelberg, Germany. ✉e-mail: rohini.kuner@pharma.uni-heidelberg.de

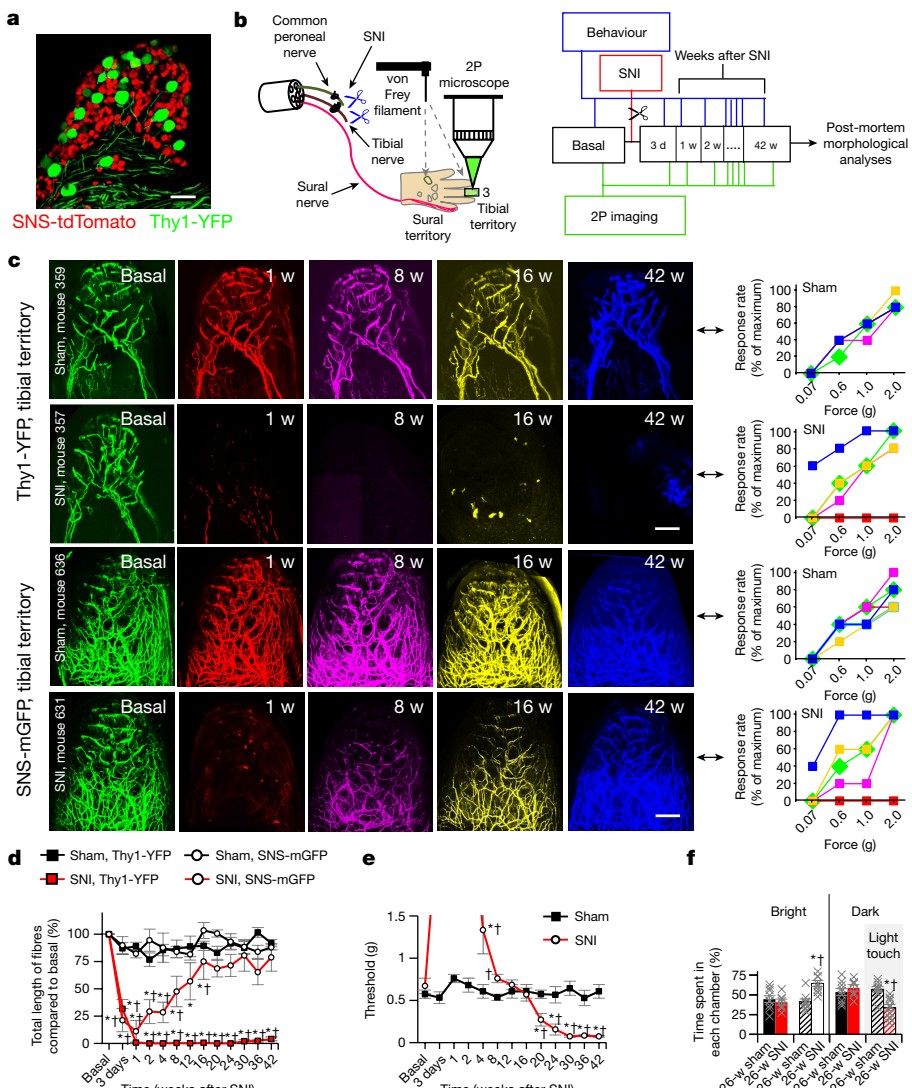

**Fig. 1 | Emergence of chronic neuropathic pain and allodynia after a period of complete loss of sensation as nociceptors repopulate denervated territories whereas tactile fibres do not regenerate. a**, Confocal image of DRGs, showing the segregation of YFP-expressing touch-sensitive neurons (Aβ-LTMRs; Thy1-YFP transgene) and tdTomato-expressing pain-sensing neurons (nociceptors; SNS-Cre transgene) (n = 5). Scale bar, 100 μm. **b**, Experimental scheme of non-invasive two photon imaging longitudinally over 42 weeks and concurrent behavioural analyses in the SNI model, which involves ligation and cutting of the tibial and common peroneal branches while leaving the sural branch intact. 2P, two-photon; d, days; w, weeks; bold number 3 indicates digit 3. **c**, Examples of imaging in individual Thy1-YFP mice and SNS-mGFP mice with concurrent analysis of mechanical sensitivity (von Frey force in grams) at the imaged digit 3 (middle end phalanx) longitudinally over 42 weeks after sham or SNI surgery; matching time points are indicated by using the same false colours in imaging (left) and behaviour (right). Scale bars, 200 μm. **d**, Quantitative summary of total length of

YFP-positive Aβ-LTMRs (square symbols; n = 4 and 4 for sham and SNI, respectively; $F_{(1,12)}$ = 8,614.43, $P$ = 1.63 × 10$^{-18}$) and mGFP-positive nociceptors (circular symbols; n = 4 and 6 for sham and SNI, respectively; $F_{(1,12)}$ = 9.153, $P$ = 0.016; two-way repeated measures ANOVA with Bonferroni multiple comparison) in the tibial innervation territory. **e**, Summary of changes in withdrawal thresholds to von Frey force in tibial territory digit (n = 9 per group; after 20 weeks: $F_{(1,4)}$ = 43.33, $P$ = 0.00223. *$P$ < 0.05 compared to baseline, †$P$ < 0.05 compared to control group (sham); two-way repeated measures ANOVA with Bonferroni multiple comparison). **f**, Aversion to light touch in the previously insensitive tibial territory in the PEAP; the dark chamber was associated with mechanical stimulation (n = 9 for sham and n = 7 for SNI in no stimulation group, and n = 9 for sham and n = 10 for SNI in mechanical stimulation group; $F_{(3,31)}$ = 8.794, $P$ = 0.000228. *$P$ < 0.05 as compared to without mechanical stimulation, †$P$ < 0.05 as compared to sham group; one-way repeated measures ANOVA with Bonferroni comparison). Data are mean ± s.e.m.

manner[23], we tracked pain-sensing high-threshold unmyelinated or thinly myelinated C and A-δ nociceptive fibres[24] in mice expressing fluorescent reporters using the SNS-Cre transgene, which uses promoter elements of the mouse Nav1.8-encoding *Scn10a* gene[25], and visualized touch-sensitive low-threshold mechanosensory afferents of Aβ neurons[7] (Aβ-LTMRs) using a strain of Thy1-YFP mice. In double transgenic SNS-tdTomato:Thy1-YFP mice, a mostly non-overlapping expression of tdTomato and YFP was found (Fig. 1a), with only 2.6% of YFP-positive neurons expressing tdTomato and only 0.4% of tdTomato-positive neurons expressing YFP (Extended Data Fig. 1, Supplementary Note 1), thereby demonstrating a clean segregation of non-nociceptive and

nociceptive neurons. For imaging of the distal skin trajectories and the extremely fine, free nerve endings of nociceptors, we expressed a myristolated membrane-bound form of enhanced green fluorescent protein (mGFP) instead of tdTomato (SNS-mGFP mice; Supplementary Note 1). We were able to repetitively visualize labelled fibres with high fidelity and quantify structural changes in the living mouse stably over several months (Supplementary Video 1). We established an in-house imaging and analytical pipeline of mathematical algorithms and machine learning approaches for thresholding, image processing, segmentation, registration, quantitative analysis and visualization of the complex four-dimensional datasets (Supplementary Figs. 1–7,

Supplementary Note 2). Because mixed injuries with damaged and undamaged nerves particularly frequently lead to neuropathic pain[2], we used the spared nerve injury (SNI) model of neuropathic pain[26] (Fig. 1b). This model allows us to study both damaged and undamaged nerve territories in a segregated manner within an individual animal by focusing on the following regions on the hind paw glabrous surface: (1) injured or denervated territory, which is normally innervated by the tibial and common peroneal nerves that are lesioned and ligated in SNI (for example, digit 3 shown in Fig. 1b), thereby simulating hindrance of normal regeneration; and (2) non-injured territory innervated largely by the sural nerve (lateral edge of digit 5 of the mouse hind paw in Fig. 1b), which is physically intact after SNI, develops exquisite mechanical hypersensitivity[26] (Extended Data Fig. 2a, b) and has been the subject of most studies on neuropathic pain using this model. We longitudinally intercalated nerve fibre imaging with behavioural analyses at the tibial-innervated digit (Fig. 1b).

## Reinnervation and emergence of pain

Figure 1c shows time series of maximum intensity projections of three-dimensional (3D) imaging stacks acquired in vivo from the middle end phalanx (tibial innervated digit) of individual Thy1-YFP and SNS-mGFP mice with corresponding stimulus–response curves to graded mechanical stimuli applied to the same area in the same mice (shown on the right of each row, curves colour-coded to the time of imaging). As expected, within three days to a week after SNI injury, the denervated tibial territory was entirely insensitive to mechanical stimulation at non-noxious (0.07–0.6 g) and noxious (1.0–2.0 g) intensities as both Aβ-LTMRs and nociceptive afferents were lost in Thy1-YFP and SNS-mGFP mice (red-coloured images and behaviour curves in Fig. 1c, Extended Data Fig. 3). Over time, the insensitive tibial territory gradually reacquired responsivity, first to noxious intensities (around 8 weeks after SNI; magenta-coloured curves) and thereafter also to non-noxious mechanical stimulation (by 16 weeks; yellow-coloured curves) as the mice concurrently showed a progressive re-innervation with nociceptive fibres, whereas Aβ fibres were still conspicuously missing in the tibial territory (Fig. 1c, Extended Data Fig. 2b). Notably, starting from week 20 after SNI, both SNS-mGFP and Thy1-YFP mice showed marked hyperalgesia as well as allodynia to mechanical stimuli in the previously insensitive regions of the tibial territory; this was still apparent when the experiment was terminated at week 42 after SNI (blue-coloured curves in Fig. 1c, Extended Data Figs. 2b, 3). Concurrently with this switch from recovery of normal sensitivity to marked hyperalgesia and allodynia as of 20 weeks after SNI, nociceptors were seen to have re-established a dense network, whereas reinnervation with tactile afferents was not detectable in the tibial territory at all (blue-coloured images in Fig. 1c). Supplementary Videos 2–5 (sham groups: 2 and 4; SNI groups: 3 and 5) show examples depicting 3D views of afferent type-specific denervation and reinnervation (or lack thereof) in the tibial territory of Thy1-YFP and SNS-mGFP mice at baseline and 42 weeks after SNI. Mice of both sexes showed similar changes.

Quantitative analyses of total fibre length, representing the cumulative length of the fibres that could be unequivocally traced in the imaged 3D volume common to all imaging sessions (typically 600 μm × 600 μm × 450 μm; $X \times Y \times Z$) in the digit, and behavioural analyses in large cohorts of SNS-mGFP and Thy1-YFP mice confirmed that the maximum observed recovery of nociceptor density (Fig. 1d) is reached just before the time point of functional manifestation of mechanical allodynia (marked drop in threshold of eliciting a withdrawal response in Fig. 1e). By contrast, Aβ-LTMRs failed to emerge in all mice by the end of the experiment at 42 weeks after SNI (Fig. 1d, Extended Data Fig. 3). This mechanical hypersensitivity could not be accounted for by hyper-reflexia and was associated with aversion, as shown by behaviour in the voluntary place escape–avoidance paradigm (PEAP) (Extended Data Fig. 2c). At 24 weeks after SNI, mice showed avoidance of a dark chamber in which they received 0.16 g stimulation in the tibial nerve territory and showed preference for a bright chamber that they would have normally shunned (Fig. 1f, Extended Data Fig. 2d). Sham-treated mice did not develop avoidance, thereby showing that this stimulus is only aversive to SNI mice when applied to the tibial territory (Fig. 1f, Extended Data Fig. 2d). Significant cold allodynia was also observed in the tibial territory, although to a lesser extent than in the sural territory (Extended Data Fig. 2e). Together, this progressive shift from a lack of sensation to exaggerated nociceptive sensitivity, allodynia and aversion in a denervated region unmasks an emergence of late-onset, chronic neuropathic pain. Whereas most neuropathic pain studies have addressed pain in uninjured domains when nerves undergo partial or mixed injury, our findings now reveal pain that develops through regeneration of fibres into the denervated component of mixed injuries. Notably, these are clinically particularly frequently associated with neuropathic pain and a high burden of suffering for patients[2,27]. We therefore refer to this phenomenon as reinnervation-induced neuropathic pain and show that it is associated with the re-emergence of nociceptors, but not Aβ-LTMRs.

In particular because these implications differ strongly from the prevailing view of the paramount importance of Aβ-LTMRs in neuropathic allodynia that is studied in the uninjured domains[4–7], we included several control experiments. First, photo damage to afferents was ruled out by the observed stability of innervation patterns of both nociceptive and non-nociceptive fibres over several months in sham-operated mice (Fig. 1c, d, Supplementary Videos 2, 4). Second, post-mortem immunohistochemical analyses with endogenous marker proteins for sensory afferents confirmed the loss of NF200-expressing large-diameter fibres and the recovery of CGRP-expressing nociceptors (Extended Data Fig. 4), thus ruling out potential caveats through pathophysiological alterations in the expression of the transgenic fluorescent reporters. Third, when SNI mice that showed reinnervation with mGFP-positive nociceptors at 42 weeks after SNI were subjected to severance and ligation of the sural nerve, all mGFP fluorescence was lost from the tibial territory (Extended Data Fig. 5), thus demonstrating that fibres invading the tibial territory in conjunction with the emergence of neuropathic pain are nociceptors sprouting as collaterals from the neighbouring, intact sural nerve.

## Differences in fibre trajectories

We next sought to address the question of why sural nociceptors enter the denervated tibial territory but Aβ-LTMRs cannot. Previous studies suggest that unmyelinated nerve fibres have an overall higher regenerative capacity than thickly myelinated fibres. This can be attributed (1) to differences in the types of axon itself; for example, a study in *Drosophila* implicates Piezo—an ion channel essential for mechanotransduction that is expressed in Aβ LTMRs more prominently than in nociceptors—as an inhibitor of axonal regeneration[28]; or (2) to the different types of Schwann cells that support axonal growth and regenerative growth by forming conduits. Chronic denervation has been linked to a decreased capacity of myelinating Schwann cells to support regenerating axons[19,29,30] and a maintenance of expression of markers of immature states—for example, the p75 neurotrophic receptor—specifically in Remak Schwann cells that ensheath smaller unmyelinated axons (such as nociceptors), which may thereby facilitate regenerative support[20]. In terms of collateral sprouting, recapitulating growth trajectories and distal connectivity is critical for achieving functional recovery. In this regard, we made two observations. First, we noted that the 3D architecture of nociceptor innervation of tibial nerve nociceptors under baseline conditions could be nearly perfectly registered with that of sural nerve nociceptors that have invaded the tibial territory at 42 weeks after SNI (Fig. 2a, Supplementary Video 5), showing that unrelated nociceptors from a different nerve are capable of exactly retracing the trajectories of the original nociceptors during

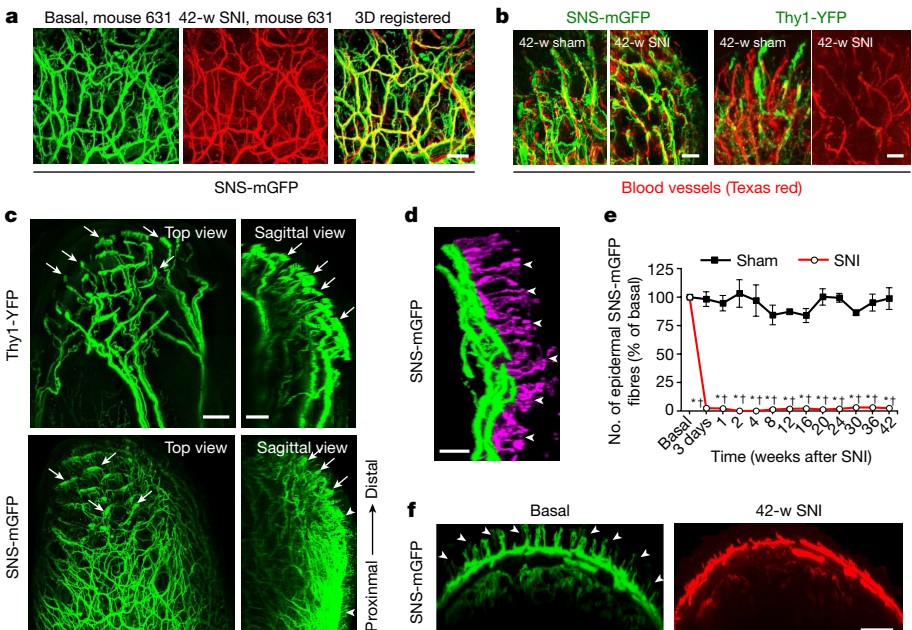

**Fig. 2 | During the emergence of reinnervation-induced neuropathic pain, collaterally sprouting nociceptors of sural origin precisely replicate the original trajectories of tibial nociceptors along blood vessels but do not invade the epidermis. a**, Comparison of the original innervation with tibial nociceptors (green) and invading sural nociceptors (false coloured in red) at 42 weeks after nerve injury; representative images of mouse 631, out of 6 similar results from experiments in 6 mice. Scale bar, 50 μm. **b**, Dual-colour multiphoton imaging of Texas-red-dextran-labelled blood vessels with YFP-positive Aβ fibres (right) and GFP-labelled nociceptors (left) in the tibial territory. Scale bars, 25 μm. Representative images, out of 6 similar results from experiments in 3 sham and 3 SNI mice. **c**, Processed 3D stacks of in vivo multiphoton images showing complete innervation pattern of a single hind paw digit by YFP-positive Aβ-LTMRs and mGFP-expressing nociceptors; arrows indicate specialized endings with Meissner-corpuscle-like morphology and arrowheads indicate intra-epidermal free endings (n = 4). Scale bars, 100 μm. **d**, Segmentation of intra-epidermal free endings (false coloured in purple; arrowheads) of nociceptors from their afferent branches (green) (n = 4). Scale bar, 25 μm. **e, f**, Quantitative summary (**e**) and typical example (**f**) demonstrating the lack of epidermal invasion (arrowheads in **f**) of collaterally sprouting sural nociceptors in the tibial territory, expressed as a percentage fraction of baseline values (n = 4 and 4 for sham and SNI, respectively; groups $F_{(1,12)} = 1,429.65$, $P = 7.53 \times 10^{-14}$; two-way repeated measures ANOVA with Bonferroni multiple comparison). *$P < 0.05$, as compared to basal values; †$P < 0.05$, as compared to sham group. Data are mean ± s.e.m. Scale bar, 100 μm.

collateral sprouting. In a second set of experiments, by acutely labelling blood vessels in the paw at 42 weeks after SNI through intravenous injection of high-molecular-weight dextran labelled with Texas red, we observed that the small distal branches of the collaterally sprouting mGFP-labelled nociceptors were intertwined with small blood vessels (Fig. 2b). By contrast, Thy1-YFP-labelled Aβ fibres showed trajectories that were distinct from blood vessel patterning in control mice and were absent from the tibial territory while intact blood vessels were detected (Fig. 2b). Because nociceptors in control mice without SNI were also observed to be in contact with blood vessels, and because blood vessels were not lost when mice underwent SNI, this suggests that nociceptors that are collaterally invading denervated areas can use small blood vessels as scaffolds to recreate the original patterning of trajectories in their distal projection zones. Structural neurovascular interactions represent an emerging field in developmental sciences, wherein nerve-derived cues—such as VEGF—are suggested to guide blood vessels over early embryonic life[31]. Our observations now reveal that morphological trajectories of blood vessels and regenerating small-diameter nerves are replicated in the context of adult regeneration, and implicate blood vessels in providing a scaffold for collaterally sprouting nociceptors in denervated tissue.

## Aberrations in terminal connectivity

We next investigated why collateral sprouting of nociceptors into the denervated territory results in allodynia rather than the simple restoration of normal nociception. Ultimately, connectivity to end organs that transduce diverse types of external sensory stimuli onto distinct types of nerve fibres in the skin is key to functional selectivity

and discernment between pain and touch, and there has been a lot of interest and progress in this regard[32]. Indeed, we found marked defects in the terminal connectivity and function of collaterally sprouting nociceptors that can account for the switch from normal nociception to allodynia. In SNS-mGFP mice, the intricate meshwork of mGFP-labelled, mostly thin fibres terminated in tiny free nerve endings in the epidermis of the skin that are characteristic to nociceptors (Fig. 2c, Supplementary Video 6), which we could segment out in the epidermis using a custom-developed algorithm (Fig. 2d; see Extended Data Fig. 6, Supplementary Note 2 for details). In contrast to the rich innervation of the epidermis by intra-epidermal free nerve endings under baseline conditions (Fig. 2c, d), collaterally sprouting nociceptors did not enter the epidermis of the tibial territory and were restricted at the basal membrane separating the dermis from the epidermis all the way up to the termination of the experiment at 42 weeks after SNI (quantitative summary in Fig. 2e and examples in Fig. 2f). That this is not an artefact of transgenic reporter expression was confirmed by immunostaining for native markers of nociceptors, such as CGRP, in the same tissue (Extended Data Fig. 4b). Thus, reinnervation-induced neuropathic allodynia was associated with an invasion of nociceptors in denervated areas that paradoxically lacked intra-epidermal free nerve endings, thereby reproducing a hallmark feature of several types of highly painful human C-fibre neuropathies[3].

Another notable observation came with respect to afferent terminations on Meissner corpuscles, which transduce touch. As expected, thick, YFP-labelled Aβ afferents were found to end in oval-shaped terminations in the glabrous skin that are characteristic of Meissner corpuscles (Figs. 2c, 3a, Supplementary Videos 6, 7), which was confirmed by labelling for S100—the protein that labels the glial sheath of Meissner

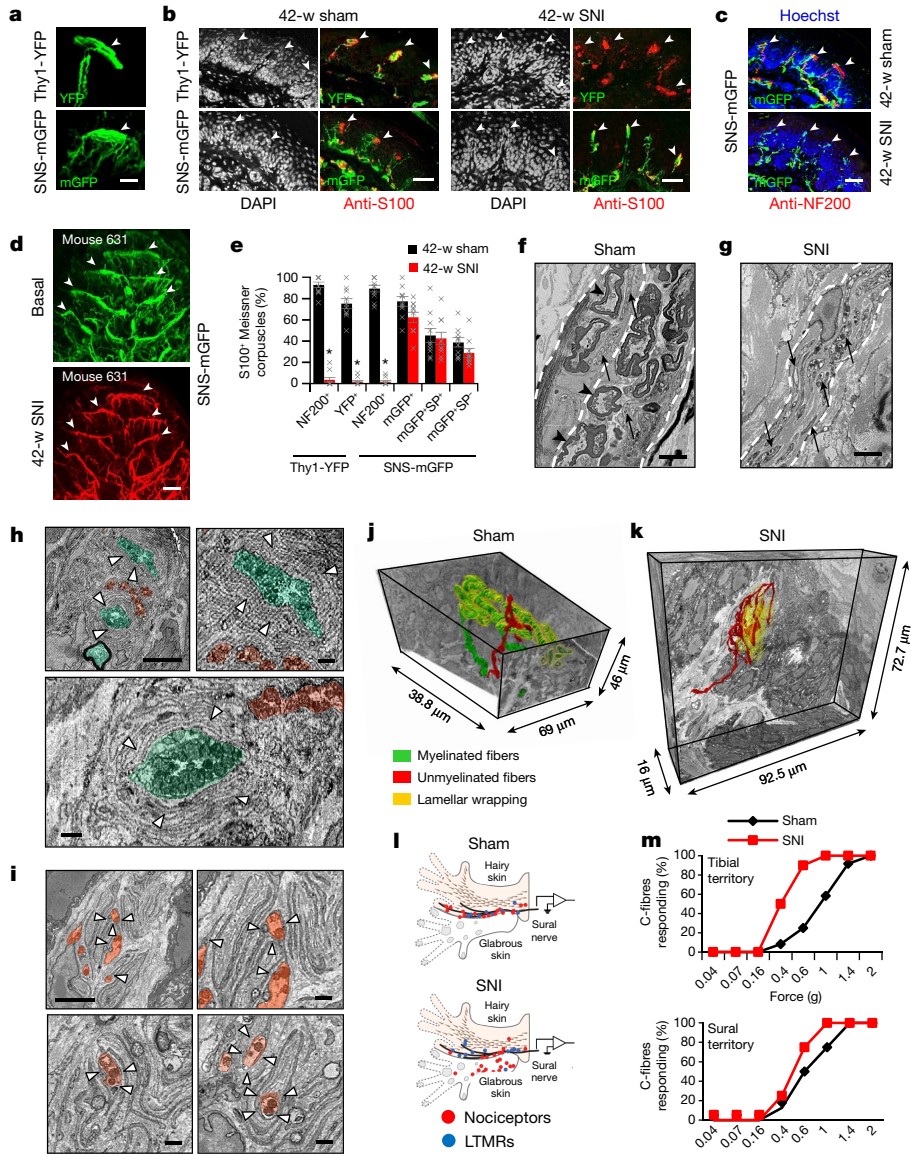

**Fig. 3 | Nociceptors pathologically switch to a tactile low-threshold fibre phenotype after collateral sprouting into denervated skin.**
**a**–**d**, High-magnification in vivo multiphoton images (**a**, **d**) and confocal images (**b**, **c**) of YFP-positive Aβ-LTMR fibres (expressing NF200 in **c**; $n$ = 4) and mGFP-expressing nociceptors (bottom) at Meissner-like structures (arrowheads; expressing S100; $n$ = 4) and nociceptor free endings in the tibial territory of control mice and after SNI (**b**–**d**). Scale bars, 25 μm (**a**); 50 μm (**b**, **c**); 100 μm (**d**). **e**, Quantitative overview of sensory afferent terminations at S100-expressing Meissner corpuscles in the tibial territory at 42 weeks after SNI or sham ($n$ = 10 per group; for groups $F_{(1,5)}$ = 225.321, $P$ = 2.37 × 10⁻⁵; two-way ANOVA with Bonferroni multiple comparison). SP, substance P. **f**–**k**, Ultrastructural high-resolution 3D analyses in the tibial territory 24–28 weeks after SNI (**g**, **i**, **k**) or in control mice (**f**, **h**, **j**) showing dermal nerves (**f**, **g**) and Meissner corpuscles (**h**–**k**). Images show myelinated axons (black arrowheads) and their terminations (false coloured in green) or unmyelinated axons (black

arrows) and their terminations (false coloured in red) and glial cell lamellae (white arrowheads) in Meissner corpuscles ($n$ = 3 SNI and 3 control mice). Scale bars, 5 μm (**f**, **g**, **h** (top left), **i** (top left); 1 μm (**h** (top right and bottom), **i** (top right and bottom). **j**, **k**, Full 3D reconstruction of Meissner innervation showing terminations of unmyelinated fibres (false coloured in red), myelinated fibres (false coloured in green) and glial cell lamellar wrapping (false coloured in yellow) in control (https://wklink.org/8342) and SNI (https://wklink.org/8231) mice. **l**, Electrophysiological single-fibre recordings demonstrating receptive fields of C-fibres (red dots) and Aβ-LTMRs (blue dots) in the sural nerve after stimulation of the tibial territory ($n$ = 12 fibres each from 3 sham and 3 SNI mice). **m**, Single-fibre recordings showing C-fibre recruitment by tactile stimuli in denervated tibial territory, but not in the intact sural territory ($n$ = 8 fibres each from 6 control and 5 SNI mice). $P$ values derived from chi square analysis for tibial territory ($P$ = 1.25 × 10⁻⁶) and for sural territory ($P$ = 0.1255). Data are mean ± s.e.m.

corpuscles (Fig. 3b). However, several mGFP-labelled nociceptive fibres were also found to terminate at the border between the epidermis and the dermis (Figs. 2c, 3a) and directly apposed S100-expressing Meissner corpuscles (Fig. 3b). In contrast to Aβ fibres, which innervate the Meissner corpuscles by coiling within the S100-expressing glial sheath, our imaging experiments showed that nociceptors form a sparse transverse meshwork of endings that colocalize with glial cells of Meissner

corpuscles (Fig. 3a, Extended Data Fig. 7; see Supplementary Video 7 for a 3D view). For decades, the largest body of literature on Meissner corpuscles referred to their exclusive innervation by myelinated, low-threshold tactile afferents[7,33], which were recently reported to segregate into two phylogenetically distinct populations of Aβ neurons[34]. However, a handful of reports have described the 'intracorpuscular' location of small-diameter unmyelinated fibres in human skin[35–37] and

in New World monkeys[38] and rodents[39]; these fibres have not received attention and their functional importance is completely unknown so far.

Of note, in the denervated tibial territory at 42 weeks after SNI, Meissner corpuscles showed a loss of connectivity with Aβ-LTMRs, which was confirmed by loss of labelling with the native LTMR marker, NF200 (Fig. 3c, Extended Data Fig. 8a). In contrast to the control situation, Meissner zones were now observed to be solely associated with nociceptive fibre terminals, as seen with mGFP-labelled terminations (note high-fidelity spatial recovery of original pattern with collaterally sprouting nociceptors in Fig. 3d, Supplementary Video 5) and with the native labelling of nociceptors with CGRP (Extended Data Fig. 8b). Quantitative analyses confirmed a marked loss of Meissner innervation by NF200-expressing fibres (Fig. 3e), and showed peptidergic and non-peptidergic nociceptors in apposition to S100-positive Meissner corpuscles (Fig. 3e, Extended Data Fig. 8c). To address the question of whether this reflects aberrant connectivity in the end organ, we performed serial block-face scanning electron microscopy to generate high-resolution representations of terminal connectivity within the tibial territory at 24 weeks after SNI. In contrast to control mice, myelinated fibres were conspicuously absent in the dermis of SNI mice, and unmyelinated axons comprised the only nerves present post-innervation (Fig. 3f, g), thus further validating our light microscopy findings. In Meissner corpuscles of control mice, the terminals of myelinated afferents were surrounded by concentric patterning of the glial cell lamellae that are critical for sensing mechanical pressure[34], and terminals of unmyelinated C-fibres were found at the outer edges of these lamellar cushions (Fig. 3h). By contrast, in the tibial territory after SNI, collaterally sprouted unmyelinated afferents showed multiple terminations dispersed in between loosely patterned lamellae and made close contacts with lamellae, with some occupying the centre of lamellar cushions (Fig. 3i)—a position only taken by Aβ fibre terminations in control mice (Fig. 3h). This finding was further validated by full 3D high-resolution reconstructions of Meissner corpuscles from control and SNI mice (images shown in Fig. 3j, k, respectively; 3D views shown in Supplementary Videos 8, 9, respectively): in control mice, myelinated afferents showed a classical patterning of winding throughout the Meissner corpuscles and were fully covered by glial cell lamellae, whereas the unmyelinated afferents did not branch extensively inside the Meissner volume and were not surrounded by concentric lamellae. In the tibial territory of SNI mice, in the absence of Aβ terminations, the collaterally sprouted unmyelinated afferents were observed to invade throughout the volume of the Meissner corpuscles, elaborately meandering in between and being surrounded by glial cell lamellae. This morphological pattern shows abnormalities of end organ connectivity and places unmyelinated afferents, as the only nerve afferent component found inside the Meissner corpuscles, physically in a position to sense changes induced by the indentation of lamellae after mechanical stimulation.

Unmyelinated C-fibres also include a subpopulation of non-peptidergic C-fibres—namely C-LTMRs—that respond to light touch in a similar manner to Aβ-LTMRs[40,41]. We therefore also tested the possibility that C-LTMRs, which typically innervate hair follicles, sprout into the denervated glabrous tibial territory by immunostaining with anti-tyrosine hydroxylase (TH) (but see Supplementary Note 3). However, only 2 of 8 mice tested at 42 weeks after SNI showed the ectopic presence of TH-positive fibres at the dermal–epidermal border (Extended Data Fig. 9, Supplementary Note 3). Our analyses thus do not provide adequate evidence to suggest that C-LTMRs sprout into the sub-epidermal zone in denervated glabrous skin. Because TH also labels adrenergic fibres, this also negates the involvement of sympathetic structural remodelling.

## C-fibres switch to tactile responsivity

We next investigated whether the remodelling of peripheral connectivity leads to functional abnormalities in nociceptors that invade into the denervated tissue. We performed electrophysiological recordings of Aβ-LTMRs and C-nociceptors at 24 weeks after SNI using the skin–nerve preparation, which was modified to span the glabrous sural as well as tibial territories. In the first set of recordings, only the stimulation of the sural territory evoked responses in sural nerve fibres in sham-treated mice (Fig. 3l). By contrast, mice at 24 weeks after SNI exhibited marked responses in sural nerve fibres when the tibial nerve glabrous territory was mechanically stimulated (Fig. 3l). These newly acquired tibial territory receptive fields for sural afferents in SNI mice were found only for nociceptors, and not for Aβ-LTMRs (with only one exception; Fig. 3l). These data thus independently and functionally verify our findings that nociceptors—but not Aβ-LTMR fibres—from the sural nerve repopulate the denervated tibial territory. In a separate set of experiments, we measured the activation thresholds of C-fibres, identified using conduction velocity measurements (Extended Data Figs. 10a, c), and observed that sural nerve C-fibre responses that were mechanically evoked from the tibial territory of SNI mice showed significantly lower thresholds as compared to physiological innervation (that is, tibial nerve C-fibre responses from the tibial territory of control mice) (Fig. 3m, Extended Data Fig. 10b). Mechanical stimulation of the undamaged sural territory evoked C-fibre responses in the sural nerve, with a tendency towards a lowering of the activation threshold after SNI compared to controls, although this tendency was not statistically significant (Fig. 3m, Extended Data Fig. 10d, e). These data reveal that collaterally sprouting nociceptors have altered activation properties such that they respond to innocuous mechanical stimuli like low-threshold afferents do, and suggest that they acquire this property during the reinnervation of the tibial territory.

An inflammatory milieu in the denervated tibial territory and persistent peripheral sensitization could be responsible for changes in nociceptor sensitivity during reinnervation. Several observations speak against this possibility. First, we did not find any remaining accumulation of immune cells, such as macrophages or T cells, in the tibial territory when neuropathic pain was manifested after 24 weeks (Extended Data Fig. 11). Second, pharmacological inhibition of classical sensitization-associated mediators such as prostaglandins, TRPV1 and TRPA1, or sequestration of nerve growth factor (NGF), did not block reinnervation neuropathic allodynia (Extended Data Fig. 12). Third, unbiased gene transcriptional analyses performed on the somata of dorsal root ganglia (DRGs) of collaterally sprouting nociceptors, which were identified through retrograde labelling from the tibial territory at 24 weeks after SNI, did not reveal the regulation of key sensitization-associated genes (see Supplementary Note 4, Extended Data Fig. 13 for details). This suggests that nociceptor sensitization phenomena that are hallmarks of other types of pain, such as acute and chronic inflammatory pain, are not causing the observed change in mechanical thresholds upon reinnervation. Rather, most of the genes that showed altered expression in collaterally sprouting nociceptors over controls were found to be linked to structural changes, such as NGF-dependent neurite outgrowth, axonal pathfinding, neuroprotection and axonal survival (Extended Data Fig. 13, Supplementary Note 4), thus further indicating structural remodelling.

## Nociceptors mediate reinnervation pain

Finally, we addressed whether nociceptors are causally responsible for reinnervation-induced neuropathic pain developing in previously denervated and insensitive areas by using diphtheria toxin (DTX)-mediated ablation of nociceptors (Fig. 4a; 98% loss of non-peptidergic nociceptors, 70% loss of peptidergic nociceptors and a statistically non-significant 20% reduction in NF200-expressing neurons after treatment with DTX). Consistent with nociceptor ablation, DTX-treated mice in the absence of nerve injury (sham) showed significantly reduced basal responsivity to mechanical von Frey stimuli in the nociceptive range (that is, 1.0 g and above; Fig. 4c, d). Measurements of mechanical

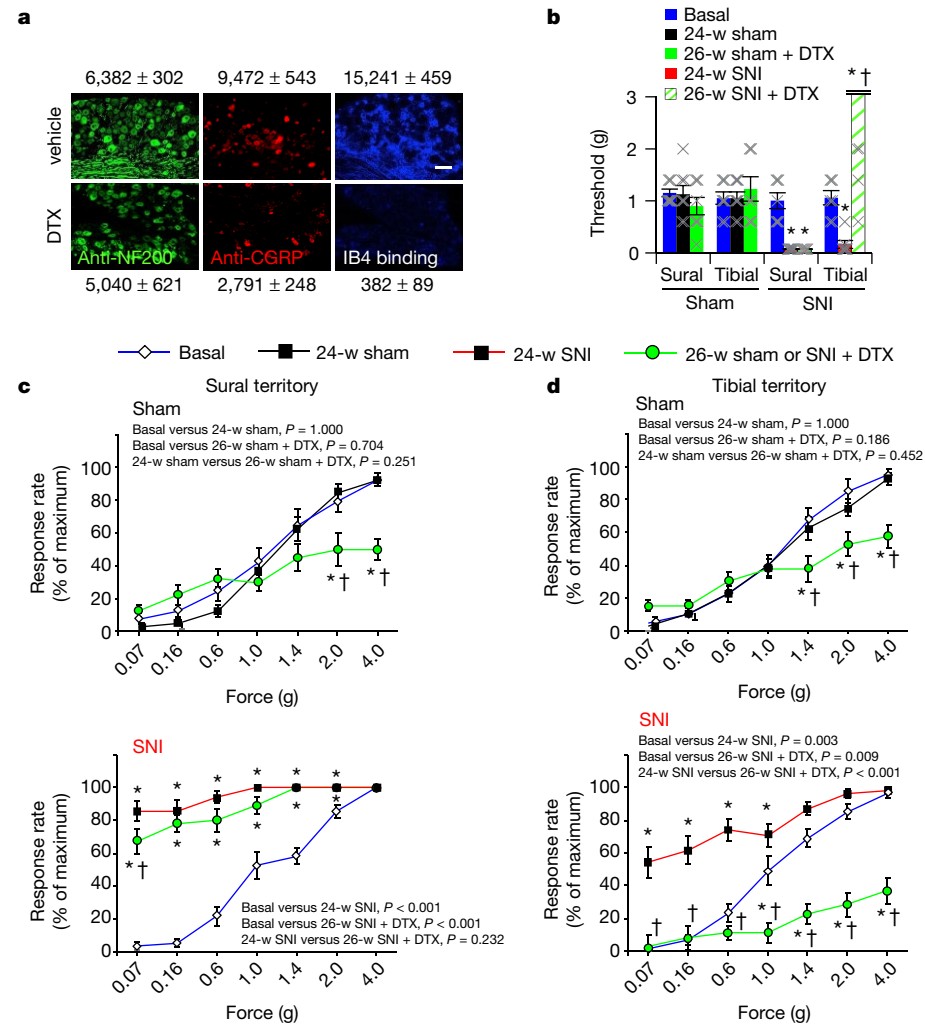

**Fig. 4 | Genetic ablation experiments reveal a causal role for nociceptors in reinnervation-induced chronic neuropathic allodynia. a**, In mice with DTX-induced ablation of nociceptors (SNS-DTR), examples and quantitative estimate of large-diameter NF200-positive neurons (left; $n = 3$ per group, degrees of freedom (DF) = 2.897, $t = 1.941$, $P = 0.124$), CGRP-positive peptidergic nociceptors (middle; $n = 3$ per group, DF = 2.802, $t = 11.179$, $P = 0.000365$) and non-peptidergic isolectin B4 (IB4)-binding nociceptors ($n = 3$ per group, DF = 2.149, $t = 31.750$, $P = 5.87 \times 10^{-6}$) after treatment with vehicle or DTX (two-tailed $t$-test). Numbers of DRG neurons from three mice per treatment are shown. Scale bar, 100 μm. **b–d**, Nociceptor ablation significantly decreases mechanical nociception and reverses reinnervation-induced allodynia in the tibial territory (**b**, **d**), but not allodynia in the intact sural territory (**b**, **c**), as demonstrated by analysis of response thresholds (**b**) and response rates (**c**, **d**) to mechanical stimulation. In **b**, $n = 8$ per group; groups $F_{(3,6)} = 25.234$, $P = 0.000842$; treatment $F_{(2,6)} = 10.001$, $P = 0.0122$; groups × treatment $F_{(6,56)} = 12.525$, $P = 6.74 \times 10^{-9}$. In **c** (sural territory): $n = 8$ per group; for sham (top part): groups $F_{(2,12)} = 1.798$, $P = 0.202$; for SNI (bottom part): groups $F_{(2,12)} = 93.888$, $P = 4.697 \times 10^{-8}$). In **d** (tibial territory): $n = 8$ per group; for sham (top part): groups $F_{(2,12)} = 2.226$, $P = 0.145$; for SNI (bottom part): groups $F_{(2,12)} = 29.514$, $P = 2.32 \times 10^{-5}$); these $P$ values correspond to group $P$ values ($P$ values shown within the figure refer to comparisons between pairs of treatment group (that is, individual pairs from three different groups)). In all panels, *$P < 0.05$ compared to baseline, †$P < 0.05$ compared to before DTX application, two-way repeated measures ANOVA with Bonferroni multiple comparison in **b**, **c**. Data are mean ± s.e.m.

thresholds after SNI showed that nociceptor ablation reversed mechanical allodynia in the denervated territory of the tibial nerve (Fig. 4b, d), but not in the territory of the uninjured sural nerve (Fig. 4b, c). To rule out that the observed phenotypic differences arise from the ablation of any cells at loci other than DRG neurons in SNS-Cre mice, we directly injected adeno-associated virions expressing caspase-3 in a Cre-dependent manner or EFGP as a control into the L3–L4 DRGs of SNS-Cre mice 24 weeks after SNI. In contrast to EGFP-expressing mice, caspase-3-mediated ablation of nociceptors completely abrogated mechanical hypersensitivity in the tibial territory, but only partially and transiently reduced hypersensitivity in the sural territory (Extended Data Fig. 14). These findings thus conclusively demonstrate that the emergence of reinnervation-induced neuropathic allodynia is mediated by nociceptors. This is in sharp contrast to allodynia reported in uninjured territories, in which loss- or gain-of-function experiments show that Aβ afferents are both necessary and sufficient to induce mechanical allodynia[5,6] and allodynia develops even when nociceptors are genetically ablated[42].

Together with the current knowledge on neuropathic pain[5,7,8,13,42], our findings reveal the existence of two completely different forms and mechanisms of mechanical allodynia after nerve injury, namely: (1) a well-studied, Aβ-fibre-dependent form with an early onset and chronic nature arising from uninjured nerve territories; and (2) a new form described here, which is found in denervated territories of injured nerves, is nociceptor-dependent, becomes manifest with a late onset during the reinnervation process, and has been consequently overlooked in most preclinical work on neuropathic pain (summarized in Extended Data Fig. 15). These findings thus fill a long-standing gap in the

representation of the complex clinical state of neuropathic pain arising from mixed nerve crush injuries that are clinically more frequently associated with chronic neuropathic pain[2,27], and indicate that both forms of allodynia should be investigated in studies of the mechanisms and therapy of pain resulting from nerve injury. Of note, the findings also bridge the fields of regeneration and chronic pain, which are mostly studied independently of each other in mainstream literature, to highlight the importance of structural plasticity and miswiring during reinnervation processes, and thus pave the way for addressing a new direction in neuropathic pain. Our findings inadvertently highlight and help to rationalize observations in a historically important experiment on nerve regeneration and pain in humans[43], which involved repeated sensory testing over four years after a self-inflicted experimental nerve injury in a healthy individual. Reporting in 1908, the authors suggested that there are two different types of 'sensation'—protopathic and epicritic—that recover at different rates after nerve injury; an early recovery of protopathic sensation causes "exquisite pain and hyperaesthesia", whereas normal sensation is restored only after a late recovery of epicritic sensation[43,44]. Our longitudinal structure–function analyses reveal the cellular identity and ensuing alterations in the terminal connectivity of the underlying classes of afferents and help to establish causality for reinnervation-induced neuropathic pain. One possibility is that this miswiring and abnormal connectivity studied at the peripheral end of sensory neurons ties in functionally with aberrant discharges that have been reported in DRG somata[1], as well as anatomical alterations and misconnectivity that have been proposed to occur at central terminals[45] after nerve injury.

Our study has also addressed several mechanistic possibilities for the manifestation of reinnervation-induced neuropathic pain. First, the lowered activation thresholds that we observed in collaterally sprouting nociceptors may result from 'sensitization memory' or 'priming'[46,47], given their origin in the 'spared' sural nerve that is subject to sensitizing influences by the inflamed milieu in the paw after nerve injury despite being morphologically intact. However, at the time of manifestation of reinnervation-induced neuropathic pain, we observed neither overt sensitization in nociceptors in the sural territory nor inflammatory changes in the tibial or sural territory. Moreover, pharmacological studies and gene expression analyses did not support the occurrence of classical peripheral sensitization by molecular mediators. Second, selective sprouting and reinnervation with sub-classes of nociceptors with particular properties supporting neuropathic pain—for example, the NP1 cluster amongst non-peptidergic nociceptors[48]—is a possibility. However, our gene expression analyses did not reveal any differences in the relative abundance of the various subclasses of nociceptors that innervate the tibial territory between sham and neuropathic mice. Third, our observations on the peripheral terminations of fibres implicate miswiring as a basis for the abnormal functional properties of collaterally sprouting nociceptors. Failure to enter the epidermis may lead to a concentrated localization of sensory transducers and excitatory ion channels at nociceptor endings under the epidermal border instead of being spread over the original thin, long filopodia-like free endings over the epidermis, leading to altered activation profile and dynamics. This irregular patterning of nociceptor terminals is also likely to change their functional connectivity with specialized cutaneous mechanosensitive glial cells at the sub-epidermal border, which are proposed to transmit nociceptive signals[49]. Consistently, ablation of these specialized Schwann cells was recently reported to cause the retraction of epidermal nerve fibres and induce allodynia[50]. Our ultrastructural morphological observations on atypical Meissner connectivity during reinnervation also raise the idea that remodelling of Meissner corpuscles anatomically links light touch to nociceptor activation in neuropathic pain, essentially 'converting' nociceptors to low-threshold afferents. Together with recent studies on end organ connectivity and function in the periphery[7,34,51,52], our results highlight the emerging concept that sensory specificity is determined by the properties of the specialized sensory 'organelles' in the skin, rather than by the nerve fibres themselves. Fourth, given that Aβ-LTMRs inhibit spinal second-order nociceptive neurons by recruiting interneurons under physiological conditions[7,12,13,15,53], a lack of Aβ input from denervated tissue in the presence of restored nociceptor input would be expected to disinhibit nociceptive transmission, thereby opening the spinal 'gate' for pain[54–56], as well as altering the top-down control of spinal nociception by brain centres[57]. The results of this study, when taken together with current knowledge, particularly support peripheral miswiring and central disinhibition as mechanisms that are likely to underlie the manifestation of reinnervation-induced neuropathic pain; our findings thus meaningfully integrate the hypothesis of peripheral structural remodelling with important concepts with regard to the plasticity of spinal circuits in neuropathic pain. Ultimately, understanding the cellular mechanisms and molecular cues that lead to faulty regeneration or aberrant rewiring of the different classes of sensory afferents will help to prevent or reverse chronic neuropathic pain after nerve injury.

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

## Methods

### Genetically modified mice

Cre reporter mice carrying either mGFP[58] or tdTomato (Jacksons lab; stock no. 007909) flanked by a transcriptional stop cassette in the Rosa26 locus were crossed with SNS-Cre mice[25,59], in which Cre recombinase is transgenically driven by 110-kb promoter elements of the mouse Nav1.8-encoding *Scn10a* gene to obtain SNS-mGFP or SNS-tdTomato mice that express fluorescent marker in peripheral nociceptive sensory neurons. Mice expressing YFP under the control of the Thy1 promoter (Thy1-YFP 16) were obtained from The Jackson Laboratory; stock no. 003709. In these mice, non-nociceptive sensory neurons are labelled. SNS-iDTR mice were obtained by crossing SNS-cre mice with mice carrying simian diphtheria toxin receptor (DTR) flanked by a transcriptional stop cassette in the Rosa26 locus[60]. For ablating nociceptors, SNS-iDTR mice were injected intraperitoneally (i.p) with 40 μg kg$^{-1}$ of DTX (Sigma D0564) twice, at an interval of 72 h. After five days, mice were acclimatized to behavioural set-up in sessions over at least two days, and behavioural analyses of baseline nociceptive sensitivity were done from day 8 after the second DTX injection, followed by nerve injury. Mice of all genotypes were backcrossed to the C57bl6 background for more than eight generations before crossing with each other. All of the animal experiments were conducted according to the ethical guidelines of the 'Protection of Animals Act' under supervision of the 'Animal Welfare Officers' of Heidelberg University and were approved by the local governing body named 'Regierungspräsidium Karlsruhe: Abteilung 3 - Landwirtschaft, Ländlicher Raum, Veterinär- und Lebensmittelwesen', Germany (approval numbers: G-206/11 and G-177/17). ARRIVE guidelines were followed. Sample sizes were based on previous experience with G-power analyses. Only adult mice (older than 8 weeks) were used. Mice of both sexes were tested in imaging, behavioural and electrophysiological experiments and in all experiments, mice were randomized and experimenters were blinded to the identity of the treatment groups.

### Pain model and behavioural analyses

**SNI.** In this pain model, two of the three branches of sciatic nerve, tibial and common peroneal nerve were cut leaving the sural branch intact as described in detail previously[26].

**von Frey test.** Mechanical hyper- or hyposensitivity induced by SNI was measured using von Frey filaments of increasing strength starting from 0.07 g to 4 g, applied to the digits or hind paw region of various skin dermatome (sural and tibial) as described previously[26].

**PEAP.** Mice were placed in a bright–dark chamber set-up with free access to both bright and dark sides. On day 1, mice were allowed to explore freely on both sides for a period of 20 min that allows us to assess their preference towards a particular chamber. Twenty-four hours later, mice were again placed in the same chamber and this time a von Frey filament of 0.16 g strength was applied to the tibial area of the hind paw at an interval of at least 10 s whenever mice entered the dark side. The time spent on either side of the chamber for a total period of 20 min was measured and the decrease in time spent or active escape or avoidance of the chamber in which mechanical stimuli applied directly reflects pain sensitivity. ANY-maze software was used for tracking mice.

### Application of pharmacological agents

Mechanical hypersensitivity was tested in mice at 24 weeks after SNI or sham surgery or 24 h after injection of complete Freund's adjuvant (CFA), before and 30 min after intraplantar application of the following drugs, each in a volume of 20 μl: diclofenac (D6899, Sigma Aldrich; 50 μg), celecoxib (PHR1683, Sigma Aldrich; 150 μg), AMG 9810 (A2731, Sigma Aldrich; 20 μg), AP-18 (A7232, Sigma Aldrich; 4.2 μg) and tanezumab (anti-NGF antibody; TAB-111, CreativeBiolabs; 30 μg).

### Intravital two-photon imaging

Mice were anaesthetized by using a narcotic mix consisting of 60 μl medetomidine (1 mg ml$^{-1}$, Pfizer), 160 μl midazolam (5 mg ml$^{-1}$, Hameln) and 40 μl fentanyl (0.1 mg ml$^{-1}$, Janssen) at a dosage of 3.1 μl per g body weight. The entire paw was embedded in 2% low-melting agarose (A6013; Sigma) in a small custom-built well with hind paw skin exposed to a 25× water immersion objective (Nikon MRD77225; numerical aperture of 1.1) of an upright TriM Scope II microscope (LaVision BioTec). A femtosecond pulsed Ti:sapphire laser (Chameleon Ultra II; Coherent) was tuned to 960 nm for multi-photon excitation of mGFP, YFP or Texas red. The laser power was linearly increased from 5% at the surface to 15% at the maximal depth to improve acquisition from deep imaging layers (Imspector software, LaVision BioTec[61]). The emitted light was split by a 575-nm dichroic mirror and filtered by a 500–550-nm bandpass filter for the green channel and 585–635-nm bandpass filter for the red channel. To cover the entire end phalanx, four adjacent images were acquired sequentially for each focal plane at 10% overlap. This was repeated for each *z*-focal plane of the stack, yielding image stacks of 450 μm × 450 μm × 500 μm at a voxel size of 427 nm × 427 nm × 1,000 nm for each of the quadrants. These were then merged into one 'superstack' covering a volume of 854 μm × 854 μm × 500 μm (see (i) in 'Automated workflow for processing of large-scale two-photon imaging data'). Blood vessels were labelled by retro-orbital injection of 50 μl of Texas-red-labelled dextran (D1830; molecular weight, 70,000 D; molecular probes). ImageJ was used for assessing the quality of the images.

### Automated workflow for processing of large-scale two-photon imaging data

The vast amount of data (more than 800 3D image stacks corresponding to approximately 35 TB of raw data) necessitated automated image analysis and use of a high-performance cluster for computation (bwMLS&WISO). Our automated analysis workflow consists of five main steps as described below and further detailed in Supplementary Note 2. Processing and quantitative analysis were performed in a blinded manner.

(i) Stitching of four 16-bit raw image stacks to capture the entire end phalanx in one 'superstack'. Four individual image stacks, each consisting of 500 frames at 1,050 pixels × 1,050 pixels were stitched together resulting in a superstack of 500 frames at 2,000 pixels × 2,000 pixels. Our workflow first generates maximum intensity projections of neighbouring stacks and then applies the feature detectors SIFT[62] or SURF[63] on the 10% overlap regions in the two MIP images. The stitching algorithm reduces and cleans the noisy influence of the original image boundary to obtain a seamlessly stitched superstack.

(ii) Removal of epidermal autofluorescence to facilitate subsequent steps of automated analysis. Epidermal autofluorescence and skin appendages were removed using a weakly supervised regularization approach developed on discrete graph spaces for perceptual image segmentation through a semi-supervised learning algorithm. Each image in a 3D image stack is segmented according to the parameters and label information derived from 3D space. Gaps in the epidermal signal and hair can be accurately classified and removed. A spectral clustering method is embedded and extended into regularization on discrete graph spaces. In consequence, the spectral graph clustering is optimized and smoothed by integrating top-down and bottom-up processes through semi-supervised learning. Then, a nonlinear diffusion filter is used to maintain semi-supervised learning, labelling and differences between foreground or background regions. Furthermore, the segmentation is penalized and adjusted using labelling prior and optimal window-based affinity functions in a regularization framework on discrete graph spaces. The algorithm

is robust in handling images from variable environments.

(iii) Sixteen-bit to eight-bit conversion of superstacks by nonlinear adaptive depth-dependent adjustment on global and local scales. To reduce the size of the superstacks for subsequent processing steps, we devised an approach that converts 16-bit to 8-bit image stacks while optimally maintaining the dynamic range of the signals. First, a global-to-local nonlinear contrast enhancement method was used to improve the global contrast on the basis of the range of intensity and the histogram of each image in a stack. To reflect the increase in laser power with imaging depth, the global-to-local contrast enhancement adjusts the histogram between the top and the bottom layers of the image stack. Second, a depth-adaptive intensity-based image enhancement method was developed for obtaining the final 8-bit enhanced image stacks. The MATLAB function im2uint8 was used for conversion of image stacks according to the global-to-local adaptive histogram in each image. Finally, to consider image stacks acquired at different time points of our longitudinal experimental design, the conversion takes advantage of normalized criteria including normalized human visual perceptual contrast.

(iv) Automatic rigid and non-rigid four-dimensional registration of stacks acquired at different time points. Superstacks acquired during a time frame of up to one year were aligned using a weakly supervised automated registration algorithm optimized for large-scale datasets. To find the largest similarity (for example, intensity localization and patterns) between two superstacks, we directly extracted representative feature points from the source and target superstacks in 3D space. To achieve this, we extracted 3D point clouds of nerve fibres from original superstacks and estimated the 3D registration parameters from the extracted 3D point clouds on the basis of the iterative closest point algorithm described by the iterative closest point (ICP) function. The 3D registration method seeks to find the best transformation $T$ that relates two entities $P$ and $Q$ whose 3D point clouds are given by $R_P$ and $R_Q$, respectively. $T_P$ was found such that the objective function $J(R_P, R_Q)$ is minimized, $J(R_P, R_Q) = \sum_{P \in R_P} \|T_P - \Psi(P)\|$, where $\Psi : P \to Q$; for $\forall P \in R_P$, $\Psi(P) \in R_Q$. The transformation $T_P$ is used to optimally align two point clouds. The function $\Psi(P)$ is usually unknown and needs to be computed. When a good initial value is given, the algorithm can achieve global convergence. First, the automatic registration system has been implemented in a stratified 3D model registration framework, which efficiently handles hierarchical pyramid multi-resolution 3D image stacks. The stratified methods for auto-registration involve 3D-to-3D pose correction, 3D-to-3D projection and 3D-to-3D linear and nonlinear registration. Furthermore, we applied the thin-plat-spline (TPS) method[64] for global registration by jointly warping local feature points onto their global position. By using this technique, the system not only achieves accurate 3D pose normalization, but also becomes reliable and avoids difficulties in the linear transformation of 3D image stack pairs. The approach is efficient and robust especially for large-scale 3D data alignment and even alignment of superstacks with partial similarity (for example, after many nerve fibres got lost after SNI).

(v) Automated neuronal segmentation, tracing and statistical neural network analysis. Quantitative measurements of fibre length of the same phalanx at different time points were made, including the length of total fibres, number of fibre endings, distribution and density of fibre endings. To improve the tracing fidelity and to measure efficiency of structural changes from all nerve fibres in the end phalanx, we developed a 3D tracing algorithm adapted for computing large-scale 3D datasets in a weighted global to local optimization manner. Nerve fibres with discontinuities caused by fluctuations of signal strength, epidermal fibres and neighbouring cells can be identified independently of the autofluorescence background. To determine morphological changes as well as the relationship between structural plasticity of nerves and pain levels over time, we developed algorithms for not only measuring the change of fibres quantitatively, but also localizing these changes in fibre networks so that data-driven and target-oriented structural plasticity analysis can be achieved. The method includes the following steps: (a) cropping of all superstacks acquired from a mouse at different time steps to the largest common volume to ascertain quantitative comparisons; (b) graph-based analysis of connectivity and fibres (see Supplementary Note 2 for details); (c) accurate 3D registration of nerve fibre changes to localize the changes of fine structures and branching changes; (d) localization and measurement of dynamic changes in the fine structure of filopodia and small branches; (e) identification of dynamic changes in fibre connections (clusters, undirected graph to directed graph); and (f) plasticity analysis of the fibre network.

(vi) Quantitative analysis of structural plasticity in SNS-mGFP and Thy1-YFP mice. In this step, we measured different parameters for thick fibres (Aβ-fibres) and thin fibres (Aβ-fibres, C-fibres) as well as fibre endings, Meissner corpuscles and epidermal fibres. In SNS-mGFP mice, we studied intra-epidermal fibres with respect to distribution, density and fine structures (for example, filopodia or small branches) and we analysed deeper fibres forming a network in the dermis. In Thy1-YFP mice, the unstable thin fibres and relatively stable thick fibres were classified into two groups and analysed in regions of interest. Fibre sprouting and regeneration were measured and localized quantitatively in space at different time points. To achieve this, we designed and implemented different analysis approaches specialized for the characteristics of two different types of data. See Supplementary Note 2 for further details.

## Immunohistochemistry

Mice were transcardially perfused with 4% PFA (Sigma) and DRGs and hind paw skin were dissected out. Tissue samples were cryo-protected with 30% sucrose overnight before cryo-sectioning. The sections were treated with 50 mM glycine in 0.05 M PBS for 15 min followed by permeabilization with 0.2% Triton-X-100 for 15 min. After blocking for 30 min with 10% normal horse serum in 0.1% PBST, sections were incubated with primary antibody such as rabbit anti-beta-tubulin III (T2200, Sigma; 1:500), anti-NF200 (CH23015, Neuromics; 1:200), anti-CGRP (24112, Immunostar; 1:200), anti-SP (GP14103 Neuromics; 1:200), anti-S100 (Z0311, Dako; 1:200), anti-TH (SO25000, Neuromics; 1:200), anti Gr-1(Mouse Ly-6G/ly-6C, MAB 1037, R&D Systems; 1.500), anti-CD8a (14-0808-82, Thermo Fisher Scientific; 1:200), anti-CD4 (14-9766-82, Thermo Fisher Scientific; 1:100) and biotinylated IB4 (1:200; B-1205, Vector) in blocking solution, overnight at 4 °C. Next day, the sections were washed once with blocking solution and twice with 0.2% PBST for 15 min. Sections were then incubated for 1 h at room temperature with the following corresponding Alexa-conjugated secondary antibodies (1:750; Thermo Fisher Scientific): donkey anti-Rabbit IgG, Alexa 488 conjugated (A32790); donkey anti-rabbit IgG, Alexa 594 conjugated (A32754); donkey anti-rabbit IgG, Alexa 647 conjugated (A32787); donkey anti-rat IgG, Alexa 647 conjugated (A48272); donkey anti-rat IgG, Alexa 594 conjugated (A48271); donkey anti-rat IgG, Alexa 488 conjugated (A48269); goat anti-guinea pig IgG, Alexa 647 conjugated (A-21450). After 3 washes for 15 min each in 0.2% PBST, sections were treated with 10 mM Tris pH 8.0 for 15 min and mounted with Mowiol and stored in the dark at 4 °C. The sections were then imaged with a confocal microscope using Leica Application Suite X (LAS X).

## Skin-nerve recordings

Mice were killed by placing them in a $CO_2$-filled chamber for 2–4 min followed by cervical dislocation. The glabrous and hairy hind paw skin were dissected free in one piece together with the sural nerve

or the tibial nerve, respectively, and placed in a heated (32 °C) organ bath chamber that was perfused with synthetic interstitial fluid (SIF buffer) consisting of 108 mM NaCl, 3.5 mM KCl, 0.7 mM MgSO$_4$, 26 mM NaHCO$_3$, 1.7 mM Na H$_2$PO$_4$, 1.5 mM CaCl$_2$, 9.5 mM sodium gluconate, 5.5 mM glucose and 7.5 mM sucrose at a pH of 7.4. As hairy and glabrous skin were dissected as one piece, only the skin covering the palm and the back of the paw, but not the skin covering the digits, could be fully preserved during the dissection procedure. The skin was placed with the corium side up in the organ bath and the nerve was placed in an adjacent chamber for fibre teasing and single-unit recording. Single units were isolated with a mechanical search stimulus applied with a glass rod and classified by conduction velocity, von Frey hair thresholds and adaptation properties to suprathreshold stimuli as previously described[53]. Mechanical ramp-and-hold stimuli were applied with a cylindrical metal rod (diameter 1 mm) that was driven by a nanomotor (MM2A-LS, Kleindiek Nanotechnik) that was coupled to a force measurement system (FMS-LS, Kleindiek Nanotechnik). The von Frey thresholds of single units were determined by mechanically stimulating the most sensitive spot of the receptive fields with von Frey filaments (Ugo Basile) and the force exerted by the weakest von Frey filament that was sufficient to evoke an action potential was considered as the von Frey threshold. The raw electrophysiological data were amplified with an AC coupled differential amplifier (Neurolog NL104 AC), filtered with a notch filter (Neurolog NL125-6), converted into a digital signal with a PowerLab 4SP (ADInstruments) and recorded at a sampling frequency of 20 kHz using LabChart 7.1 (ADInstruments).

### DRG cell labelling and RNA sequencing
Mice were injected in the specified nerve territory of digits either with 2 µg of cholera toxin B (CTB) conjugated with Alexa 488/594 (C34775/ C34777, Thermo Fisher Scientific) or 40 µg of fast blue (17740-1, Polysciences). L3–L4 DRGs were collected 48 h later and cells were dissociated with collagenase IV (1 mg ml$^{-1}$, Sigma-Aldrich, C5138) and trypsin (0.5 mg ml$^{-1}$, Sigma-Aldrich, T1005) for 30 min each at 37 °C, washed, placed on laminin-coated slides and counterstained with Alexa Fluor 568 conjugate (2.5 µg ml$^{-1}$, Isolectin GS-IB from *Griffonia simplicifolia*, Alexa Fluor 568 conjugate, Invitrogen, I21412) for 10–15 min at room temperature. IB4-positive and IB4-negative cells were identified and manually collected using a fire polished pipette, immediately shock frozen in liquid nitrogen and expelled into PBS with RNAse inhibitor (Takara 2313A). Cell lysates were directly processed to reverse transcription using the previously published SmartSeq2 protocol[65]. Libraries were prepared on the basis of the tagmentation protocol described previously[66]. The cDNA was generated using 18 pre-amplification cycles. Libraries were sequenced with an Illumina NextSeq 500.

Sequencing reads were mapped to the GRCm38 mouse reference genome using STAR[67] (v.2.7.1a) using default parameters and extracting also gene counts (quantMode GeneCounts) based on GRCm38.101 gene annotation. Differential gene expression analysis was performed using DESeq2[68] (v.1.28.1) and only genes having a false discovery rate (FDR) lower than 10% were considered as significant. Functional enrichment analysis on the significant differentially expressed genes was assessed using the MGSA R package[69] (v.1.36.0). We used the R package BisqueRNA[70] to decompose our bulk expression data based on a reference single-cell sequencing dataset[48].

### Three-dimensional electron microscopy imaging
Mice were transcardially perfused using fixative containing 2.5% paraformaldehyde (Sigma), 1.25% glutaraldehyde (Serva) and 2 mM calcium chloride (Sigma) in 80 mM cacodylate buffer adjusted to pH 7.4 with an osmolarity of 700–800 mOsmol l$^{-1}$. Skin (digit) tissue from the corresponding nerve territory was dissected and postfixed at 4 °C overnight. Samples were then stained using the Hua protocol[71]. Sample blocks were infiltrated in Spurr's resin. Three-dimensional

electron microscopy imaging was acquired using serial block-face electron microscopy[72]. Serial sectioning was done using a custom-built microtome that was operated by custom-written software[73], placed in a scanning electron microscope (FEI, Thermo Fisher Scientific). Electron microscopy images were acquired at a resolution of 11.24 nm × 11.24 nm × 30 nm. The stacks of electron microscopy images were aligned using custom-written software described previously[73,74] and aligned image stacks were then uploaded in webKnossos[75] for further analysis and visualization.

### Intraganglionic injections
Adeno-associated virions were injected into the DRGs as described previously[76,77]. In brief, AAV-EF1a-flexed-taCasp3-TEVp or AAV-GFP (approx. $1 \times 10^{13}$–$4 \times 10^{13}$ viral genomes per ml) was mixed with 0.1% fast green to assess the injection efficiency. This mixture was injected unilaterally into L3 and L4 DRG neurons using a glass pipette with a diameter of approximately 25 µm. Mice were allowed to recover for three weeks before behavioural assessment.

### Statistical analyses
All data were calculated and are presented as mean ± s.e.m. A one-way or two-way ANOVA for repeated measures followed by Bonferroni's post-hoc test or Tukey's test was used to determine statistically significant differences for multiple group comparisons. For comparisons involving two groups, a two-tailed *t*-test was used. Chi square analysis was used in electrophysiological analyses of proportion of responding C-fibres. Changes with $P < 0.05$ were considered to be significant. Sigma plot, Microsoft Excel and GraphPad Prism software were used for statistical analyses.

### Reporting summary
Further information on research design is available in the Nature Research Reporting Summary linked to this paper.

## Data availability
All of the raw data for behavioural, electrophysiological and immunohistochemical analyses are provided in the source data files and in the figures. RNA sequencing data are available through the European Nucleotide Archive (https://www.ebi.ac.uk/ena) under the accession number PRJEB50184. The raw data for multiphoton imaging and electron microscopy analyses will be made available upon request. Source data are available for this paper.

## Code availability
Electron microscopy acquisition codes are available at Gitlab (https://gitlab.mpcdf.mpg.de/connectomics/emacquisitionmacro.git) and the codes used for analysis of multiphoton imaging data are available at GitHub (https://github.com/zheng-tklab/pns_2photon_longitudinal-3Ddata-analysis).

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

**Acknowledgements** We thank S. Arber, A. Basbaum and Y. De Koninck for feedback on the study and the manuscript; S. Arber for the gift of the Tau-loxP-mGFP mice; N. Gehrig, H.-J. Wrede and D. Baumgartl-Ahlert for technical assistance; H. Wissler for his support with visualization of 3D electron microscopy data; and M. Kurejova and M. Seefeld for help with the early stages of setting up multiphoton imaging in mouse paw. The research leading to these results has received funding from the following sources: an ERC Advanced Investigator grant to R.K. (Pain Plasticity 294293); grants from the Deutsche Forschungsgemeinschaft to R.K. (SFB1158, projects B01, B06), to T.K. (SFB1158, project B08), to S.G.L. (SFB1158, project A01) and to V.G. (SFB1158, project A03); a grant to B.O. (project number 371923335); and grant CIDEGENT/2020/052 from Generalitat Valenciana to F.J.T. R.K. is a member of the Molecular Medicine Partnership Unit of the European Molecular Biology Laboratory and Medical Faculty Heidelberg. V.G. and T.A.N. were partially supported by a post-doctoral fellowship and physician scientist fellowship, respectively, from the Medical Faculty Heidelberg. D.M. was partially supported by a post-doctoral fellowship from Excellence Cluster CellNetworks. We acknowledge support from the Interdisciplinary Neurobehavioral Core (INBC) for the behavioural experiments, the data storage service SDS@hd and bwMLS&WISO HPC supported by the state of Baden-Württemberg and the German Research Foundation (DFG) through grants INST 35/1314-1 FUGG and INST 35/1134-1 FUGG, respectively.

**Author contributions** V.G., D.M., T.A.N. and N.A. performed all wet experiments under the supervision of R.K., apart from the establishment of longitudinal multiphoton imaging on mouse paw, which was performed by V.G. under the supervision of T.K., the electrophysiology experiments, which were performed by F.J.T. and S.G.L., and the RNA sequencing analyses, which were performed by J.P. under the supervision of V.B. H.Z. developed mathematical algorithms for image analysis under the supervision of B.O. and T.K. J.L. performed bioinformatics analyses on RNA sequencing. V.G. performed electron microscopy analyses under the supervision of M.H., with conceptual input from T.K. V.G. and R.K. conceptualized the project and T.K. and B.O. provided regular conceptual input. V.G. and H.Z. prepared the figures and videos. R.K. wrote the manuscript and all of the authors provided conceptual input in writing the manuscript and presenting the data. All of the authors read the manuscript in the submitted form.

**Competing interests** The authors declare no competing interests.

**Additional information**
**Correspondence and requests for materials** should be addressed to Rohini Kuner.

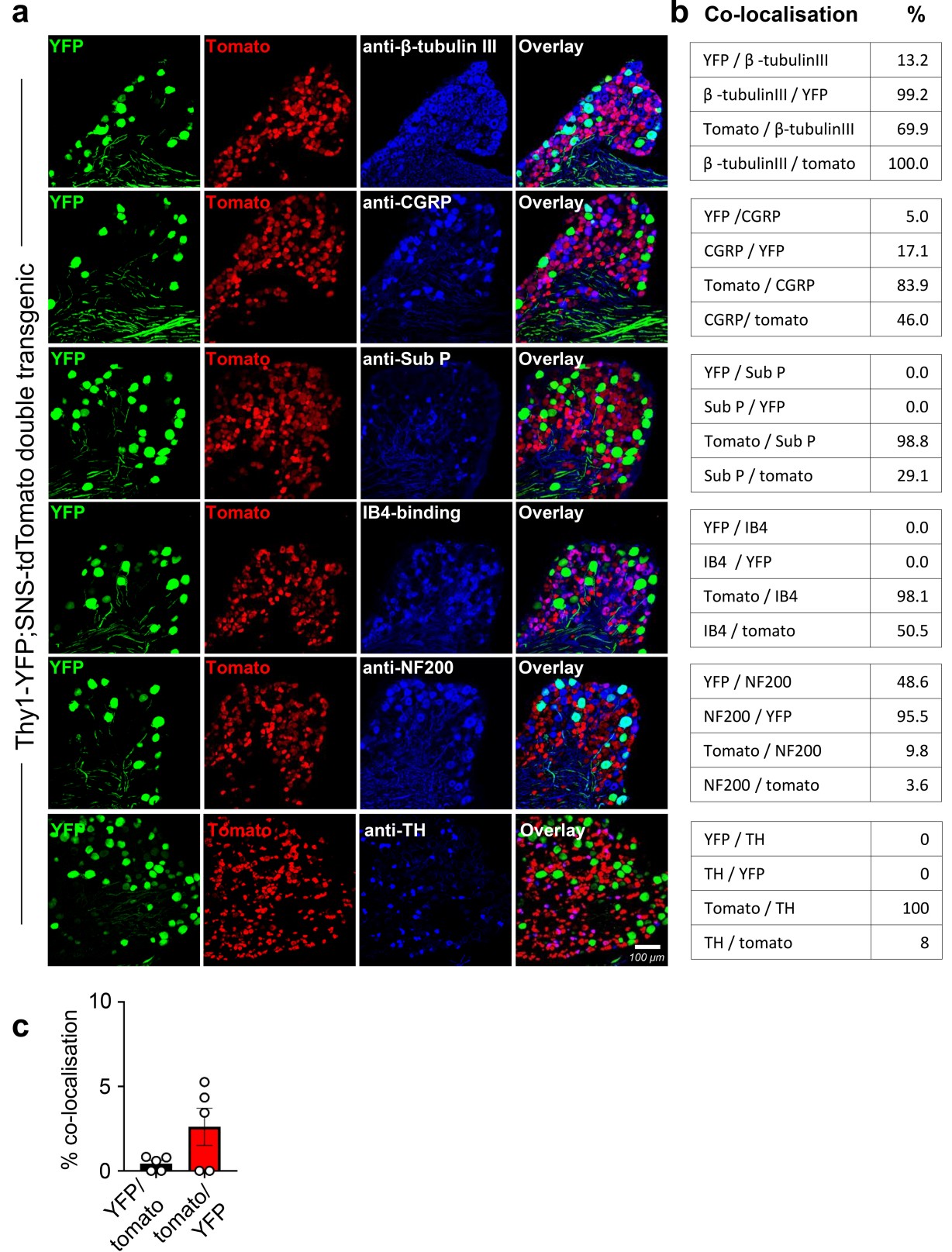

**Extended Data Fig. 1 | Comprehensive characterization of the neurochemical identity of fluorescently marked cells in the DRGs of Thy1-YFP mice and SNS-tdTomato mice.** (**a**) Confocal images of YFP and Tomato expression are shown with immunohistochemical staining of diverse sensory neuron populations. Scale bar = 100 μm. (**b**) Quantitative degree of co-localization with markers of distinct classes of DRG neurons (n = 5 mice/group). (**c**) Quantification of lack of colocalization of Thy1-labelled and tdTomato-labelled neuronal population mice in double-transgenic mice. (n = 5 mice/group). Data shown as mean ± S.E.M.

**b** Co-localisation / %

| Co-localisation | % |
|---|---|
| YFP / β-tubulinIII | 13.2 |
| β-tubulinIII / YFP | 99.2 |
| Tomato / β-tubulinIII | 69.9 |
| β-tubulinIII / tomato | 100.0 |
| YFP /CGRP | 5.0 |
| CGRP / YFP | 17.1 |
| Tomato / CGRP | 83.9 |
| CGRP/ tomato | 46.0 |
| YFP / Sub P | 0.0 |
| Sub P / YFP | 0.0 |
| Tomato / Sub P | 98.8 |
| Sub P / tomato | 29.1 |
| YFP / IB4 | 0.0 |
| IB4 / YFP | 0.0 |
| Tomato / IB4 | 98.1 |
| IB4 / tomato | 50.5 |
| YFP / NF200 | 48.6 |
| NF200 / YFP | 95.5 |
| Tomato / NF200 | 9.8 |
| NF200 / tomato | 3.6 |
| YFP / TH | 0 |
| TH / YFP | 0 |
| Tomato / TH | 100 |
| TH / tomato | 8 |

**a** Sural territory (digit)

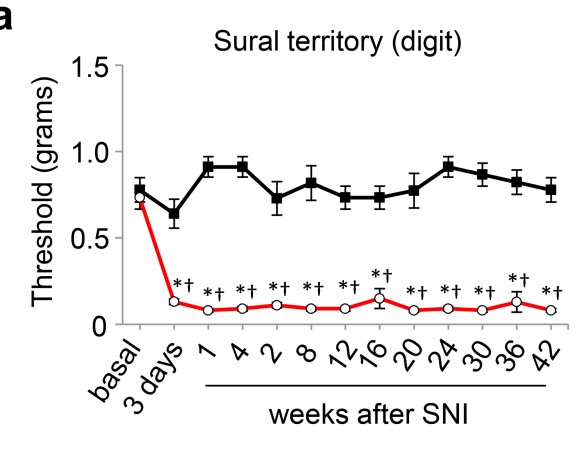

**b** Sural territory (paw central surface)

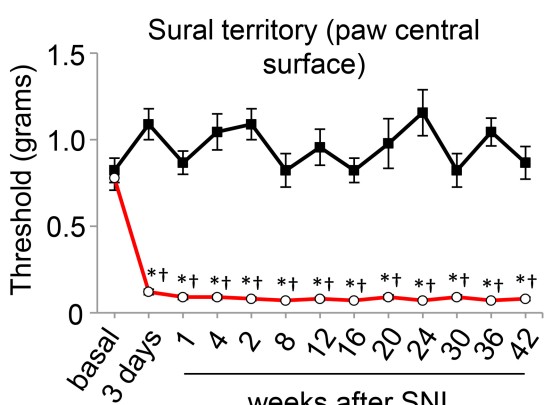

Tibial territory (paw central surface)

**c** Place escape/avoidance paradigm (PEAP)

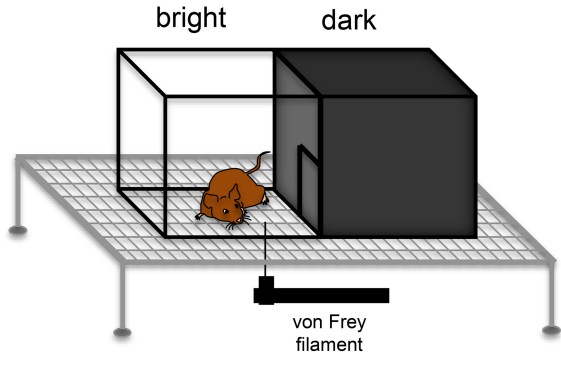

von Frey filament

**d**

sham

SNI

time (min)

**e** Acetone drop test

Sural digit    Tibial digit

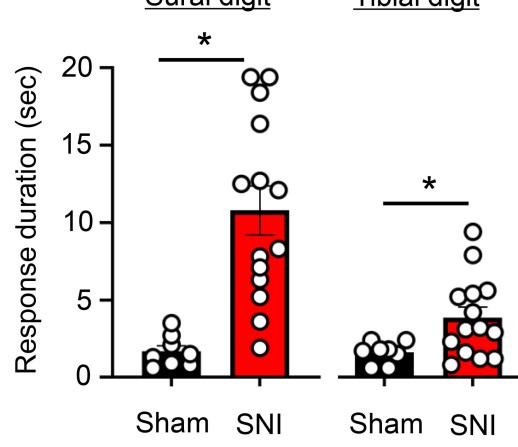

—■— Sham
—○— SNI

**Extended Data Fig. 2** | See next page for caption.

**Extended Data Fig. 2 | Development of late-onset, chronic neuropathic pain after a period of complete loss of sensitivity in denervated skin of mice with nerve injury, referred to henceforth as reinnervation-induced neuropathic pain.** (a, b) In the SNI model, intense mechanical allodynia develops in the territory of the undamaged sural nerve, as previously described, as evidenced by drop in response threshold to application of graded mechanical force via von Frey hairs to the plantar surface of the hind paw. Shown are data from the digit (**a**) and central part of the paw plantar surface (upper panel in **b**) in the sural territory. In **a** N = 9 per group: $F_{(1,12)}$ = 329.423, P = 4.31E-10. As expected, mice initially show a complete loss of sensitivity in the territory of the ligated and severed tibial nerve, recover sensitivity around 12 weeks post-SNI and unexpectedly show marked reduction in mechanical response threshold as of 16 weeks post-SNI. In **b** (lower panel), data from the central paw surface in the tibial territory are shown, complementary to data from the corresponding digit shown in main Fig. 1e. N = 9 per group; sural territory: $F_{(1,12)}$ = 284.970, P = 9.99E-10; tibial territory after 20 weeks: $F_{(1,4)}$ = 86.057, P = 0.00075; In **a**, **b**,* represents $p < 0.05$ compared to baseline, †$p < 0.05$ compared to control group (sham), two-way repeated measures ANOVA with Bonferroni multiple comparison. Data shown as mean ± S.E.M. (**c**, **d**) Passive escape avoidance behaviour after application of 0.16 g von Frey force to the tibial innervation territory (middle digit) either ipsilateral to the nerve injury in the dark chamber at 24 weeks post sham- or SNI surgery. Shown are schematic overview (**c**), one typical example each from SNI and sham mice (**d**), complementary to main Fig. 1f. (**e**) Increased duration of nocifensive responses to acetone, demonstrating significant cold allodynia in undamaged sural territory and the denervated tibial territory after collateral sprouting at 24-26 weeks post-SNI (N = 8 for sham and 14 for SNI; sural territory: t = 4.27, df = 20, F = 33.91, P = 0.00037; tibial territory: t = 2.392, df = 20, F = 13.91, P = 0.0266; *$p < 0.05$; two-tailed unpaired t test).

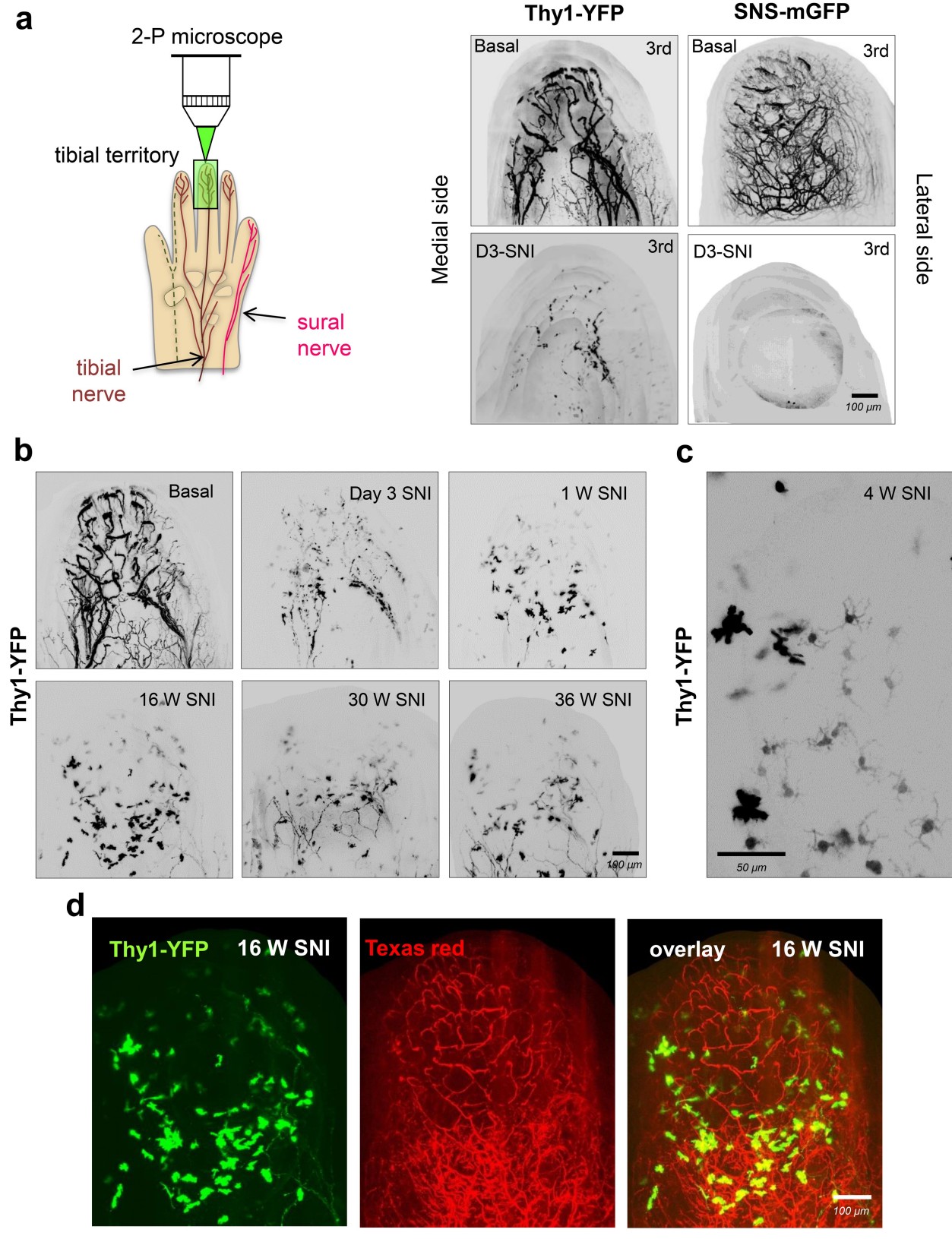

**Extended Data Fig. 3 | Alterations in tibial innervation territory (middle digit) after loss of tibial and common peroneal nerves through SNI.**
(**a**) Schematic representation of multiphoton non-invasive imaging in tibial digit (3ʳᵈ digit) (right panel). At day 3 after SNI, both large diameter sensory fibres (YFP-positive) and nociceptive fibres (mGFP-positive) are fully lost in the respect transgenic mouse lines (left panel) (N = 4). (**b**) After degeneration of YFP-positive fibres in Thy1-YFP mice, YFP fluorescence is seen in cell-like structures; by 30-36 weeks, there is faint re-emergence of YFP fluorescence in a few medium-diameter nerves although the large fibres ending in Meissner corpuscles are still missing. (N = 4) (**c**) High magnification view of YFP-expressing cells, which show a stellate morphology resembling dendritic-type immune cells (N = 2); see Supplementary Note 1 for details. (d) YFP-positive cells found in the vicinity of blood vessels labelled via Texas-red-conjugated Dextran (N = 2). Scale bar = 100 µm in a, b, d and 50 µm in c.

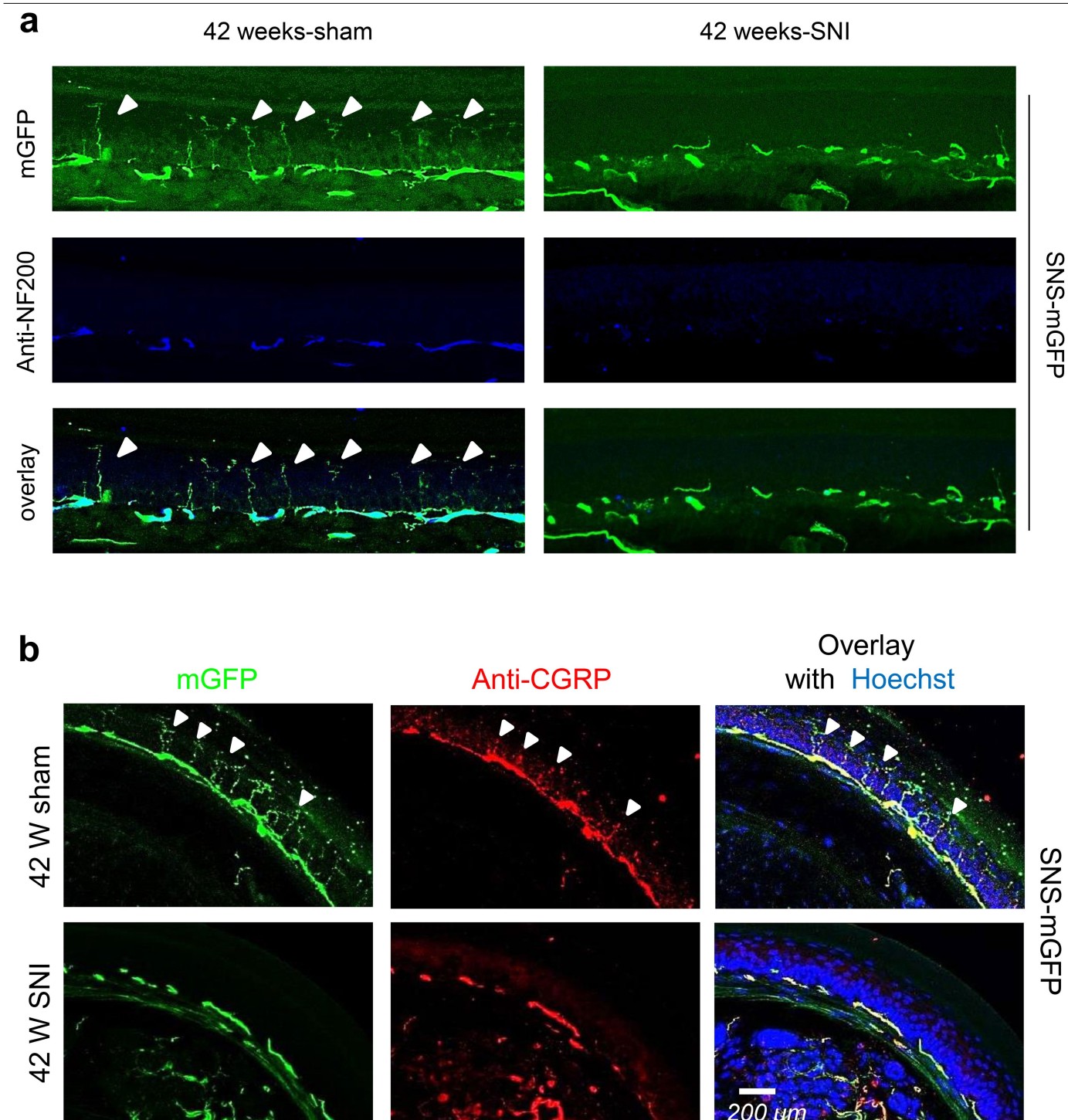

**Extended Data Fig. 4** | See next page for caption.

**Extended Data Fig. 4 | Further demonstration of the absence of reinnervation with Aβ-LTMR and recovery of nociceptor innervation with the emergence of reinnervation-induced neuropathic pain in the tibial territory through co-immunostaining with endogenous marker proteins.** (**a**, **b**) Example confocal images represent the skin of the area at the centre of the hind paw in panel **a** and the digit in the tibial nerve territory in panel **b**. Both panels show nociceptors expressing mGFP in SNS-mGFP mice that repopulate the tibial innervation territory at the at 42 weeks post-SNI or sham surgery. Large diameter (Aβ) fibres are identified via immunostaining for Neurofilament 200 (NF200) in panel **a** and peptidergic nociceptors are identified via immunostaining for the peptidergic nociceptor marker, calcitonin gene related peptide (CGRP) in panel **b**. Hoechst dye (blue) was used to counterstain nuclei in panel **b**. In all panels, arrowheads represent intra-epidermal nerve fibres. Taken together, both panels show that collaterally sprouting nociceptors arrive at dermal-epidermal border in SNI mice, despite a loss of NF200-immunoreactive Aβ fibres; however, they do not enter the epidermis (arrowheads). These images complement with endogenous markers of sensory afferents the data shown with transgenic fluorescent markers in main Fig.1d and Fig. 2e, f.

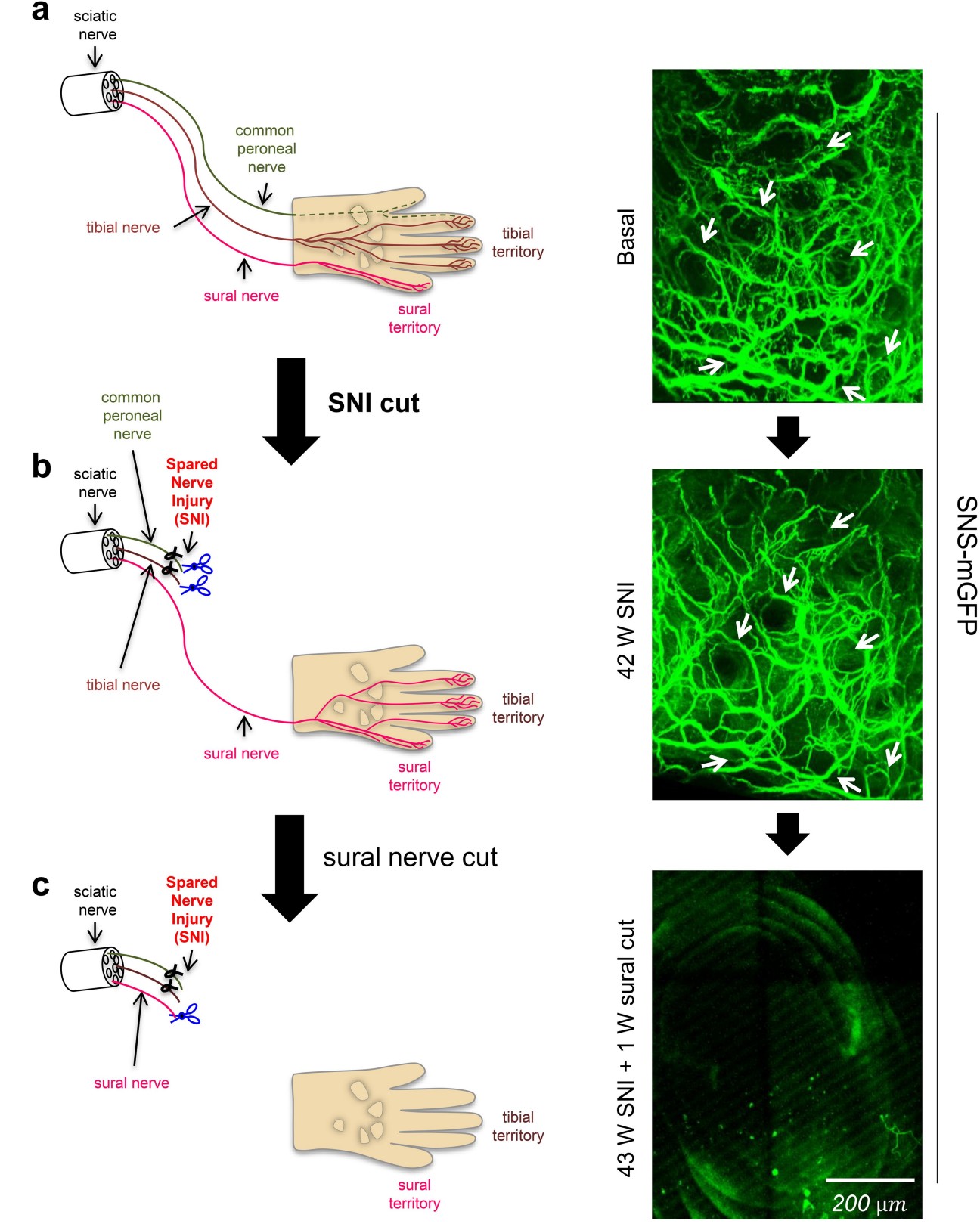

**Extended Data Fig. 5 | Analysis establishing the sural origin of GFP-expressing fibres that populate the denervated tibial territory after SNI.** Experimental steps are schematically shown on the left and the corresponding multiphoton analyses are shown on the right. (**a**) Innervation pattern of mGFP-expressing nociceptive nerves in the middle digit (tibial territory) in SNS-mGFP prior to SNI. (**b**) Sprouted mGFP-positive fibres that repopulate the denervated territory when analysed at 42 weeks post-SNI. (**c**) Complete loss of GFP-expressing fibres in the same mouse 1 week after transection of the sural nerve at 42 W post-SNI, demonstrating the sural origin of the collaterally sprouted nociceptive fibres in the denervated tibial territory (N = 2). Scale bar = 200 μm.

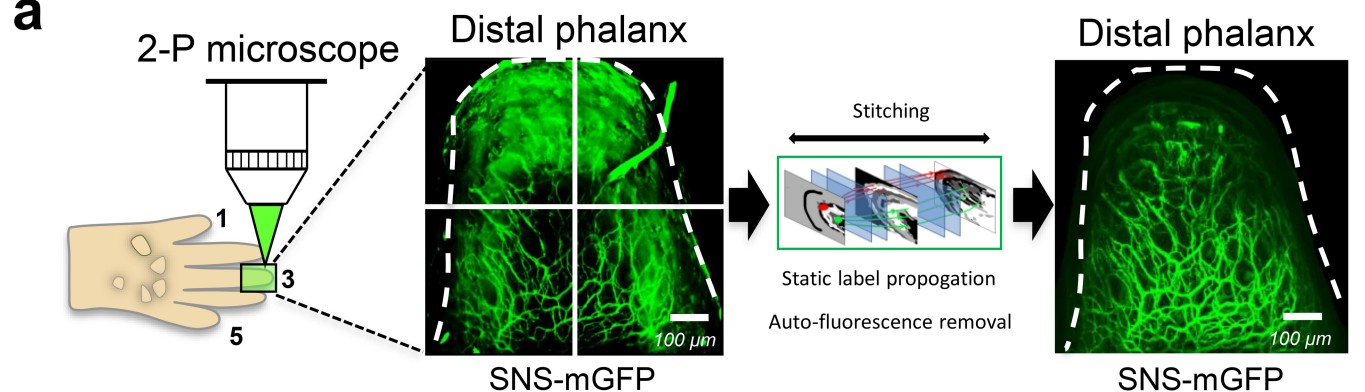

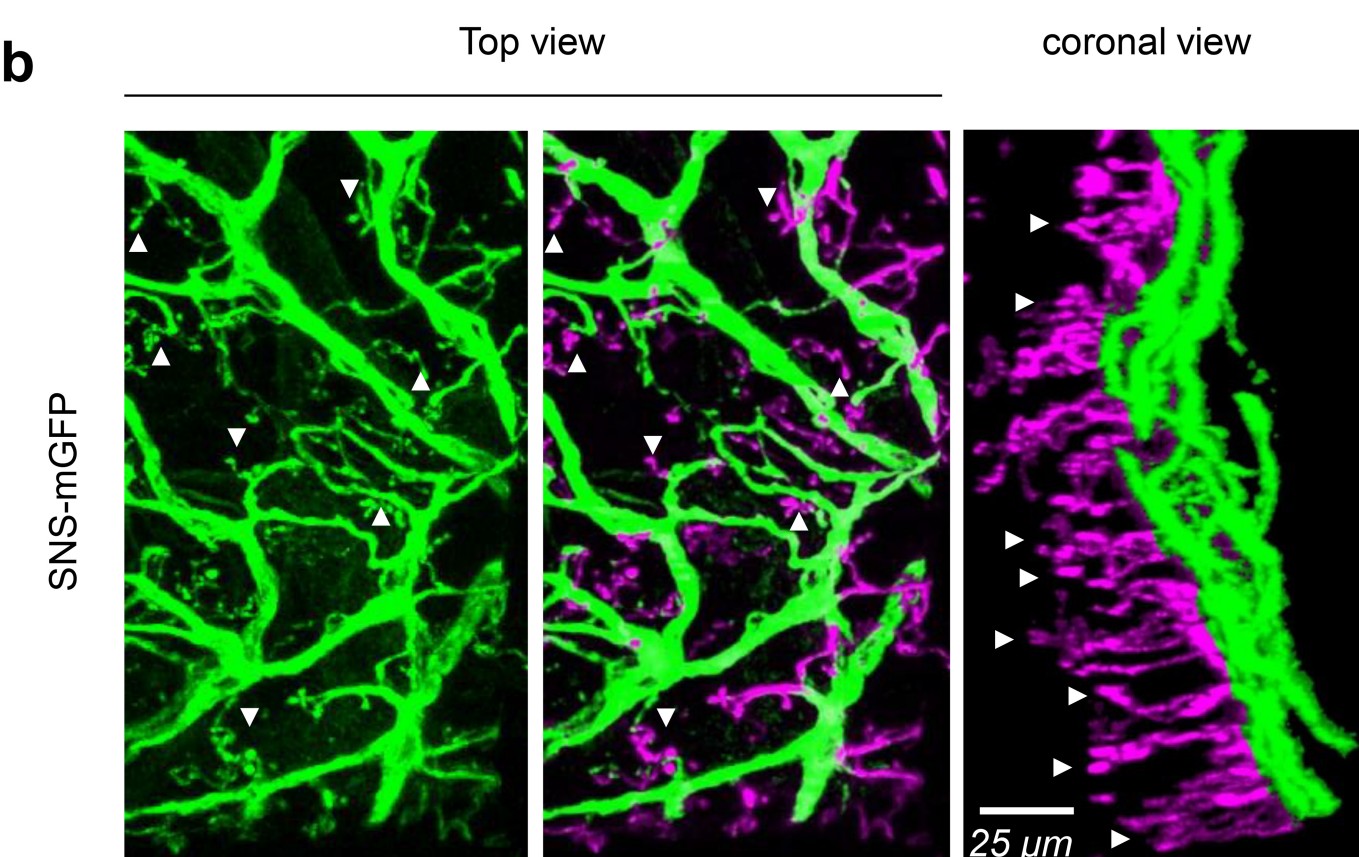

**Extended Data Fig. 6 | Workflow for quantitative analyses of sensory fibres in in vivo multiphoton imaging datasets. (a)** Workflow used for extraction of signals over background in two photon imaging of individual digits in mouse hind paw. Examples show skin afferents of mice with SNS-Cre-derived expression of membrane GFP expression (mGFP) in small-diameter nociceptors. **(b)** Segmentation of free endings (shown in false colour purple, arrowheads) of mGFP-labelled nociceptors and their delineation from afferent branches (shown in green) in top view and transverse view, which enabled 3D quantitative analyses (N = 4). This figure is complementary to main Fig. 2c, d.

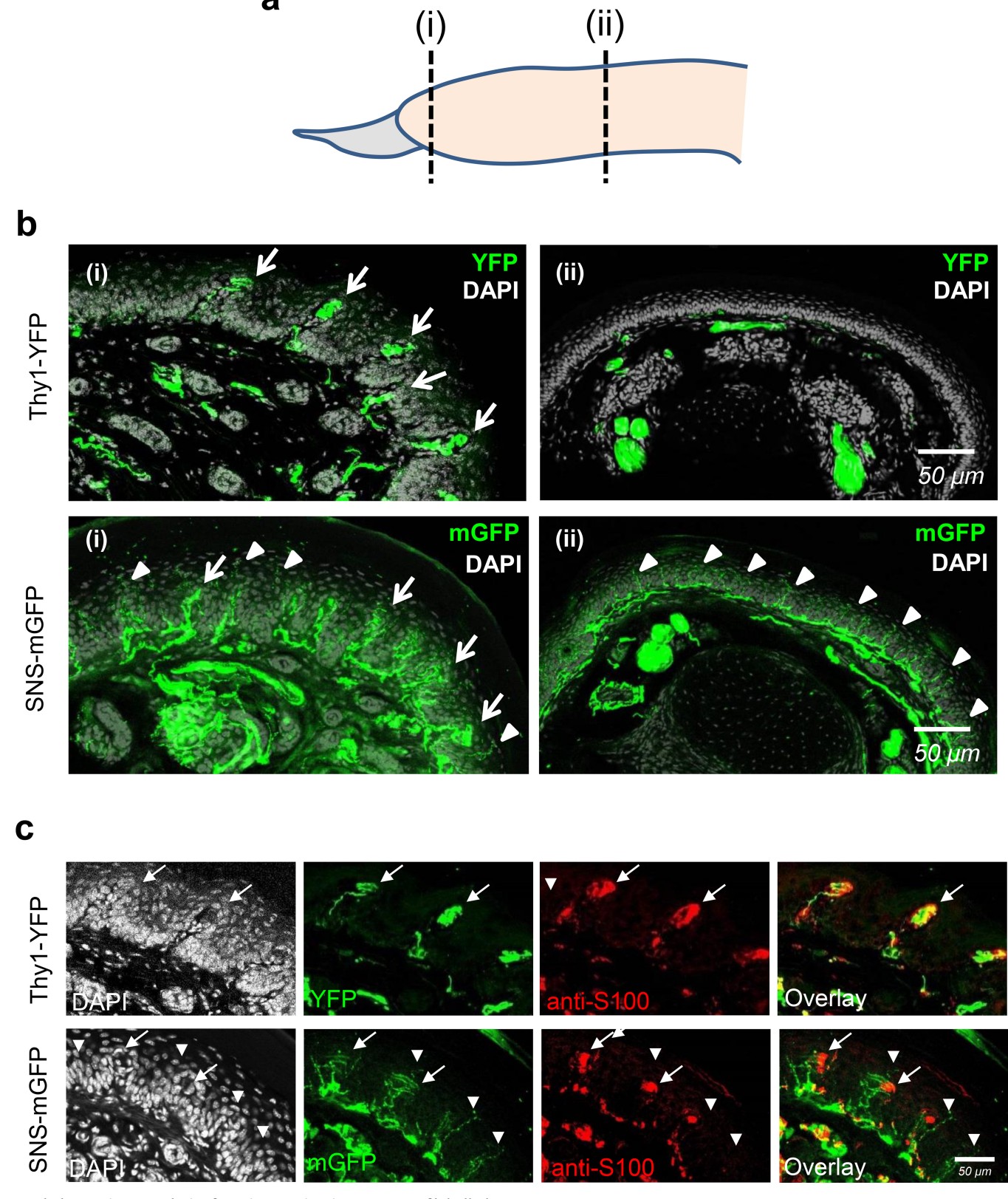

**Extended Data Fig. 7 | Analysis of precise termination patterns of labelled nociceptors and Aβ-LTMRs in the hind paw skin of SNS-mGFP and Thy1-YFP mice, respectively.** (**a**) Schematic representation of parts of the paw digit where sensory fibre terminations were studied. (**b**) Upper panels show labelling of Aβ-LTMRs in Thy1-YFP mice. Shown are DAPI-stained images depicting the dermal invaginations into the epidermis (arrows, left panels), harbouring YFP-expressing large diameter fibres ending in Meissner corpuscles (arrowheads, left panels) in distal regions (i) and YFP-expressing fibres more proximally (ii) in the digits (right panels). Lower panels show terminations of labelled nociceptors in SNS-mGFP mice in form of mesh-like structures surrounding Meissner corpuscles at dermal invaginations in DAPI-stained images distally (i) and epidermal free nerve endings (arrowheads) distally (i) and proximally (ii). (**c**) Anti-S100 immunohistochemistry to identify Meissner corpuscles (arrows) in paw sections of Thy1-YFP and SNS-mGFP mice (N = 4). Arrowheads indicate free nerve endings. Panel c is a more complete representation of main Fig. 3b.

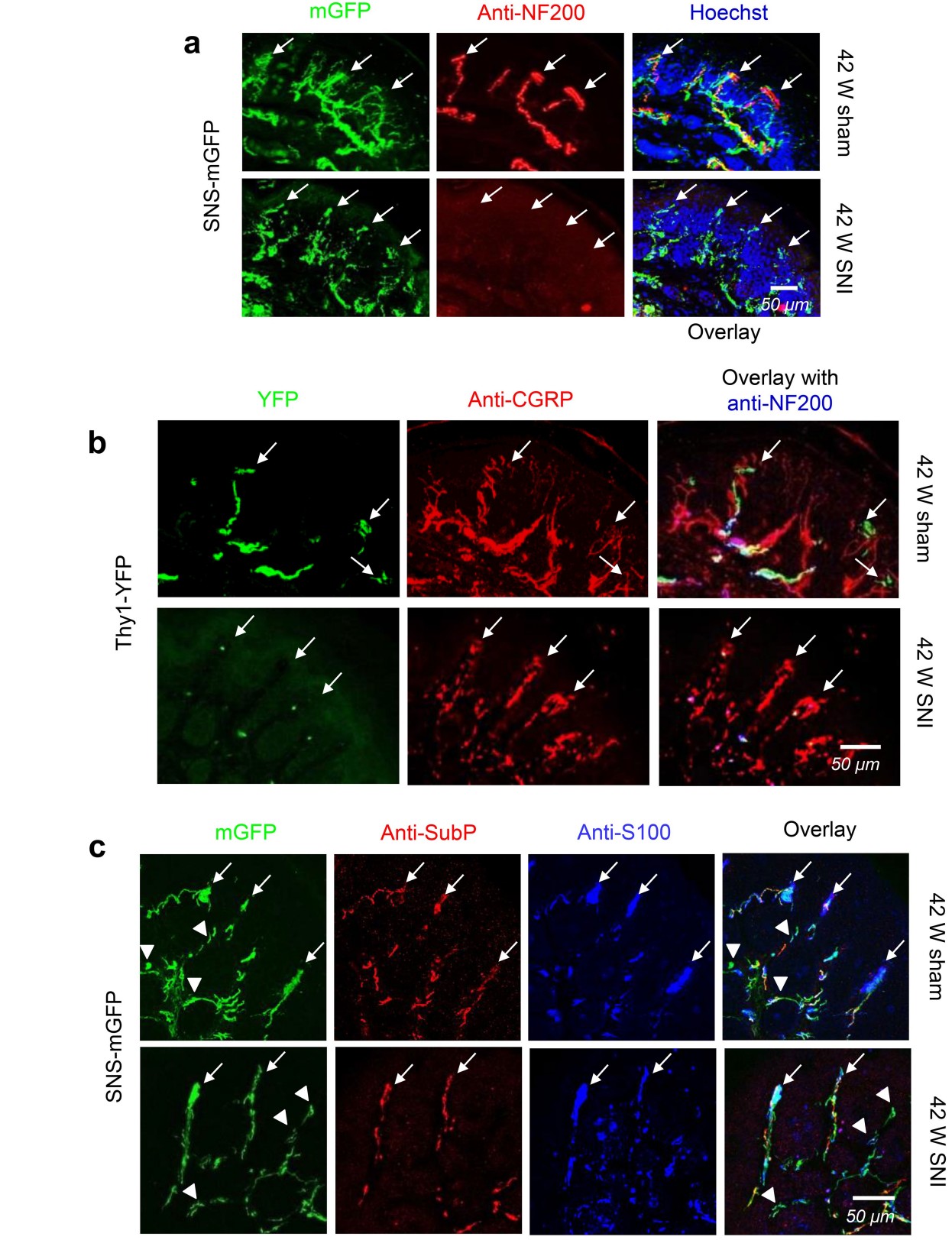

**Extended Data Fig. 8** | See next page for caption.

**Extended Data Fig. 8 | Further demonstration of loss of Aβ-LTMR terminations and recovery of nociceptor terminations at Meissner corpuscle zones in the tibial territory of mice with SNI through co-immunostaining with endogenous marker proteins.** (a) Confocal analyses showing recovery of collaterally sprouted nociceptor terminations (SNS-mGFP) at Meissner zones, whereas Aβ-LTMR innervation is lost as seen with native marker protein NF200 in the tibial territory post-SNI. These images correspond to the overlay image shown in main Fig. 3c. (**b**) Both YFP fluorescence as well as NF200-expressing terminations are lost, whereas collaterally sprouted nociceptors expressing the peptide CGRP are seen in the Meissner zone at the dermis–epidermis border in the tibial territory 42 weeks post-SNI. These examples extend the data shown in main Fig. 3b, e (N = 4). (**c**) Complementary to the quantitative summary shown in main Fig. 3d, this panel shows Meissner corpuscle cells identified via S100-immunoreactivity that are seen in close proximity of collaterally sprouted mGFP-expressing nociceptors in tibial territory of SNI mice (42 weeks) and sham controls, of which some are Substance P-negative (arrowheads) and some are Substance P-positive (arrows) (N = 4).

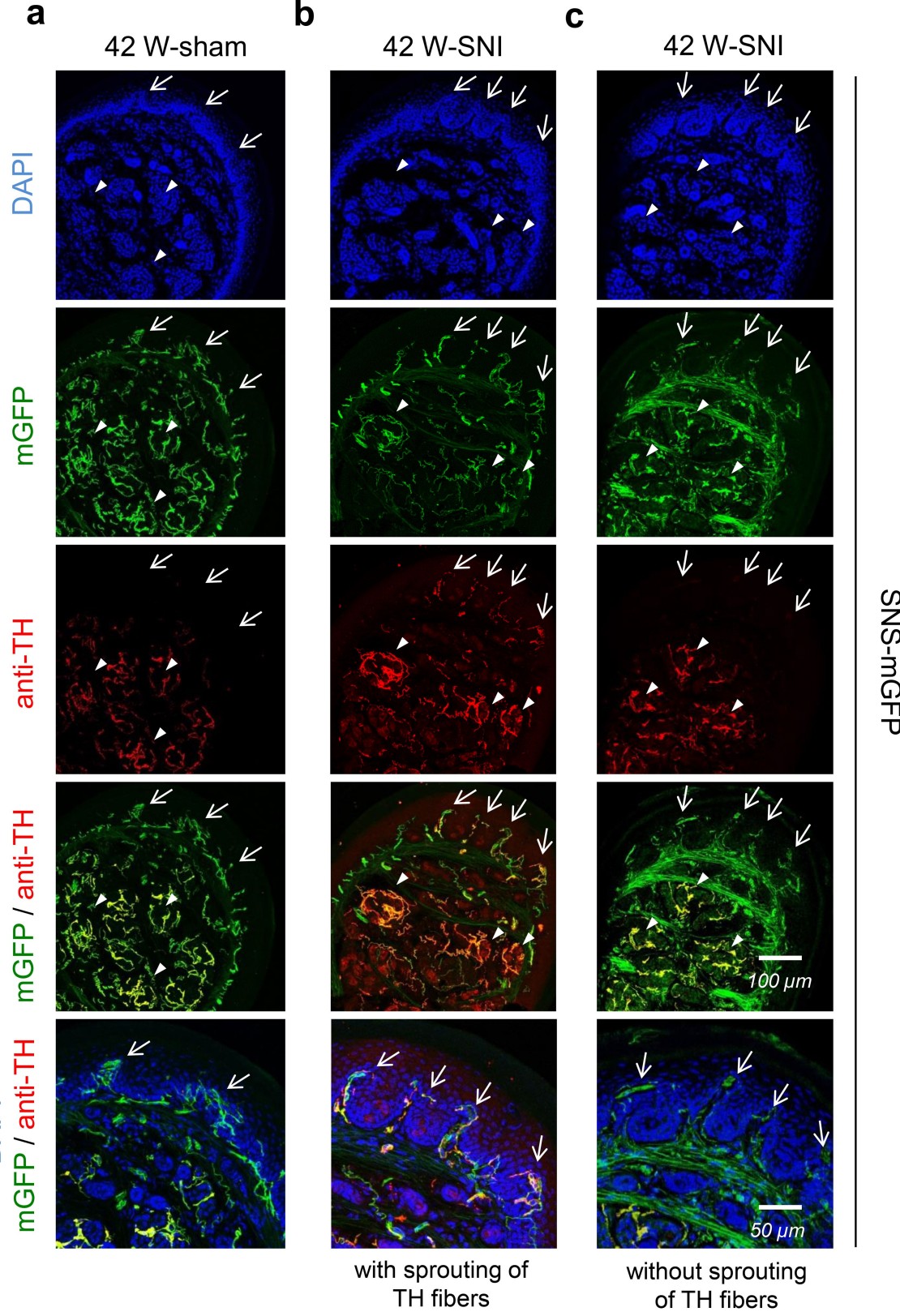

**Extended Data Fig. 9 | C-LTMRs in denervated tibial territories in mice at 42 weeks after SNI.** Examples of immunostaining for tyrosine hydroxylase (TH), a marker of C-LTMRs, which also stains sympathetic fibres, at 42 weeks after sham (**a**) and SNI (**b**, **c**) treatment in SNS-mGFP mice. Shown are examples of SNI mice (2 out of 8 mice) which showed ectopic presence of TH-expressing fibres at the dermal border (arrows in b) and SNI mice (6 out of 8 mice) that did not show any differences to sham-treatment (c). Scale bar = 50 μm.

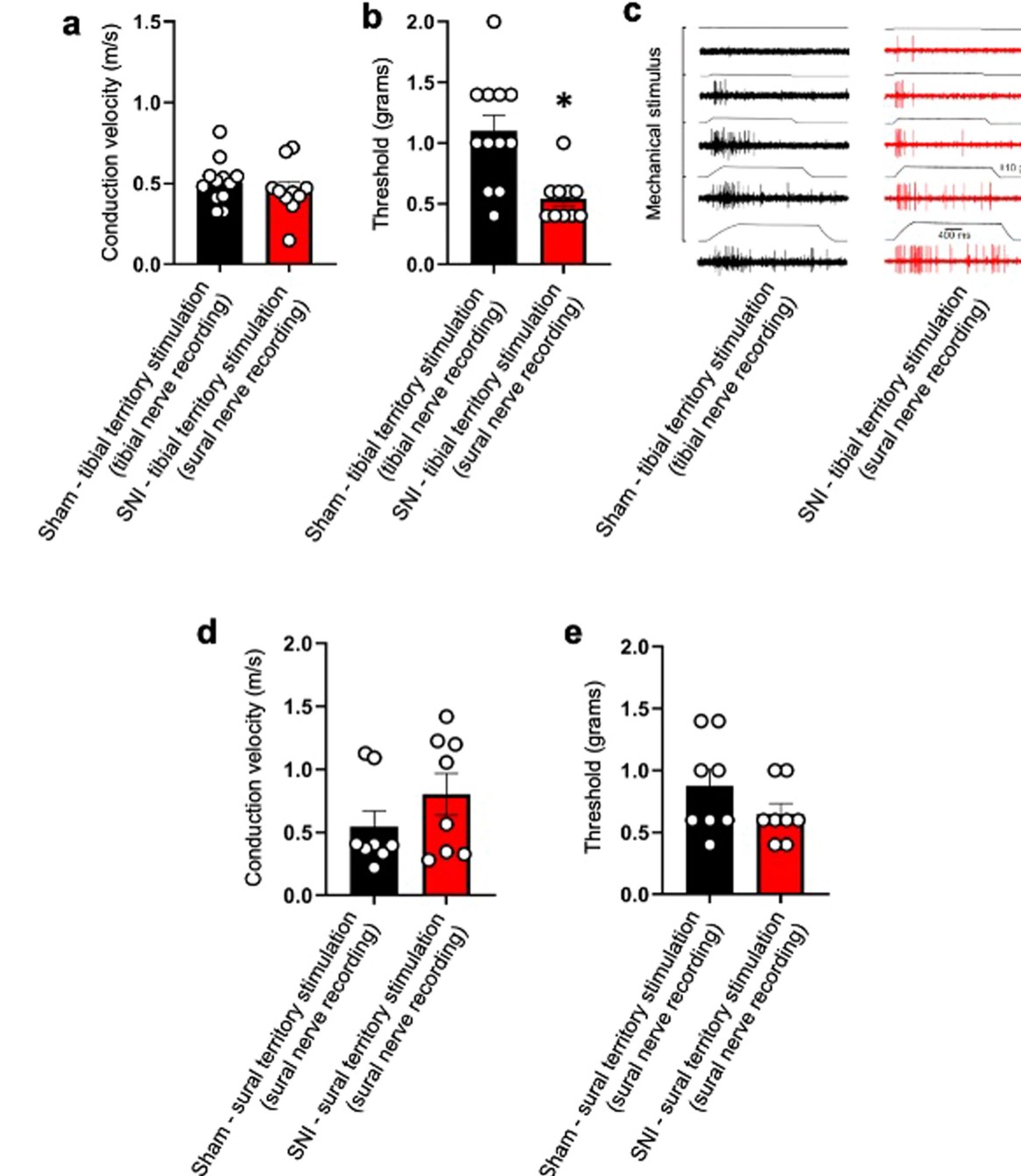

**Extended Data Fig. 10 |** See next page for caption.

**Extended Data Fig. 10 | Electrophysiological single-fibre recordings from skin nerve preparation from sural or tibial nerve with attached paw skin after mechanical stimulation over the sural (intact) and tibial (denervated) territories in mice at 24 weeks after SNI as compared to sham-treated mice.** (**a**–**c**) C-fibres identified via measurement of conduction velocity (a) and typical examples of evoked C-fibre responses in the tibial (sham) or sural (SNI) nerve after stimulation of the tibial territory (c) are shown. C-fibre response thresholds after stimulation of sural or tibial territories as indicated are shown (b). In panels a and b, N = 12 fibres from 3 sham mice and 12 fibres from 3 SNI mice; Unpaired two-tailed t-test; P = 0.4673, t = 0.74, df = 20, F = 1.37; in panel **a** and P = 0.00147, t = 3.68, df = 20, F = 5.55 in panel **b**. (**d**, **e**) Conduction velocity (d) and response threshold (e) of evoked C-fibre responses in the sural nerve (sham and SNI) after stimulation of the sural territory. In d and e, N = 8 fibres each from 6 sham and 5 SNI mice. Unpaired two-tailed t-test; P = 0.2362, t = 1.23, df = 14, F = 1.758 in panel **d** and P = 0.1788, t = 1.42, df = 14, F = 2.724 in panel **e**. Data are shown as mean ± S.E.M. This figure extends the data shown in main Fig. 3l, m.

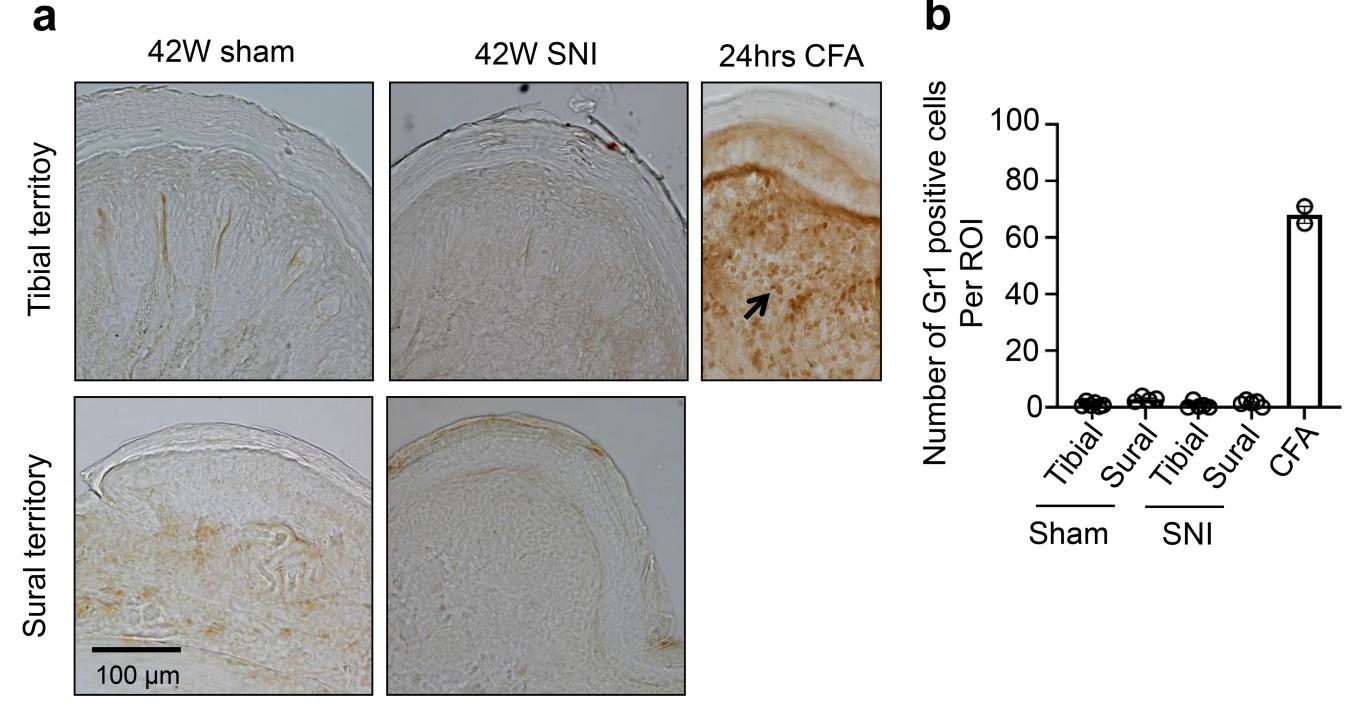

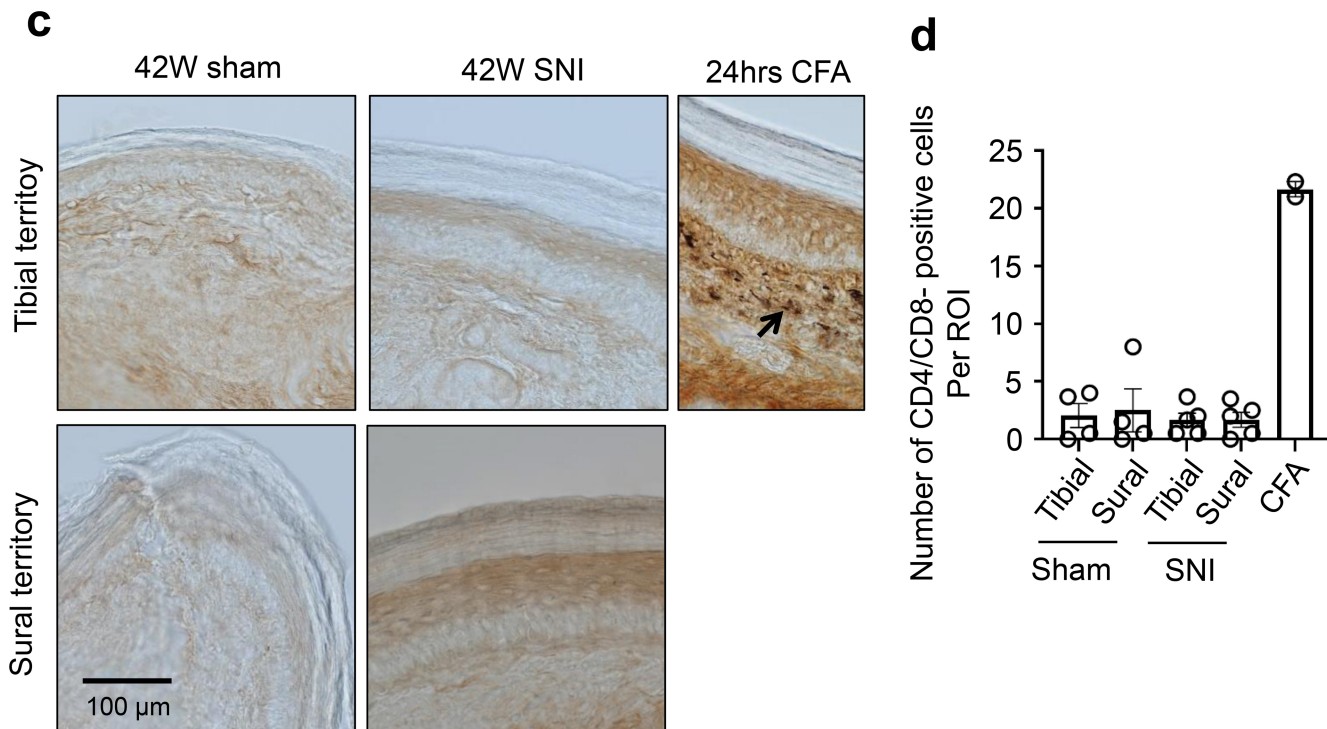

**Extended Data Fig. 11 | Analysis of potential immune cell accumulation in injured and uninjured nerve territories at the time of emergence of reinnervation neuropathic allodynia.** Typical examples (**a**, **c**) and quantification (**b**, **d**) demonstrating lack of immune cell infiltration and accumulation in the tibial or sural nerve territories at 24-26 weeks post-SNI, using tissue from hind paw inflamed with Complete Freund's Adjuvant (CFA) as positive controls. Shown are data with anti-Gr1 immunohistochemistry to identify macrophages (**a**, **b**; arrows in **a**) and anti-CD4/CD8 to identify T-cells (**c**, **d**; arrows in **c**). In both cases, N = 2–3 sections each from the 4 mice for sham-sural, 6 mice for sham-tibial group, 5 mice each from SNI-sural and SNI-tibial groups and 2 mice injected with CFA. ROI: region of interest.

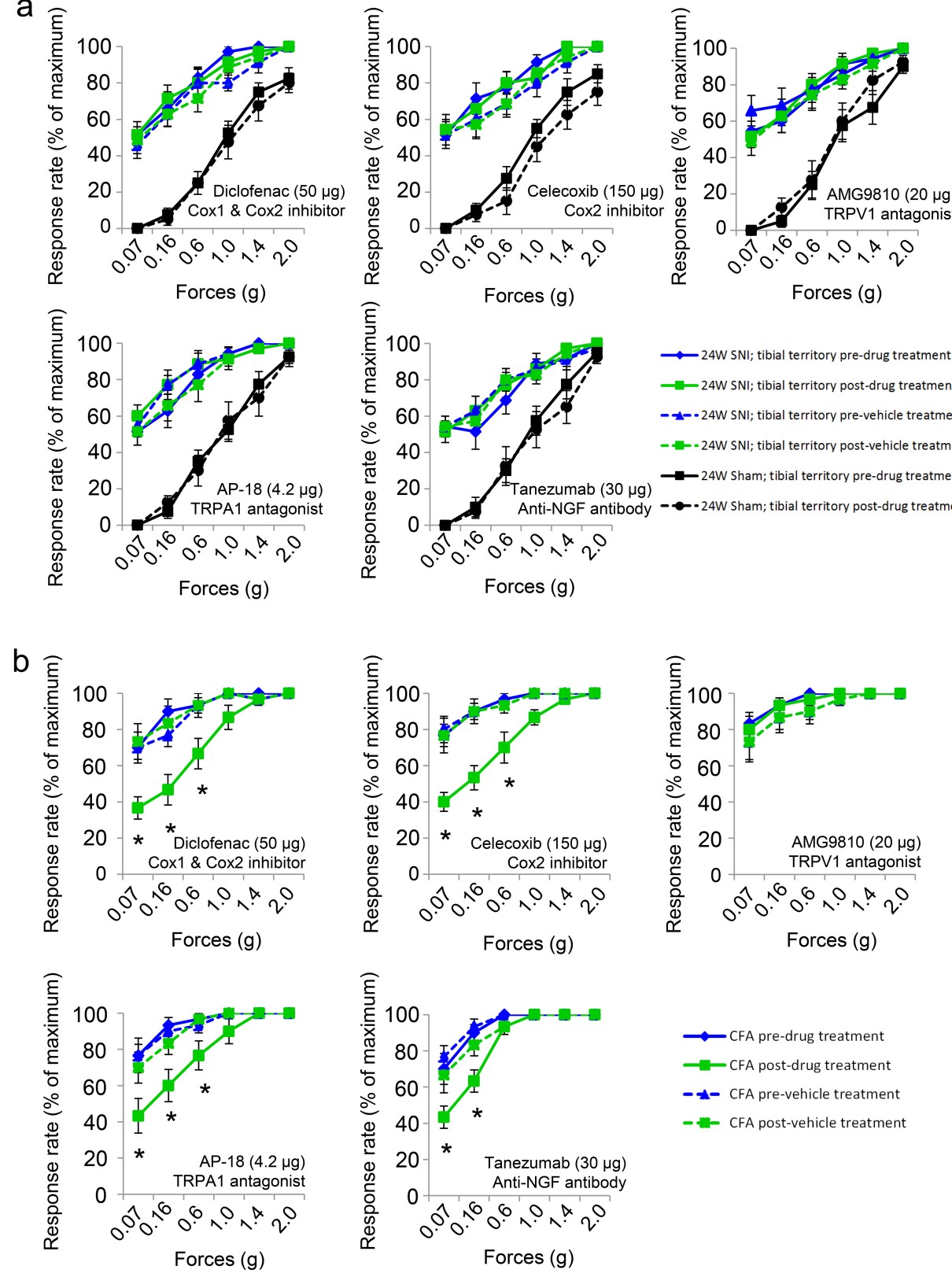

**Extended Data Fig. 12** | See next page for caption.

**Extended Data Fig. 12 | Effects of pharmacological inhibition of classical mediators of sensitization on reinnervation neuropathic allodynia.** Mechanical hypersensitivity in the tibial territory 24-26 weeks post-SNI is shown as compared to control (**a**), using mice with CFA-induced paw inflammation as positive controls for mechanisms of peripheral sensitization (**b**). Shown are data with inhibition of prostaglandin synthesis via blockade of both Cox-1 and Cox-2 (Diclofenac), Cox2-selective inhibition (Celecoxib), blockade of TRPV1 or TRPA1 or sequestration of NGF using a neutralizing antibody. In both models, vehicle-treated mice were used as negative controls. Note that blockade of prostaglandin synthesis, TRPA1 inhibition and NGF sequestration led to significant decrease in inflammatory mechanical allodynia, but did not affect reinnervation mechanical allodynia. N = 7 mice for and SNI-vehicle group, 8 mice for Sham-drug group and 6 mice each for CFA–drug and CFA–vehicle groups. For group comparison in panel a, $F_{(3,15)} = 1.608$, $P = 0.223$ (Diclofenac), $F_{(3,15)} = 0.541$, $P = 0.66$ (celecoxib), $F_{(3,15)} = 0.279$, $P = 0.84$ (AMG9810), $F_{(3,15)} = 0.649$, $P = 0.594$ (AP18) and $F_{(3,15)} = 0.121$, $P = 0.947$ (Tanezumab). For group comparison in panel b, $F_{(3,15)} = 29.222$, $P = 1.64E-06$ (Diclofenac), $F_{(3,15)} = 11.842$, $P = 0.000308$ (celecoxib), $F_{(3,15)} = 0.846$, $P = 0.49$ (AMG9810), $F_{(3,15)} = 10.0$, $P = 0.00071$ (AP18) and $F_{(3,15)} = 4.778$, $P = 0.0157$ (Tanezumab). Two-way ANOVA of repeated measures followed by Bonferroni's test for multiple comparisons was performed (* < p 0.05). Data are shown as mean ± S.E.M.

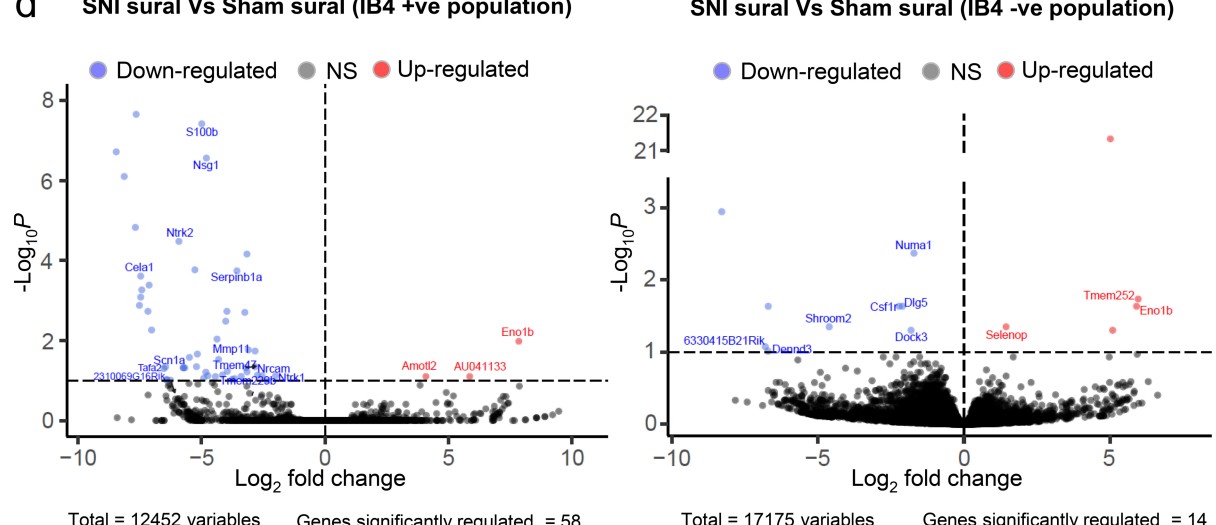

**Extended Data Fig. 13** | See next page for caption.

**Extended Data Fig. 13 | Overview of gene expression analyses of DRG neurons innervating the sural territory or collaterally sprouting into the tibial territory at 24 weeks after SNI as compared to control mice.**
(**a**, **b**) Volcano plots and summary of results (**a**) and putative functions ascribed to regulated genes or their families (**b**) in collaterally sprouting nociceptors in the tibial territory in SNI mice as compared to nociceptors innervating the tibial territory in control mice. Red and blue dots represent upregulated and downregulated genes, respectively. Unlabelled dots represent sequences lacking annotation. (**c**) Relative abundance of sub-populations of nociceptors was not significantly different between collaterally sprouting nociceptors in the tibial territory as compared to nociceptors innervating the tibial territory in control mice. ANOVA of random measures revealed lack of statistical significance. (**d**) Volcano plots and summary of results of genes regulated in nociceptors in the sural nerve territory of SNI mice as compared to control mice. Owing to space limits, not all regulated genes represented by blue and red dots are labelled in panel d. Further details are given in Supplementary Note 4 and sequencing reads are placed on the European Nucleotide Archive (https://www.ebi.ac.uk/ena) under the accession number PRJEB50184.

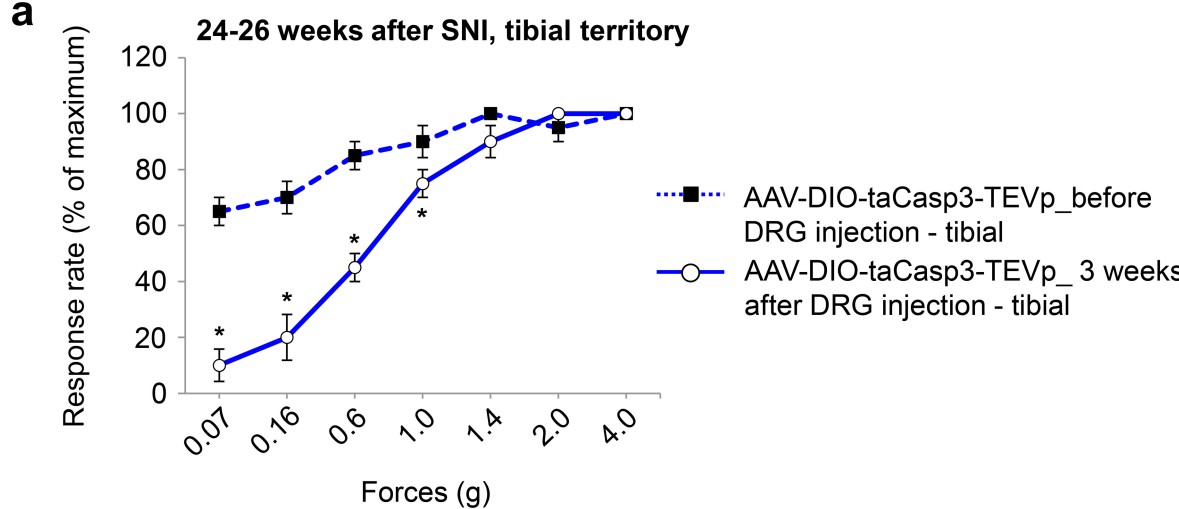

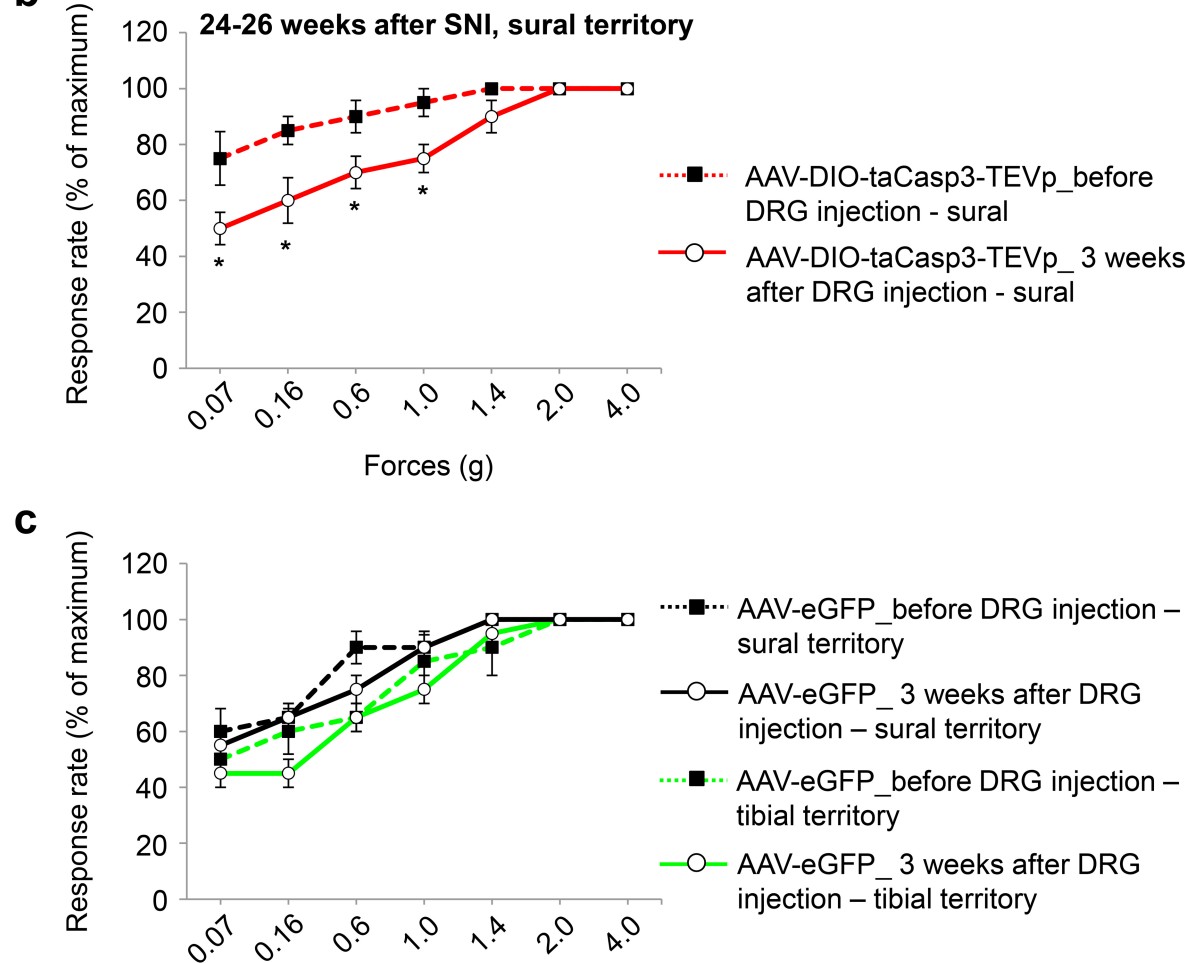

**Extended Data Fig. 14 | Effects of caspase-3-mediated nociceptor ablation specifically in L3–L4 DRGs on reinnervation neuropathic allodynia.** Shown is the behavioural effect of adeno-associated virus injection-mediated Cre-dependent caspase-3 expression in L3-L4 DRGs at 24-26 wks post-sham treatment or SNI on withdrawal responses to graded von Frey mechanical stimuli in the latter-most digit (sural territory, panel **a**) (n = 4 per group; $F_{(1,6)} = 33.0$, P = 0.01) or the middle digit (tibial territory, panel **b**) (n = 4 per group; $F_{(1,6)} = 15.0$, P = 0.030). Panel **c** represents mechanical sensitivity in mice receiving AAV-GFP (control) (n = 4 per group; for tibial, $F_{(1,6)} = 1.417$, P = 0.312 and for sural, $F_{(1,6)} = 3.0$, P = 0.182) injections. In all panels, * represents $p < 0.05$ compared to baseline, †$p < 0.05$ compared to control group (sham), two-way repeated measures ANOVA with Bonferroni multiple comparison. Data shown as mean ± S.E.M.

Reinnervation-induced neuropathic pain induced by collateral sprouting of nociceptors in the absence of low threshold tactile (Aβ) fiber regeneration, resulting in abnormal end organ connectivity and lack of masking effect of Aβ fibers on spinal response to nociceptor input

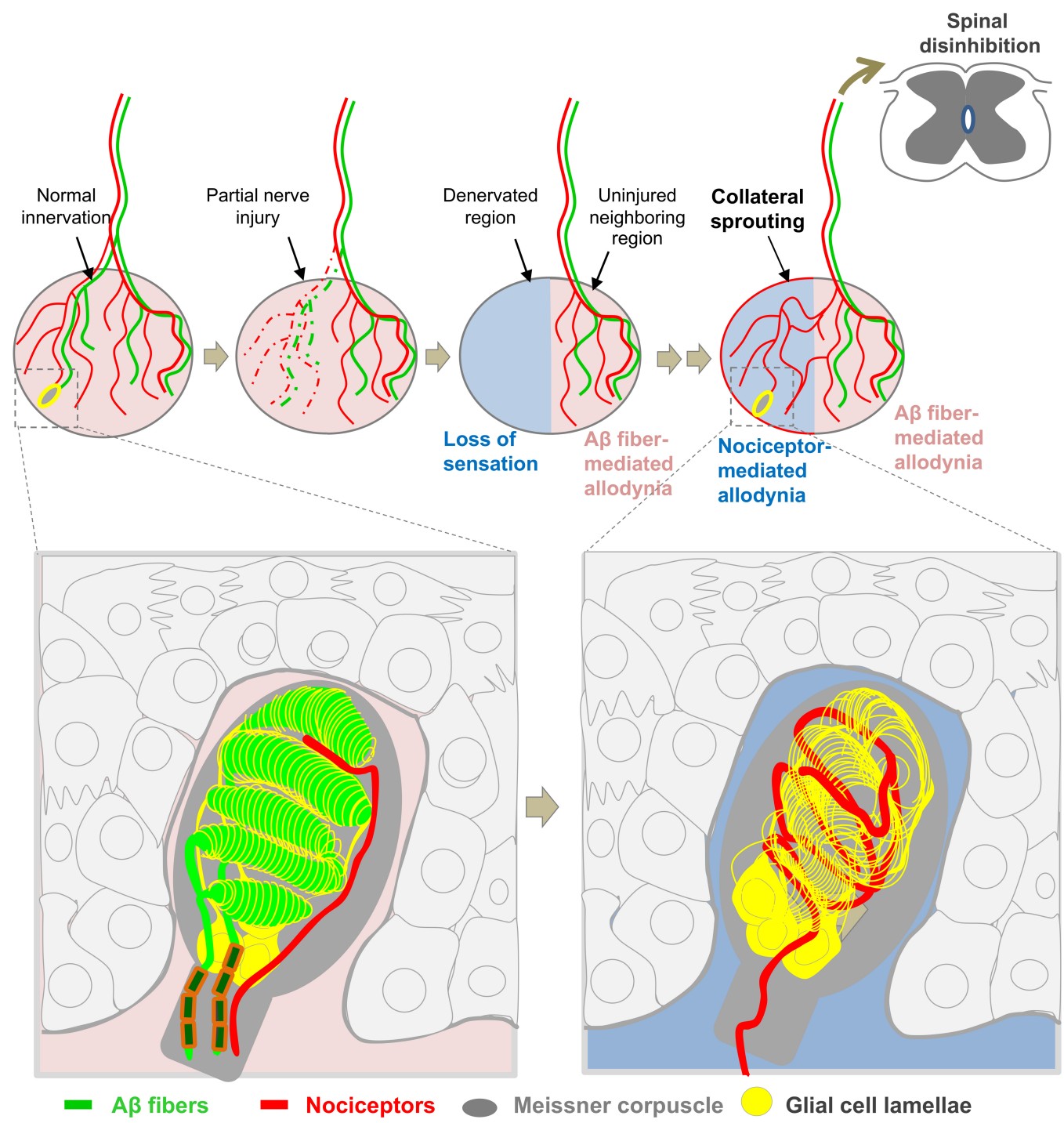

**Extended Data Fig. 15 | Schematic model.** The scheme depicts the proposed differential roles of nociceptors and Aβ-LTMRs in mediating two distinct types of neuropathic allodynia, both of which come about when areas undergoing denervation and intact nerve territories are intermingled, as is clinically the most frequent scenario with nerve trauma-associated neuropathic pain. In the novel reinnervation neuropathic allodynia described in this study, in the absence of Aβ reinnervation, collaterally sprouting C-fibres show abnormal pattern of connectivity in tactile-sensing Meissner corpuscles.

# nature research

# Reporting Summary

Nature Research wishes to improve the reproducibility of the work that we publish. This form provides structure for consistency and transparency in reporting. For further information on Nature Research policies, see our Editorial Policies and the Editorial Policy Checklist.

## Statistics

For all statistical analyses, confirm that the following items are present in the figure legend, table legend, main text, or Methods section.

| n/a | Confirmed | |
|---|---|---|
| ☐ | ☒ | The exact sample size (*n*) for each experimental group/condition, given as a discrete number and unit of measurement |
| ☐ | ☒ | A statement on whether measurements were taken from distinct samples or whether the same sample was measured repeatedly |
| ☐ | ☒ | The statistical test(s) used AND whether they are one- or two-sided<br>*Only common tests should be described solely by name; describe more complex techniques in the Methods section.* |
| ☒ | ☐ | A description of all covariates tested |
| ☐ | ☒ | A description of any assumptions or corrections, such as tests of normality and adjustment for multiple comparisons |
| ☐ | ☒ | A full description of the statistical parameters including central tendency (e.g. means) or other basic estimates (e.g. regression coefficient) AND variation (e.g. standard deviation) or associated estimates of uncertainty (e.g. confidence intervals) |
| ☐ | ☒ | For null hypothesis testing, the test statistic (e.g. *F*, *t*, *r*) with confidence intervals, effect sizes, degrees of freedom and *P* value noted<br>*Give P values as exact values whenever suitable.* |
| ☒ | ☐ | For Bayesian analysis, information on the choice of priors and Markov chain Monte Carlo settings |
| ☒ | ☐ | For hierarchical and complex designs, identification of the appropriate level for tests and full reporting of outcomes |
| ☒ | ☐ | Estimates of effect sizes (e.g. Cohen's *d*, Pearson's *r*), indicating how they were calculated |

*Our web collection on statistics for biologists contains articles on many of the points above.*

## Software and code

Policy information about availability of computer code

| Data collection | Imspector Pro (LaVision); Leica Application Suite X (LAS X); several custom-written softwares for EM image acquisition - available at Gitlab (https://gitlab.mpcdf.mpg.de/connectomics/emacquisitionmacro.git). |
|---|---|
| Data analysis | Software used for analysis include Sigmaplot, Image J, Microsoft Excel, ANYmaze, custom-written codes for image analysis of multiphoton imaging data - available at Github (https://github.com/zheng-tklab/pns_2photon_longitudinal-3Ddata-analysis). |

For manuscripts utilizing custom algorithms or software that are central to the research but not yet described in published literature, software must be made available to editors and reviewers. We strongly encourage code deposition in a community repository (e.g. GitHub). See the Nature Research guidelines for submitting code & software for further information.

## Data

Policy information about availability of data

All manuscripts must include a data availability statement. This statement should provide the following information, where applicable:

- Accession codes, unique identifiers, or web links for publicly available datasets
- A list of figures that have associated raw data
- A description of any restrictions on data availability

All of the raw data for behavior analyses, electrophysiology and immunohistochemistry analyses are provided in source data file and in the figures. RNA sequencing data are available on the European Nucleotide Archive (https://www.ebi.ac.uk/ena) under the accession number PRJEB50184. The raw data for multiphoton imaging and EM analyses are available upon request.

# Field-specific reporting

Please select the one below that is the best fit for your research. If you are not sure, read the appropriate sections before making your selection.

☒ Life sciences          ☐ Behavioural & social sciences          ☐ Ecological, evolutionary & environmental sciences

For a reference copy of the document with all sections, see nature.com/documents/nr-reporting-summary-flat.pdf

# Life sciences study design

All studies must disclose on these points even when the disclosure is negative.

| | |
|---|---|
| Sample size | Our sample sizes are also similar to those reported in previous publications. In previous studies we have determined the sample size using G-power analyses and therefore have a very clear set of what sample size is required for the behavioral and histochemical data reported. |
| Data exclusions | A mouse showing autotomy following spared nerve injury (SNI) was excluded from the analysis as it damages the imaging tissue volume. |
| Replication | All experiments were successfully replicated multiple times with several animals. The precise animal numbers are given in the figure legends. |
| Randomization | Groups were randomized and mice were allocated to experimental groups by a researcher different from the experimentor |
| Blinding | Experimentor was blinded to the identity of mice being analyzed in behavioral tests |

# Reporting for specific materials, systems and methods

We require information from authors about some types of materials, experimental systems and methods used in many studies. Here, indicate whether each material, system or method listed is relevant to your study. If you are not sure if a list item applies to your research, read the appropriate section before selecting a response.

## Materials & experimental systems

| n/a | Involved in the study |
|---|---|
| ☐ | ☒ Antibodies |
| ☒ | ☐ Eukaryotic cell lines |
| ☒ | ☐ Palaeontology and archaeology |
| ☐ | ☒ Animals and other organisms |
| ☒ | ☐ Human research participants |
| ☒ | ☐ Clinical data |
| ☒ | ☐ Dual use research of concern |

## Methods

| n/a | Involved in the study |
|---|---|
| ☒ | ☐ ChIP-seq |
| ☒ | ☐ Flow cytometry |
| ☒ | ☐ MRI-based neuroimaging |

## Antibodies

| | |
|---|---|
| Antibodies used | Primary antibodies:<br>Anti-beta-tubulin III (T2200, sigma; 1:500, raised in Rabbit), anti-NF200 (CH23015, Neuromics; 1:200, raised in Chicken), anti-CGRP (24112, Immunostar; 1:200, raised in Rabbit), anti-Substance P (GP14103 Neuromics; 1:200, raised in Ginuea Pig ), anti-S100 (Z0311, Dako; 1:200), anti-TROMA-I (Krt8, DSHB; 1:1000), anti-TH (SO25000, Neuromics; 1:200) and biotinylated Isolectin B4, IB4 (B-1205, Vector, 1:200), anti Gr-1(Mouse Ly-6G/ly-6C)(MAB 1037, R&D systems; 1.500), anti-CD8a (14-0808-82, Thermo Fisher; 1:200), anti-CD4 (14-9766-82, Thermo Fisher; 1:100)<br><br>Secondary antibodies:<br>Donkey anti-Rabbit IgG, Alexa 488 conjugated (Cat # A32790)<br>Donkey anti-Rabbit IgG, Alexa 594 conjugated (Cat # A32754)<br>Donkey anti-Rabbit IgG, Alexa 647 conjugated (Cat # A32787)<br>Donkey anti-Rat IgG, Alexa 647 conjugated (Cat # A48272)<br>Donkey anti-Rat IgG, Alexa 594 conjugated (Cat # A48271)<br>Donkey anti-Rat IgG, Alexa 488 conjugated  (Cat # A48269)<br>Goat anti-Guinea pig IgG, Alexa 647 conjugated (Cat # A-21450)<br><br>All of these above secondary antibodies were purchased from Thermo Fisher Scientific. |
| Validation | All the primary antibodies were used in non-living tissue for immunohistochemistry (IHC). These antibodies are extensively used for IHC purpose by the scientific community.<br><br>References: |

Neubarth NL, Emanuel AJ, Liu Y, Springel MW, Handler A, Zhang Q, Lehnert BP, Guo C, Orefice LL, Abdelaziz A, DeLisle MM, Iskols M, Rhyins J, Kim SJ, Cattel SJ, Regehr W, Harvey CD, Drugowitsch J, Ginty DD. Meissner corpuscles and their spatially intermingled afferents underlie gentle touch perception. Science. 2020 Jun 19;368(6497):eabb2751.

Selvaraj D, Gangadharan V, Michalski CW, Kurejova M, Stösser S, Srivastava K, Schweizerhof M, Waltenberger J, Ferrara N, Heppenstall P, Shibuya M, Augustin HG, Kuner R. A Functional Role for VEGFR1 Expressed in Peripheral Sensory Neurons in Cancer Pain. Cancer Cell. 2015 Jun 8;27(6):780-96.

Tong Liu, Temugin Berta, Zhen-Zhong Xu,Chul-Kyu Park,Ling Zhang, Ning Lü, Qin Liu, Yang Liu, Yong-Jing Gao, Yen-Chin Liu, Qiufu Ma, Xinzhong Dong, and Ru-Rong Ji. (2012). TLR3 deficiency impairs spinal cord synaptic transmission, central sensitization, and pruritus in mice. J Clin Invest. 122(6): 2195–2207.

# Animals and other organisms

Policy information about studies involving animals; ARRIVE guidelines recommended for reporting animal research

| | |
|---|---|
| Laboratory animals | Adult (8-56 weeks) C57Bl6 male/female mice (25 - 30 g) of wild-type, Thy1-YFP, SNS-mGFP, SNS-tdTomato, SNS-iDTR were used in this study.Mice were housed in groups of 2–3 per cage (in ventilation unit) with food and water ad libitum on a 12 h light / 12 h dark cycle. Room temparature and humidity were ranging from 20-23 °C and 40-60%. A detailed description of transgenic lines are available in supplementary information. |
| Wild animals | No wild animals were used in this study |
| Field-collected samples | No field collected samples were used this study |
| Ethics oversight | All of the animal experiments were conducted according to the ethical guidelines of 'Protection of Animals Act' under supervision of the 'Animal Welfare Officers' of Heidelberg University and were approved by the local governing body named 'Regierungspräsidium Karlsruhe: Abteilung 3 - Landwirtschaft, Ländlicher Raum, Veterinär- und Lebensmittelwesen', Germany (Approval numbers: G-206/11 and G-177/17). ARRIVE guidelines were followed. |

Note that full information on the approval of the study protocol must also be provided in the manuscript.

