## [Peer Review File · Nature]

Manuscript Title: Neuropathic pain caused by mis-wiring and abnormal end organ targeting

Reviewer Comments & Author Rebuttals

Reviewer Reports on the Initial Version:

Referees' comments:

Referee #1 (Remarks to the Author):

This is an interesting and detailed study offering novel insight into the mechanical allodynia associated with nerve injury-induced neuropathic pain, utilizing an important new way to measure changes in skin innervation by A-fibers and nociceptors over time in mice.

Regeneration is regrowth after injury of an injured axon – collateral sprouting is not regrowth, but novel growth of a non-injured neuron, they need to rename the pain they describe.

Do they have any thermal sensitivity data? If they do, they should include it if not why did they not measure this?

Essentially is cold allodynia present in the tibial territory and does it arise from collaterally sprouting nociceptors?

Why collaterally sprouted sural nociceptors have a lower mechanical threshold than those in the sural territory is as they admit, uncertain – they suggest it may be due to peripheral sensitization – is there any active inflammation in the tibial skin, does the threshold change in response to Cox2 inhibitors or Trpv1 antagonists?

EM images or high-resolution images of the basket-like structure of the nociceptors around Merkel endings is needed, the video provided is not detailed enough and doesn't show Merkel counterstain.

What triggers the intact sural nociceptors to grow into the denervated territory – is it NGF – do the sprouting nociceptors show induction of regeneration-associated genes?

They should highlight several aspects of their findings a bit more – that mechanical allodynia after nerve injury has two completely different mechanisms, an early and sustained A beta mediated form in the non-denervated territory (sural) and a nociceptor mediated one in the denervated territory (tibial) and that the latter only manifests very long after the injury and in consequence >90% of preclinical studies do not look at this chronic form of neuropathic pain.

How would they suggest testing for the presence of this new mechanism of mechanical allodynia in patients? What are the implications for therapies targeting mechanical allodynia?

Did they test mice of both genders?

Were all measurements blinded to the treatment?

Minor comments

Fig. 1 legend tibial should be sural!

When they use the term “total length” what exactly do they mean?

Although this lab developed the SNS-cre mouse line it is better to now call it Scn10a-cre (as does Jax) which is more informative.

In the abstract they use the term painful stimuli – pain is a response, not a stimulus and as they show it requires noxious stimuli in a naïve animal and innocuous in a nerve-injured mouse, they should be more specific.

Fig 4a. is not readable, the image quality and size needs to be better

Referee #2 (Remarks to the Author):

In this manuscript, the authors use the classic spared nerve injury (SNI) mouse model to describe a novel mechanism of neuropathic pain associated with regenerating nociceptors. Through elegant non-invasive longitudinal imaging of genetically labeled A β -LTMRs and nociceptors, along with behavioral measurements of mechanical sensitivity, the authors demonstrate that the denervated tibial nerve region of the mouse hind paw's glabrous skin becomes reinnervated by nociceptors, but not LTMRs, from the spared sural nerve. Interestingly, this nociceptor reinnervation correlates with the development of mechanical allodynia in the tibial region, and genetic ablation of nociceptors shows that they are required for the expression of this allodynia. The data is high quality and presented well, and this new model of regenerative neuropathic pain will be of great interest to the field.

The manuscript's major weakness exists in identify the underlying mechanism to explain the phenomenon (Fig. 3). In Figure 3 and the discussion, the authors emphasize the possibility that “atypical Meissner [corpuscle] connectivity” after nociceptor regeneration “converts” nociceptors from high-threshold to low-threshold. This “conversion” model is mainly built on morphological characterizations. The manuscript title is “Neuropathic pain caused by mis-wiring and abnormal end organ targeting”. However, if we look at the data, what the authors really showed is a partial regeneration by sural nerve after the tibial denervation. Since nociceptors normally innervate the Meissner corpuscles, as shown by previous studies by the authors' own data (80%), the main morphological change in the denervated tibial area is not the re-innervation of Meissner corpuscles (~60%) but the loss of free terminals in the epidermis. The title claim “mis-wiring and abnormal end organ targeting” seems exaggerating. The authors did some physiological recordings to compare the tibial nerve of sham condition and sural nerve (innervating the tibial region) of SNI condition (Fig. 3f-g). This is not the best design of experiment though. Since there are two variables (sham vs SNI, tibial vs sural) in the comparison, it is hard to draw conclusion from the current experiment. In short, the evidence to support this “conversion” model is weak and does not rule out equally or more likely explanations: 1) The regenerating nociceptors may have a different transcriptional profile from

baseline nociceptors that lower their mechanical threshold (peripheral sensitization, molecular instead of end organ morphological changes), and 2) loss of A β -LTMRs may lead to disinhibition of low-threshold responses from C fibers (gate control theory). Given the data presented in the manuscript and our current knowledge, the authors should include this possibility in the conclusion and discussion.

To differentiate between “abnormal end organ targeting” vs “peripheral sensitization” possibilities, the authors will need to expand their recording experiment in Fig. 3f-g. The fair experimental design and comparison should be: sham sural, sham tibial, SNI sural (sural side, normal morphology), SNI sural (tibial side, abnormal morphology). If this experiment shows that sural C fibers of SNI mice have a lower threshold in both sides, it suggests that molecular changes are present in these regenerating nociceptors and contribute to peripheral sensitization. To further examine molecular differences, the authors could follow up with RNA sequencing to compare sham and SNI sural nociceptors. Then, manipulating or knocking down major molecular players identified to show their role in regenerative neuropathic pain would greatly improve the mechanistic insight of this manuscript.

Other minor comments are –

1) Meissner corpuscle innervation by CGRP+ fibers has been shown in rodents before. See Yamamoto et al., 1988 (doi: 10.1016/0006-8993(88)90179-5) and other previous publications.

2) Fig 3f – this schematic should say “tibial nerve” for the sham recording, not sural nerve. It is also inaccurate to call these C fibers nociceptors for this experiment. Nociceptor is defined based on sensory threshold but not conduction velocity. Since the C fiber threshold has left shifted in SNI condition, the authors would not have a good way to tell/define whether the C fibers they recorded are nociceptors or not.

3) An additional potential experiment would be to sparsely label regenerating neurons and characterize their sprouting process on an individual, rather than population, level. As this neuropathic pain model is new to the field, a deeper characterization of its features will strengthen the study.

Referee #3 (Remarks to the Author):

Kuner and colleagues demonstrate a novel mechanism of neuropathic pain that develops over long periods and has never before been observed. In essence they demonstrate that de-nervated areas of skin after neuropathic injury are re-innervated with nociceptors that make aberrant terminations, and when deleted, loss of nociceptors abolish the development of neuropathic pain. This is highly original and of great significance.

The quality of the presentation, methodology, statistics and validity of the approach are unimpeachable. The long term study with newly developed technology (e.g. mGFP) that allowed the study is a model for future studies on chronology of a variety of disorders and in these terms alone is of great importance.

The relationship between blood vessels and re-innervation is observational and the mechanism awaits further experimentation. In terms of relevance to human neuropathic pain conditions, this work is of major significance. Of course, it is important to remember that other mechanisms may also contribute to neuropathic pain in the many conditions that lead to this type of pain and this is accurately reflected in the discussion.

The referencing, supplementary material and figure presentation are all fine. Figure 3g and 4d are key elements in the story.

In summary, this is a paper of immense significance for pain biologists and clinicians in terms of the focus of interest for new clinical interventions. It provides a wholly new mechanism and focuses the pain community on both peripheral as well as central changes that contribute to neuropathic pain.

Author Rebuttals to Initial Comments:

Responses to reviewers by Gangadharan et al. ms# 2021-04-06145:

The authors are very grateful to all reviewers for the highly constructive, knowledgeable and insightful reviews. The reviewer comments and suggestions were immensely helpful in improving the quality of the reporting and in designing new experiments to address key open questions. We have addressed all concerns and critical points and are delighted to present additional data from new experiments involving ultrastructural analyses using electron microscopy, electrophysiology, gene transcription analyses and pharmacological interventions. The manuscript has been stringently revised and we are pleased to report that all new data support the main inferences of the study and reveal new insights into mechanistic underpinnings.

Point-by-point responses are given below and the changes in main text are marked in blue text. For the reviewers' convenience, new data are represented in figures labelled with Roman numerics I to VIII at the end of this document to help differentiate them from figures in the main manuscript (labelled with Arabic numerics).

Referees' comments:

Referee #1 (Remarks to the Author):

This is an interesting and detailed study offering novel insight into the mechanical allodynia associated with nerve injury-induced neuropathic pain, utilizing an important new way to measure changes in skin innervation by A-fibers and nociceptors over time in mice.

Response: We thank the reviewer for the positive assessment and helpful feedback, which we have addressed and implemented as discussed below.

Regeneration is regrowth after injury of an injured axon – collateral sprouting is not regrowth, but novel growth of a non-injured neuron, they need to rename the pain they describe.

Response: The reviewer has raised an important distinction and we fully agree that it is important to describe this novel form of neuropathic pain in a semantically accurate manner. We had used the term 'regenerative pain' previously since it falls under the umbrella of the field of regeneration. As pointed out by the reviewer, we agree that regeneration of damaged fibers and collateral sprouting constitute two distinct modes of reinnervation, which is also described in the introduction section of the manuscript. We have now revised the term accordingly and think that 'reinnervation-induced pain' would be a more appropriate term since abnormalities in reinnervation following sprouting emerged to be the main contributing factor in our analyses (please also see responses to the next points along these lines).

Do they have any thermal sensitivity data? If they do, they should include it if not why did they not measure this?

Essentially is cold allodynia present in the tibial territory and does it arise from collaterally sprouting nociceptors?

Response: This is an interesting point. We initially did not test cold allodynia in the tibial territory because with conventional cold plate test, in which temperature changes are applied over the whole plantar surface in a freely-moving rodent, it is not possible to distinguish between the sural territory and the tibial territory. However, we fully agree with the reviewer that testing cold allodynia is important since it could strengthen the annotation of the changes we are observing to neuropathic pain.

We have now performed new experiments in which we placed a small volume of acetone (constant across all mice) on the tibial or sural territory of the skin, which rapidly evaporated and cooled the skin, and then tested for behavioral changes thereafter (i.e., as independently of mechanical stimulation as possible). Neuropathic mice demonstrated increased duration of nocifensive responses, such as paw flinches and paw lifting, to acetone as compared to control mice. We observed that this effect was strongest in the intact sural territory, but also

significant in the denervated tibial territory following reinnervation at 25 weeks post-SNI, indicating that cold allodynia is also a feature of this novel form of neuropathic pain (please see **Fig. I** below). This panel is shown as Extended Figure 2e in the revised manuscript.

We regret that owing to time limits, we were not able to perform new experiments with fiber ablation. However, since we now have evidence from a large number of independent methods (live imaging, immunohistochemistry, new ultrastructural data from electron microscopy, electrophysiology) showing that C-fibers, but not low threshold A β fibers, repopulate and reinnervate the denervated tibial territory, the probability is very high that cold allodynia is mediated by C-fibres in the reinnervated tibial territory post-SNI. In support, recent studies show that although Nav1.8-expressing neurons are activated in extreme cold, but not by cooling under physiological conditions (Luiz et al. 2019), they are involved in cold allodynia in several models of neuropathic pain (McDonald et al. 2021).

Why collaterally sprouted sural nociceptors have a lower mechanical threshold than those in the sural territory is as they admit, uncertain – they suggest it may be due to peripheral sensitization – is there any active inflammation in the tibial skin, does the threshold change in response to Cox2 inhibitors or Trpv1 antagonists?

Response: The authors would like to thank the reviewer for raising this point. Indeed, the suggested experiments are very helpful in delineating and clarifying whether peripheral sensitization is manifest and whether inflammatory processes play a role. We have now accordingly performed the following experiments:

1. We tested for the presence of immune cells that are typical for inflammation associated with tissue- and nerve-damage, such as macrophages (**Fig. II below**, new Extended Figure 11) and T-cells, in both sural and tibial territories at the time when sprouting-associated allodynia was fully established (25 weeks post-SNI). Neither of the two territories showed any build-up of immune cells and were indistinguishable from skin of naïve mice, in contrast to positive controls comprising mice with paw inflammation induced by complete Freund's adjuvant (CFA). Although immune cells are found in both territories early after nerve injury, it is unsurprising that the immune response is reduced or absent from the tissue this late after the initial injury, when the reinnervation-associated pain becomes manifest.
2. As suggested by the reviewer, we addressed whether mechanical allodynia in the tibial territory following reinnervation (24 weeks post-SNI) is dependent upon the classical mechanisms of peripheral sensitization, such as upregulation of TRP channels or actions of prostaglandins. Pharmacological blockade of either TRPV1, TRPA1, TrkA, COX1/COX2 or COX2-specific did not reverse neuropathic mechanical hypersensitivity in the tibial territory of SNI mice (**Fig. III below**, new Extended Figure 12). The validity of the drug application and dosage regime was established in positive control experiments involving mice with CFA-induced acute paw inflammation, a classical example of pain induced by peripheral sensitization (**Fig. IV below**, new Extended Figure 12).
3. As suggested by the reviewer further below, as well as by reviewer#2, we performed gene expression analyses on dorsal root ganglia (DRG) neuronal somata of sural afferents that send collateral sprouts to the tibial territory, which were identified by using territory-selective local injection of a retrograde tracer. As compared to control mice, in SNI mice (24-26 weeks post-SNI), we did not observe any upregulation of genes encoding reported mediators of peripheral sensitization. For example, there was no upregulation of ion-channels transducing sensory stimuli, such as TRPV1, TRPA1, Piezo2 or ATP-gated channels (P2X3), or enhancing membrane excitability, such as diverse sodium channels, calcium channels or potassium channels that have been linked to nociceptor sensitization, amongst others. Moreover, we performed similar back-labelling experiments on DRG neuronal somata of fibers with

terminations in sural territory and also did not observe any indication of gene expression changes indicative of nociceptor sensitization at this chronic time post-SNI. Data from IB4-positive (putative non-peptidergic nociceptors) and IB4-negative populations are shown below in **Fig. V** and **Fig. VI**, respectively (Extended Fig. 12 of the revised manuscript).

Taken together, these new data make two points: (i) there is lack of local inflammation at this late stage after SNI when reinnervation-induced allodynia is manifest, and, (ii) sural nociceptors collaterally sprouting to the tibial territory do not show significant gene expression changes that are typical for peripheral sensitization. This suggests that nociceptor sensitization phenomena that are hallmarks of other types of pain, such as acute and chronic inflammatory pain, are not causing the observed change in mechanical thresholds upon reinnervation. Indirectly, by exclusion, these findings point to a role for abnormalities in connectivity during reinnervation result in mechanical hypersensitivity after establishing reinnervation, for which we also now provide direct evidence in new electron microscopy (EM) data, as explained below.

We amended the manuscript on p. 15 (lines 399 to 417) to include these important points.

EM images or high-resolution images of the basket-like structure of the nociceptors around Merkel endings is needed, the video provided is not detailed enough and doesn't show Merkel counterstain.

Response: The authors fully agree with the reviewer and are grateful for the excellent advice of adding EM analysis, which we were happy to follow up on. We have now performed Serial Block-face Scanning Electron Microscopy to generate high-resolution 3D images from the tibial territory following reinnervation at 24-42 weeks post-SNI. We are particularly pleased to report that not only did these make it possible to observe nociceptor terminations in the skin end organs (Meissner corpuscles since this is glabrous skin) at a high resolution of both control mice and mice with SNI (red-labelled structures in **Fig. VIc** and **Fig.VIId**, respectively; shown in Figure 3 of the revised manuscript), as requested by the reviewer, but also now make a much stronger case for abnormal connectivity in the tibial territory following collateral sprouting in mice with SNI (**Fig. VIc-f**; please see full description in legend of **Fig. VII** and response to reviewer#2 below). Importantly, we also provide data on fully reconstructed Meissner corpuscle innervation (**Fig. VIe, f**), which now show the high-resolution 3D representations of nociceptor terminals (and their placement relative to A β fibers and glial cell lamella) in control mice and SNI mice below.

We amended the manuscript on p. 13/14 (lines 335 to 359) accordingly.

What triggers the intact sural nociceptors to grow into the denervated territory – is it NGF – do the sprouting nociceptors show induction of regeneration-associated genes?

Response: This is a very interesting question. Previous studies have implicated several growth factors, including NGF, glial-derived nerve growth factor (GDNF), artemin, laminin etc., in outgrowth of different types of sensory nerves following injury (e.g., Tucker et al. 2006; Mills et al. 2006; Kelamangalath et al. 2016). NGF is an excellent candidate for outgrowth of some types of nociceptors, and we do see genes contributing to or modulating NGF signaling amongst those regulated in collaterally sprouting nociceptors (please see below). However, because the NGF receptor TrkA is not expressed in all classes of nociceptors in adult stage (Usoskin et al. 2016), it is likely not sufficient to account for all types of nociceptors that we detect in the tibial territory following collateral sprouting from the sural nerve. Another mechanism underlying selectivity of collateral sprouting of nociceptors may be given by suppression of the outgrowth of A β fibers. For example, Piezo2 has been recently suggested to mediate suppression of axon outgrowth in *drosophila melanogaster* (Song et al. 2019), which may affect A β fibers more than nociceptors owing to higher levels of expression. Thus,

there are diverse possibilities that will be very interesting to dissect and functionally interrogate in new, long-term longitudinal studies, using various genetic interventions, sparse labelling of afferents, transgenic reporters for end organs, amongst others.

To answer the reviewer's questions about regeneration-associated genes, we now present gene expression analysis on DRG neuronal somata of nociceptors that collaterally sprouted into the tibial territory at 24-26 weeks post-SNI. Indeed, instead of finding regulation of genes linked to peripheral sensitization as explained above, we found that the most of genes that were significantly differentially expressed in collaterally sprouting nociceptors over sham conditions are directly linked to structural changes, such as neurite outgrowth, axonal pathfinding, neuroprotection and axonal survival, or belong to families that have been ascribed these functions. Some of these genes are indeed downstream of NGF and reported to be important for NGF to stimulate neurite outgrowth, supporting the point about structural remodelling and defects in end organ connectivity as underlying mechanisms for the observed phenotypic changes. A brief overview of functions ascribed to regulated genes or their families is given here and details are given in Table I and Table II at the end of the document.

Putative functions	Tibial territory: genes regulated in SNI vs. Sham
NGF signalling, neurite outgrowth, axonal pathfinding	Rin1, Plat, Olfml3, Tspan11
Neuroprotection, axonal survival post-injury	Mcur1, Eno1b, Plat, Olfml3, Gstm2, Tmem184c, Tspan11
Detoxification, lysosomal regulation, inflammasomes, caspases and ubiquitination	Mcur1, Rnf187, Pstpip1, Clvs2, Gstm2, Tmem184c
Cytoskeletal regulation	Rin1, Knstrn, Lrguk
Transcription factors, nuclear signalling, cell differentiation	Msc, Ovol2, Knstrn, Cela1

We amended the manuscript on p. 15 (lines 406 to 417) accordingly and added Supplementary Note 4 in addition to Extended Fig. 13).

They should highlight several aspects of their findings a bit more – that mechanical allodynia after nerve injury has two completely different mechanisms, an early and sustained A beta mediated form in the non-denervated territory (sural) and a nociceptor mediated one in the denervated territory (tibial) and that the latter only manifests very long after the injury and in consequence >90% of preclinical studies do not look at this chronic form of neuropathic pain.

Response: The authors appreciate the reviewer's point. We agree that the key hallmarks of the novel nociceptor-mediated neuropathic pain described in this study are that it is manifest in previously denervated territory, has a delayed onset post-injury and is not tested in a majority of preclinical studies. This is entirely distinct from the more widely studied form of neuropathic pain that originates from non-denervated areas, has a more rapid onset and is not dependent on nociceptors.

As per the reviewer's advice, we have now striven to highlight this distinction in the revised manuscript on page 17, lines 487-493. This is also represented in Extended Fig. 15.

How would they suggest testing for the presence of this new mechanism of mechanical allodynia in patients? What are the implications for therapies targeting mechanical allodynia?

Response: This is an interesting question. The neuropathic pain described here would be applicable to complete nerve transections as well as to conditions of partial nerve injury in which regeneration and regrowth of damaged nerve fibers is hindered or delayed, leading to collateral sprouting. We would suggest that these patients should be regularly subjected to follow-up neurological testing, informed about the possibility of mechanical allodynia developing in denervated regions with a delayed onset and encouraged to seek help from pain clinics. Using neurological testing, quantitative sensory testing and minimally invasive

electrophysiological recordings, the nociceptor origin of mechanical allodynia in this type of neuropathic pain can be determined. The implications for therapy are that in such conditions, mechanical allodynia would be expected to respond to pharmacological inhibition of transmission in nociceptors, either peripherally or by blocking their input into the spinal cord, e.g., nociceptor-specific sodium channel blockers (Nav1.8, Nav1.9), QX314+TRPV1 ligands come to mind for specific blockade of nociceptors (Binshtok et al. 2007). The mechanisms described here also provide scope for testing non-pharmacological neurostimulation regimes that could inhibit nociceptor function specifically.

Did they test mice of both genders?

Response: Yes, experiments pertaining to behavioral manifestation of allodynia after reinnervation, in vivo imaging and electrophysiology were carried out in mice of both sexes. This is now explained under methods more clearly (page 23, lines 729-731). We did not observe any obvious sex differences.

Were all measurements blinded to the treatment?

Response: Yes, the experimenters were always blinded to the treatment groups in all experiments, including behavioral, imaging, immunohistochemical experiments, electrophysiology and computational analyses. We have now placed this more prominently in the methods section (page 23, lines 730-731).

Minor comments

Fig. 1 legend tibial should be sural!

Response: Thank you, the legend has been corrected.

When they use the term “total length” what exactly do they mean?

Response: The authors apologize that owing to length restrictions, some of the terms are not explained in sufficient detail in the main text. Total length applies to the length of the fibers that could be traced until a depth of 400-450 microns into the tissue. Because there can be minor differences in the angle of placing the digit beneath the objective during multiphoton microscopy, we took care to work out a maximum common volume that was fully covered in all recording sessions, which was typically 600 microns * 600 microns * 450 microns (X*Y*Z). The total length of all epidermal and dermal fibers in this common volume is represented for a particular category of afferents (nociceptors or A β).

We have now clarified this in the text on lines 176-178.

Although this lab developed the SNS-cre mouse line it is better to now call it Scn10a-cre (as does Jax) which is more informative.

Response: We understand the reviewer’s point. We would like to clarify that there are two distinct lines targeting Cre recombinase to the murine *Scn10a* locus, namely: (i) Scn10a-Cre (Stirling et al. 2005), which is a knock-in line in the murine *Scn10a* locus, and (ii) a transgenic line using 110 kb of the promoter elements of the mouse *Scn10a* gene in a Bacterial Artificial Chromosome (BAC)(SNS-Cre; Agarwal et al. Genesis, 2004), which was generated in our laboratory. Although these two lines only show minor differences, because they are indeed genetically distinct from one another, it is important to keep their names apart as per guidelines for ensuring transparency and clarity. We agree with the reviewer about pointing out that the mouse line employs the *Scn10a* expression profile, which is stated on page 3 line 70 and explained more clearly under methods (page 22, lines 715-716).

In the abstract they use the term painful stimuli – pain is a response, not a stimulus and as they show it requires noxious stimuli in a naïve animal and innocuous in a nerve-injured mouse, they should be more specific.

Response: We fully agree with the reviewer, and have now corrected this error in the abstract.

Fig 4a. is not readable, the image quality and size need to be better

Response: We thank the reviewer for bringing this to our attention, and have now increased the size as well as the image quality in the revised Fig. 4a in the main text.

Referee #2 (Remarks to the Author):

In this manuscript, the authors use the classic spared nerve injury (SNI) mouse model to describe a novel mechanism of neuropathic pain associated with regenerating nociceptors. Through elegant non-invasive longitudinal imaging of genetically labelled A β -LTMRs and nociceptors, along with behavioral measurements of mechanical sensitivity, the authors demonstrate that the denervated tibial nerve region of the mouse hind paw's glabrous skin becomes reinnervated by nociceptors, but not LTMRs, from the spared sural nerve. Interestingly, this nociceptor reinnervation correlates with the development of mechanical allodynia in the tibial region, and genetic ablation of nociceptors shows that they are required for the expression of this allodynia. The data is high quality and presented well, and this new model of regenerative neuropathic pain will be of great interest to the field.

Response: The authors sincerely thank the reviewer for the positive appraisal and the constructive and helpful criticism, which has spurred us to perform additional experiments and add insightful new data as described below.

The manuscript's major weakness exists in identify the underlying mechanism to explain the phenomenon (Fig. 3). In Figure 3 and the discussion, the authors emphasize the possibility that "atypical Meissner [corpuscle] connectivity" after nociceptor regeneration "converts" nociceptors from high-threshold to low-threshold. This "conversion" model is mainly built on morphological characterizations. The manuscript title is "Neuropathic pain caused by mis-wiring and abnormal end organ targeting". However, if we look at the data, what the authors really showed is a partial regeneration by sural nerve after the tibial denervation. Since nociceptors normally innervate the Meissner corpuscles, as shown by previous studies by the authors' own data (80%), the main morphological change in the denervated tibial area is not the re-innervation of Meissner corpuscles (~60%) but the loss of free terminals in the epidermis. The title claim "mis-wiring and abnormal end organ targeting" seems exaggerating.

Response: We thank the reviewer for this insightful comment and have taken major steps to overcome this limitation. The term 'mis-wiring and abnormal end organ targeting' in the title was meant to imply both, the loss of intraepidermal endings of nociceptors and switch in the Meissner corpuscle terminations from dual innervation by large myelinated (non-nociceptive) and unmyelinated (nociceptive) fibers to nociceptors alone during the reinnervation of the denervated tibial territory. As explained in the subsequent paragraphs, we can now directly demonstrate abnormal end organ targeting. We therefore think that the title is now better justified and the underlying mechanisms are clarified in more detail.

Because nociceptors were observed to target Meissner corpuscles in both naïve (physiological) and neuropathic conditions, the authors fully appreciate that it is a very fair question to ask what exactly is different and can meaningfully lead to a functional change in mechanical sensitivity. We are excited to present data from new experiments that were stimulated by the reviewer's question, in which we performed serial block-face scanning electron microscopy on the tibial territory at 24-26 weeks post-SNI. The resulting ultrastructural high-resolution 3D representations of Meissner corpuscles now provide new evidence for abnormal end organ connectivity after collateral sprouting. These data are represented in **Fig. VII** below and make the following points:

- (i) Myelinated fibers are missing in the tibial territory following reinnervation; unmyelinated nerve fibers are the only neural component detected (**Fig. VIIb**, compared to control mice in **Fig. VIIa**). This fully validates all of our previous findings in the manuscript.
- (ii) Within the Meissner corpuscles of control mice, the previously reported concentric patterning of the glial cell lamellae (e.g., Neubarth et al. 2020), which is critical for sensing mechanical pressure, is seen around the terminals of myelinated afferents (false-coloured in green in **Fig. VIIc**, identified by tracing myelinated axons in serial sections), while unmyelinated C-fibers (false-coloured in red in **Fig. VIIc**, identified by tracing unmyelinated axons in serial sections) are found at the outer edges of these lamellar cushions. In contrast, in the tibial territory post-SNI, in the absence of myelinated afferent terminals, collaterally sprouted unmyelinated afferents show multiple terminations that are dispersed in between the lamellae (**Fig. VIId**). Although the lamellae are more loosely patterned than in control mice, it is clearly evident that they come in closer contact with unmyelinated C-fibers in the tibial territory of SNI mice. Some unmyelinated C-fibers are also seen at the centre of lamellar cushions in SNI mice, a position only taken by A β fiber terminations in control mice.
- (iii) Finally, we have put major efforts and succeeded in fully reconstructing Meissner corpuscle innervation in skeleton reconstruction analyses, which to our knowledge, is the first time that this is being reported. The high-resolution 3D data now make a clear case for abnormal connectivity. In control, physiological conditions, the myelinated A β afferents (false-coloured green in **Fig. VIIe**) show a classical patterning, winding throughout the Meissner and are fully covered by glial cell lamellae (false-coloured yellow in **Fig. VIIe**); in contrast, the unmyelinated afferents (false-coloured red in **Fig. VIIe**) do not branch extensively and are not surrounded by lamellae. In tibial territory of SNI mice, the collaterally sprouted unmyelinated afferents are found throughout the Meissner corpuscle volume, winding in between and importantly, being surrounded by glial cell lamellae (false-coloured yellow in **Fig. VIIf**), while A β fibers are missing (**Fig. VIIf**). This puts unmyelinated afferents, as the only nerve afferent component found inside the Meissner, physically in a position to sense changes induced by indentation of lamellae induced by mechanical stimuli.

We amended the manuscript on p. 13/14 (lines 335 to 359) accordingly.

The authors did some physiological recordings to compare the tibial nerve of sham condition and sural nerve (innervating the tibial region) of SNI condition (Fig. 3f-g). This is not the best design of experiment though. Since there are two variables (sham vs SNI, tibial vs sural) in the comparison, it is hard to draw conclusion from the current experiment. In short, the evidence to support this “conversion” model is weak and does not rule out equally or more likely explanations: 1) The regenerating nociceptors may have a different transcriptional profile from baseline nociceptors that lower their mechanical threshold (peripheral sensitization, molecular instead of end organ morphological changes), and 2) loss of A β -LTMRs may lead to disinhibition of low-threshold responses from C fibers (gate control theory). Given the data presented in the manuscript and our current knowledge, the authors should include this possibility in the conclusion and discussion.

To differentiate between “abnormal end organ targeting” vs “peripheral sensitization” possibilities, the authors will need to expand their recording experiment in Fig. 3f-g. The fair experimental design and comparison should be: sham sural, sham tibial, SNI sural (sural side, normal morphology), SNI sural (tibial side, abnormal morphology). If this experiment shows that sural C fibers of SNI mice have a lower threshold in both sides, it suggests that molecular changes are present in these regenerating nociceptors and contribute to peripheral sensitization. To further examine molecular differences, the authors could follow up with RNA sequencing to compare sham and SNI sural nociceptors. Then, manipulating or

knocking down major molecular players identified to show their role in regenerative neuropathic pain would greatly improve the mechanistic insight of this manuscript.

Response: The authors truly appreciate the reviewer's scholarly points, which are very helpful in working out mechanistic underpinnings.

We have now addressed possible mechanisms suggested by the reviewer, going through them point-by-point below. In addition to the 3 possibilities listed by the reviewer, we have also considered a 4th possibility below:

(i) Abnormal end organ targeting:

This was discussed in detail in response to the reviewer's point above (please also see **Fig. VII**). The EM data on ultrastructural analyses in the tibial territory and 3D representations of Meissner corpuscle innervation now furnish fresh evidence for abnormal end organ connectivity of collateral sprouting.

(ii) The regenerating nociceptors may have a different transcriptional profile from baseline nociceptors (peripheral sensitization, molecular instead of end organ morphological changes):

We have now performed unbiased sequencing and gene transcription analyses in mice 24 weeks post-SNI as suggested by the reviewer, using nerve territory-specific retrograde labelling to identify DRG somata, and make the following observations:

(a) The collaterally sprouting nociceptors do not show a gene expression profile that would be indicative of peripheral sensitization as there are no significant differences in expression of genes typically linked to activation or signal propagation properties in nociceptors and no molecular regulation that would support enhanced excitability (full sequencing data will be uploaded in a public repository; examples are shown in **Fig. V** and **Fig. VI** representing IB4-positive and IB4-negative populations, respectively). Similarly, sural nerve nociceptors innervating the undamaged sural territory also did not show significant molecular changes supportive of increased excitability (**Fig. V** and **Fig. VI**).

(b) The small number of genes that are significantly different in expression levels between collaterally sprouting nociceptors and the original tibial nociceptors largely pertain to molecules linked to structural integrity, including neurite outgrowth, axonal pathfinding and connectivity as well as protection against oxidative stress and neurodegeneration, which again support morphological remodelling as the key process ongoing in the nociceptors populating the denervated tibial territory (**Table I-Table II**).

Taken together, there is little evidence to support molecular differences leading to peripheral sensitization.

We also appreciate the reviewer's suggestion to test response thresholds in the sural territory in SNI mice, which we have now performed. Overall, we found this technically challenging since it was difficult to distinguish between the intermingled glabrous and hairy innervation, which are suggested to have different response thresholds, at the lateral edge of the paw comprising the sural territory. Moreover, the way the paw skin is cut away from the tissue and bone beneath for the skin-nerve preparation often involves damage, especially when dissecting the lateral edge with hairy and glabrous skin in one piece, rendering it hard to get good quality C-fiber recordings. Although we tried our best to overcome these challenges, these difficulties are likely reflected in the somewhat lower thresholds we recorded in sural territory of sham (control) mice. The data show that:

(a) In the undamaged sural territory, there appears to be a small drop in the response threshold in C-fibers over control mice, but it is not statistically significant (**Fig. VIIIa**, $p = 0.18$). The proportion of recruited C-fibers to graded mechanical stimuli is not significantly different between sham and SNI mice in the sural territory (**Fig. VIIIb**). Data shown in **Fig. 3m** and **Extended Fig. 10** of the revised manuscript).

(b) Only collaterally sprouting C-fibers show a statistically significant drop in mechanical threshold after reinnervating the tibial territory (**Fig. VIIIc**, $p = 0.001$). This is also significant

when compared to the response threshold in the sural territory in sham mice ($p = 0.04$). Recruitment of C-fibers to mechanical stimuli shows a significant leftward shift in the tibial territory post-SNI as compared to control condition (**Fig. VIIIId**). Data shown in Fig. 3m and Extended Fig. 10 of the revised manuscript).

Although we acknowledge that these data are not unequivocal, they help clarify that sural nociceptors are not consistently sensitized *per se* prior to reinnervating the tibial territory, particularly after taking this together with the lack of corresponding molecular changes. This is also consistent with a previous study by Stucky and colleagues, who noted increased number of action potentials at high intensities of mechanical stimulation but no change in thresholds in sural nociceptors in the SNI model (Smith et al. 2013). These findings also fit to our behavioural data, where we observed only a small decrease in mechanical hypersensitivity in the sural territory after nociceptor ablation (**Fig. 4c** of the main text) and to the knowledge in the field on the importance of A β fibers for allodynia in the undamaged sural territory (Abrahamsen et al. 2008; Dhandapani et al. 2018).

(iv) We also propose taking into consideration the hypothetical scenario that a certain subclass of nociceptors preferentially sprouts into the denervated tibial territory, leading to changes in activation profile. In our previous immunohistochemistry experiments, we did not find differences between the relative abundance of IB4+ and IB4-negative nociceptors targeting Meissner corpuscles across sham and SNI mice (**Fig. 3e** of main text), rendering it unlikely that this is the case. However, certain populations, for example the *Mrgprd*-expressing population of IB4-binding non-peptidergic (NP) nociceptors has been linked to neuropathic pain (Chen et al. 2006). Particularly the NP1 sub-population derived from single cell gene transcription studies (Ukoskin et al. 2016) is reported to selectively express receptors for lysophosphatidic acid, which elicits neuropathic pain (Usoskin et al. 2016). Therefore, we implemented deconvolution analyses in gene expression data to work out the relative abundance of sub-populations of nociceptors and found no differences. For example, the relative abundance of the NP1 sub-population in IB4-positive nociceptors is similar in collaterally sprouted (64% \pm 4 %) as compared to native innervation of the tibial territory (70% \pm 10 %).

(v) Loss of A β -LTMRs may lead to disinhibition of low-threshold responses from C fibers (gate control theory): Our view is 100% aligned to the reviewer's opinion. We had listed this as a possible mechanism, and regret that it was lost in the limited space allocated to the discussion in the manuscript text. We have now discussed it more prominently in the revised text.

Taken together, our data, placed in the context of current knowledge, indicate that the mechanistic underpinnings of the neuropathic pain occurring following reinnervation during regeneration can be ascribed to abnormal connectivity in the periphery and imbalance at the central spinal level of nociceptive processing. These considerations and inferences are now added to the discussion section (pages 18 and 19).

Finally, we agree that knocking out differentially expressed molecular players of sensitization would be a highly promising avenue to learn more about the mechanistic underpinnings. However, as explained above, we did not find strongly regulated key players that would qualify for this and we also think that adding this layer of experiments is beyond the scope of the work presented here.

Other minor comments are –

1) Meissner corpuscle innervation by CGRP+ fibers has been shown in rodents before. See Yamamoto et al., 1988 (doi: 10.1016/0006-8993(88)90179-5) and other previous publications.

Response: Thank you for bringing this to our notice. The study by Yamamoto et al. 1988 is now cited and the text has been revised accordingly.

2) Fig 3f – this schematic should say “tibial nerve” for the sham recording, not sural nerve. It is also inaccurate to call these C fibers nociceptors for this experiment. Nociceptor is defined based on sensory threshold but not conduction velocity. Since the C fiber threshold has left shifted in SNI condition, the authors would not have a good way to tell/define whether the C fibers they recorded are nociceptors or not.

Response: We fully agree with the reviewer regarding the nomenclature and have now labelled these afferents as ‘C-fibers’.

Former Fig. 3f (now Fig. 3l) is not meant to be a schematic representation for the data shown in former Fig. 3g (now Fig. 3m), and only the latter involves tibial recordings in sham mice – we apologize that this was not presented with sufficient clarity, which we have now amended.

Fig. 3l represents a distinct experiment employing sural nerve recordings in both sham and SNI conditions, and is actually a data figure showing that: (i) in sham mice, sural nerve responses can be mechanically-evoked from the sural territory at the lateral edge between the glabrous and hairy skin (represented by dots in the schematic), but not from the tibial territory (glabrous skin); (ii) In SNI mice, in contrast, sural nerve responses can be evoked from the tibial territory, indicating collateral sprouting of the sural nerve into the tibial territory and this holds true only for C-fiber responses (red dots), while A β fiber LTMR responses (blue dots) are missing. Thus, this experiment independently and functionally verifies our observations from imaging and EM experiments.

3) An additional potential experiment would be to sparsely label regenerating neurons and characterize their sprouting process on an individual, rather than population, level. As this neuropathic pain model is new to the field, a deeper characterization of its features will strengthen the study.

Response: We thank the reviewer for this insightful suggestion. Although it is beyond the scope of the present study, we fully agree that this would be an excellent option for future studies. Sparsely labelling regenerating fibers, and additionally implementing the Brainbow approach, will permit labelling different afferents in distinctive fluorescent hues and dynamically following individual trajectories and particularly, studying end organ connectivity. For the latter, it will be also revealing to cross in mice expressing other fluorescent reporters in end organ cells. These approaches will be particularly helpful in studying mechanisms underlying selective collateral sprouting observed for nociceptors using genetic and pharmacological interventions. We look forward to embarking on such studies in the future and supporting other groups in the field in addressing these exciting questions with the tools and analysis pipelines developed during this study.

During the revision, we have now added detailed EM analyses, which permitted us to study the trajectories of single myelinated and unmyelinated sensory fibers in the skin, and we put in major efforts to fully reconstruct their endings in Meissner corpuscles. Thus, we hope that this serves the purpose of visualizing single afferents and their trajectories in this study. Moreover, it adds important insightful details at the ultrastructural level on the morphology and connectivity of fiber terminals.

Referee #3 (Remarks to the Author):

Kuner and colleagues demonstrate a novel mechanism of neuropathic pain that develops over long periods and has never before been observed. In essence they demonstrate that de-nervated areas of skin after neuropathic injury are re-innervated with nociceptors that make aberrant terminations, and when deleted, loss of nociceptors abolishes the development of neuropathic pain. This is highly original and of great significance.

The quality of the presentation, methodology, statistics and validity of the approach are unimpeachable. The long term study with newly developed technology (e.g. mGFP) that allowed the study is a model for future studies on chronology of a variety of disorders and in these terms alone is of great importance.

The relationship between blood vessels and re-innervation is observational and the mechanism awaits further experimentation. In terms of relevance to human neuropathic pain conditions, this work is of major significance. Of course, it is important to remember that other mechanisms may also contribute to neuropathic pain in the many conditions that lead to this type of pain and this is accurately reflected in the discussion.

The referencing, supplementary material and figure presentation are all fine. Figure 3g and 4d are key elements in the story.

In summary, this is a paper of immense significance for pain biologists and clinicians in terms of the focus of interest for new clinical interventions. It provides a wholly new mechanism and focuses the pain community on both peripheral as well as central changes that contribute to neuropathic pain.

Response: The authors are very grateful for the reviewer's expert and positive appraisal of this work. We particularly appreciate that the reviewer pointed out a major significance to human neuropathic pain and the novelty and originality of mechanistic insights gained. In the revised manuscript, we have continued to strive to reflect in a balanced manner that other mechanisms also contribute to neuropathic pain.

References cited in responses:

Abrahamsen B, Zhao J, Asante CO, Cendan CM, Marsh S, Martinez-Barbera JP, Nassar MA, Dickenson AH, Wood JN. The cell and molecular basis of mechanical, cold, and inflammatory pain. *Science*. 2008 Aug 1;321(5889):702-5. PMID: 18669863.

Agarwal N, Offermanns S, Kuner R. Conditional gene deletion in primary nociceptive neurons of trigeminal ganglia and dorsal root ganglia. *Genesis*. 2004 Mar;38(3):122-9. PMID: 15048809.

Binshtok AM, Bean BP, Woolf CJ. Inhibition of nociceptors by TRPV1-mediated entry of impermeant sodium channel blockers. *Nature*. 2007 Oct 4;449(7162):607-10. PMID: 17914397.

Chen CL, Broom DC, Liu Y, de Nooij JC, Li Z, Cen C, Samad OA, Jessell TM, Woolf CJ, Ma Q. Runx1 determines nociceptive sensory neuron phenotype and is required for thermal and neuropathic pain. *Neuron*. 2006 Feb 2;49(3):365-77. PMID: 16446141.

Dhandapani R, Arokiaraj CM, Taberner FJ, Pacifico P, Raja S, Nocchi L, Portulano C, Franciosa F, Maffei M, Hussain AF, de Castro Reis F, Reymond L, Perlas E, Garcovich S, Barth S, Johnsson K, Lechner SG, Heppenstall PA. Control of mechanical pain hypersensitivity in mice through ligand-targeted photoablation of TrkB-positive sensory neurons. *Nat Commun*. 2018 Apr 24;9(1):1640. PMID: 29691410.

Kelamangalath L, Tang X, Bezik K, Sterling N, Son YJ, Smith GM. Neurotrophin selectivity in organizing topographic regeneration of nociceptive afferents. *Exp Neurol*. 2015 Sep;271:262-78. PMID: 26054884

Luiz AP, MacDonald DI, Santana-Varela S, Millet Q, Sikandar S, Wood JN, Emery EC. Cold sensing by Nav1.8-positive and Nav1.8-negative sensory neurons. *Proc Natl Acad Sci U S A*. 2019 Feb 26;116(9):3811-3816. PMID: 30755524

MacDonald DI, Luiz AP, Iseppon F, Millet Q, Emery EC, Wood JN. Silent cold-sensing neurons contribute to cold allodynia in neuropathic pain. *Brain*. 2021 Jul 28;144(6):1711-1726. PMID: 33693512

Mills CD, Allchorne AJ, Griffin RS, Woolf CJ, Costigan M. GDNF selectively promotes regeneration of injury-primed sensory neurons in the lesioned spinal cord. *Mol Cell Neurosci*. 2007 Oct;36(2):185-94. PMID: 17702601

Neubarth NL, Emanuel AJ, Liu Y, Springel MW, Handler A, Zhang Q, Lehnert BP, Guo C, Orefice LL, Abdelaziz A, DeLisle MM, Iskols M, Rhyins J, Kim SJ, Cattell SJ, Regehr W, Harvey CD, Drugowitsch J, Ginty DD. Meissner corpuscles and their spatially intermingled afferents underlie gentle touch perception. *Science*. 2020 Jun 19;368(6497):eabb2751. PMID: 32554568

Smith AK, O'Hara CL, Stucky CL. Mechanical sensitization of cutaneous sensory fibers in the spared nerve injury mouse model. *Mol Pain*. 2013 Nov 29;9:61. PMID: 24286165

Song Y, Li D, Farrelly O, Miles L, Li F, Kim SE, Lo TY, Wang F, Li T, Thompson-Peer KL, Gong J, Murthy SE, Coste B, Yakubovich N, Patapoutian A, Xiang Y, Rompolas P, Jan LY, Jan YN. The Mechanosensitive Ion Channel Piezo Inhibits Axon Regeneration. *Neuron*. 2019 Apr 17;102(2):373-389.e6. PMID: 30819546

Stirling LC, Forlani G, Baker MD, Wood JN, Matthews EA, Dickenson AH, Nassar MA. Nociceptor-specific gene deletion using heterozygous NaV1.8-Cre recombinase mice. *Pain*. 2005 Jan;113(1-2):27-36. PMID: 15621361.

Tucker BA, Rahimtula M, Mearow KM. Laminin and growth factor receptor activation stimulates differential growth responses in subpopulations of adult DRG neurons. *Eur J Neurosci*. 2006 Aug;24(3):676-90. PMID: 16930399.

Usoskin D, Furlan A, Islam S, Abdo H, Lönnnerberg P, Lou D, Hjerling-Leffler J, Haeggström J, Kharchenko O, Kharchenko PV, Linnarsson S, Ernfors P. Unbiased classification of sensory neuron types by large-scale single-cell RNA sequencing. *Nat Neurosci*. 2015 Jan;18(1):145-53. PMID: 25420068.

Fig. I

Fig. I: Increased duration of nocifensive responses to acetone, demonstrating significant cold allodynia in both, the undamaged sural territory and the denervated tibial territory after collateral sprouting at 24-26 weeks post-SNI (n = 8 for sham and 14 for SNI; *p < 0.05; two-tailed unpaired t test).

Fig. II

Fig. II: Typical examples (a) and quantification (b) demonstrating lack of immune cell infiltration and accumulation in the tibial or sural nerve territories at late stages post-SNI, using tissue from hindpaw inflamed with Complete Freund's Adjuvant (CFA) as positive controls. Shown are data with anti-Gr1 immunohistochemistry to identify macrophages (a, b) and anti-CD4/CD8 to identify T-cells. Similar results were obtained with T-cell marker proteins (n = 2-3 sections each from 4-6 mice/group).

Fig. III and Fig. IV: Impact of pharmacological inhibition of prostaglandin synthesis (panels a and b), blockade of TRPV1 (panel c) or TRPA1 (panel d) or sequestration of NGF (panel e) on mechanical allodynia in the tibial territory 24-26 weeks post-SNI (Fig. 3) or all over the plantar surface in mice with CFA-induced paw inflammation (Fig. 4, included here as positive controls). Vehicle-treated mice and sham-operated mice were used as negative controls. ANOVA of random measures followed by Tukey's test for multiple comparisons. * < p 0.05; n = 6-8 mice/group.

- ◆— 24W SNI; tibial territory pre-drug treatment
- 24W SNI; tibial territory post-drug treatment
- -▲- - 24W SNI; tibial territory pre-vehicle treatment
- -■- - 24W SNI; tibial territory post-vehicle treatment
- 24W Sham; tibial territory pre-drug treatment
- -●- - 24W Sham; tibial territory post-drug treatment

Fig. V
Fig. V: Gene expression data from DRG somata that were retrogradely labelled from either sural or tibial territories in mice at 24-26 weeks post-SNI or in control mice (n = 4 mice/group). Shown are data from IB4-positive cells. There are no statistically significant changes, although the sodium channels *Scn10a* (Nav1.8) and *Scn11a* (Nav1.9) showed a trend for increase in the sural territory, but not in the collaterals to the tibial territory.

Fig. VI**a****Sham tibial territory Vs SNI tibial territory****b****Sham sural territory Vs SNI sural territory****c****SNI sural territory Vs SNI tibial territory**
Fig.VI: Gene expression data from DRG somata that were retrogradely labelled from either sural or tibial territories in mice at 24-26 weeks post-SNI or in control mice (n = 4 mice/group). Shown are data from IB4-negative cells. There were no significant differences in expression of the ion-channels represented.

Fig. VII: High resolution ultrastructural images and 3D representations derived via Serial Sectioning Block-face Scanning Electron Microscopy on mouse tibial territory in the hindpaw digit either in control (sham) mice or mice at 24-28 weeks post-SNI.

(a, b) Images of dermal fibres in the vicinity of Meissner corpuscles; black arrowheads represent myelinated fibres (A β) and black arrows represent unmyelinated C-fibres. Note lack of A β fibres in collateral sprouts in SNI mice in panel b.

(c, d) 2D example images from serial sections of Meissner corpuscles, with A β terminations false-coloured in green and C-fibres terminations false-coloured in red. White arrowheads represent glial cell-derived lamellae. Note the concentric lamellae surrounding A β terminals in sham mice in panel c. Images in panel d show abnormal placement of C fibre terminals in close contact with lamellar wrapping in the absence of A β fibres, including at the centre of lamellar cushions. Figures are representative of 3 sham and 4 SNI mice.

(e, f) Full 3D representations of a Meissner corpuscle each from the tibial digit of control (e) and SNI (f) mice with high resolution view of patterning of an A β fibres (green) and C fibres (red) and their placement in relation to lamellar wrapping from the glial cell (false coloured in yellow). In the absence of A β fibres in SNI mice, C-fibres meander through Meissner corpuscles of SNI mice extensively and are wrapped by lamellae.

- Myelinated fibers
- Unmyelinated fibers
- Lamellar wrapping

Fig. VIII

Fig. VIII: Electrophysiological single fibre recordings following mechanical stimulation of either sural territory (panels a, b) or tibial territory (panels c, d) in control mice or at 24-26 weeks post-SNI. Shown are response thresholds (panels a, c) or overall proportion of fibres responding to particular intensities of mechanical stimulation (panels b, d). (a, b) Recordings were performed in C-fibres of the sural nerve for both sham and SNI groups. (c, d) Recordings were made from C-fibres in the tibial nerve in sham mice and from the sural nerve in SNI mice, which collaterally sprouts into the tibial territory after the tibial nerve is completely transected proximally. Two-tailed unpaired t-test was performed on data in panels a and c (p^* less than 0.05) and Chi-square analyses were performed on data shown in panels b and d. In a and b, $n = 8$ fibres from 6 control mice and 8 fibres from 5 SNI mice. In c and d, $n = 12$ fibres from 3 sham mice and 12 fibres from 3 SNI mice.

Table I: Transcription analysis on IB4-negative DRG neurons innervating the tibial territory:

List of gene upregulated in SNI 24 wks over sham mice:				
Number	Gene name	Protein	Putative function	Reference
1.	Plat	tissue plasminogen activator	Serine protease; Induced after nerve crush injury; tPA-plasmin signaling catalyses fibrin degradation and protects sciatic nerve axons upon injury	Akassoglou et al., JCB, 2000 PMID: 10831618
2.	Rin1	Ras and Rab interactor 1	Effector for Ras-GTPase, cytoskeletal modulator in neurons, endosomal regulation - activated downstream of NGF and EGF - NGF-induced neurite outgrowth blocked by dominant-negative RIN or Rin knockdown - In C. elegans , Rin1 regulates axonal pathfinding and neuronal cell migration	Spencer et a. JBC, 2002, PMID: 11877426 Shi et al. JBC, 2005 PMID: 16157584, Doi et al., Development, 2013, PMID: 23900541
3.	Olfml3	olfactomedin-like 3	Secreted glycoprotein; activated downstream of TGFβ1 signaling in microglia; downregulated in ALS and increased expression improves functional parameters in ALS models; zebrafish homologs show axonal elongation defects	Neidert et al. Frontiers in Immunology, PMID: 30093905
4.	Tmem184c	Trans-membrane protein 184c	Function unknown; its close family member Tmem184b localizes to recycling endosomes in axons; role in axonal degeneration; mutants show swellings with multivesicular bodies and abnormal sensory terminal morphology	Bhattacharya et al. J Neurosci. 2016
5.	Tspan11	tetraspanin 11	Function unknown, but other members of the family are implicated in neuroprotection; e.g. tetraspanin1 was found to be expressed in TrkA-expressing DRG neurons, increases the surface targeting of TrkA and is required for NGF-induced neurite outgrowth ; Tetraspanin 17 protects dopaminergic neurons, etc.	Ferrero et al. 2020 Cell Mol Life Sci., PMID: 31440771, Masoudi et al. PLoS Genet. 2014, PMID: 25474638
6.	Mcur1	mitochondria 1 calcium uniporter regulator 1	Critically required for formation of mitochondrial calcium uniporter (MCU); promotes mitochondrial bioenergetics, which is dysregulated in neurodegeneration; implicated in neuroprotection.	Tomar et al. Cell Reports, 2016 PMID: 27184846; Jung et al. Front. Cell Develop. Biology 2020 PMID: 33392190

7.	Gstm2	glutathione S-transferase mu 2	Phase II detoxification enzyme, involved in ROS detoxification, exerts a protective effect in response to metabolic stress, protects cells against autophagy and lysosome dysfunction - Function in neurons unknown - family member Gsta4 is associated with a highly significant increase in the survival of axotomized motoneurons and central neurons after brain injury	Ström et al. 2012, Neuromol. Med. PMID: 22160604 Huenchuguala et al. 2014, Autophagy, PMID: 24434817
8.	Ppp2r1b	protein phosphatase 2, regulatory subunit A, beta	Serine threonine protein phosphatase; neural function unknown	Uniprot.org; genecards.org
9.	Knstrn	kinetochore-localized astrin/SPAG 5 binding	Small Kinetochore-Associated Protein, promotes stable microtubule-kinetochore attachments -Neural function unknown, but human mutation associated with optic nerve atrophy	Sharfe et al. 2018 PMID: 29180244
10.	Ovol2	ovo like zinc finger 2	Transcription factor in embryogenesis; Required in neural tube development Functions in adult nervous system unknown	Mackay et al. Developmental Biology, 2006, PMID: 16423343
11.	Msc	Musculin	Basic helix-loop-helix transcription factor; functions in neurons unknown	Uniprot.org; genecards.org
12.	Lrrc17	leucine rich repeat containing 17	Function in neurons unknown	Uniprot.org; genecards.org
13.	ENSMUSG-00000076617	NA	NA	NA
14.	ENSMUSG-00000080893	NA	NA	

List of gene downregulated in SNI 24 wks over sham mice:

Number	Gene name	Protein	Putative function	Reference
1.	Lrguk	leucine-rich repeats and guanylate kinase	Predicted to enable guanylate kinase activity, cytoskeletal regulator; function in neurons unknown	Okuda et al. 2017, FASEB J. PMID: 28003339

		domain containing		
2.	Mrgpra3	MAS-related GPR, member A3	Marker for the non-peptidergic (NP2) cluster of DRG nociceptors; - Mrgpra3 -expressing neurons implicated in nocifensive and pruriceptive responses; - reduced in inflammatory pain models - Mixed MrgprA3 - MrgprC11 agonists induce hypersensitivity in bladder afferents	Usoskin et al., 2016 Nat. Neurosci, PMID: 25420068. Grundy et al. J. Neurosci 2021 PMID: 33727332
3.	Celal	chymotrypsin-like elastase family, member 1	Serine endopeptidase; neuronal function unknown; predicted to be a negative regulator of post-embryonic development and differentiation, predicted to be part of Wnt signaling	Uniprot.org, Ensembl databases

Table II: Transcription analysis on IB4-positive neurons DRG innervating the tibial territory:

List of gene upregulated in SNI 24 wks over sham mice:				
Number	Gene name	Protein	Putative function	Reference
1.	Eno1b	enolase 1B	Glycolytic enzyme and can act as a surface receptor for plasminogen - role in oxidative stress and protects against hypoxia - novel protective role described in cerebral ischemia-induced neuronal injury, where it is upregulated post-injury, and also in stroke models	Jiang et al. ACS Chem Neurosci. 2019, PMID: 30943007 ; Jiang et al. 2021 Neurosci. Lett
2.	ENSMUSG00000080893	NA	NA	
List of gene downregulated in SNI 24 wks over sham mice:				
Number	Gene name	Protein	Putative function	Reference
1.	Clvs2	clavesin 2	Neuron-specific; enriched in Clathrin-coated vesicles; Knockdown leads to alterations in lysosomal morphology	Katoh et al., JBC, 2009, PMID: 19651769
2.	Pstpip1	proline-serine-threonine phosphatase-interacting protein 1	Role in inflammation, implicated in caspase and IL1 regulation cascade; functions in neurons unknown	Holzinger and Roth, Curr. Opin. Rheumatology, 2016, PMID: 27464597

3.	Rnf187	ring finger protein 187	E3 ubiquitin-protein ligase, involved in protein ubiquitination pathway, functions in neurons unknown	Uniprot.org
4.	Gpr150	G protein-coupled receptor 150	Orphan GPCR; suggested to be related to Oxytocin-Vasopressin receptor family, function unknown	Gloriam et al. 2005, Biochim Biophys Acta PMID: 15777626.

Reviewer Reports on the First Revision:

Referees' comments:

Referee #1 (Remarks to the Author):

The authors have adequately addressed the reviewers comments

Referee #2 (Remarks to the Author):

The authors have successfully addressed points raised by the reviewer with substantial revisions. Overall, their conclusions have been significantly strengthened by the new experiments, and no major concerns remain. This manuscript will provide important and novel insights into the role of nociceptor reinnervation during SNI-induced neuropathic pain and is recommended for publication.

One suggestion to further enhance exploration of abnormal end-organ innervation would be to investigate nociceptor innervation of Merkel cells in the denervated tibial territory after SNI. A quick way to determine whether this innervation pattern is present would be to stain for Merkel cells (e.g. anti-CK8 antibody) with SNS-mGFP labeled nociceptors. Given the known function of Merkel cells in facilitating low-threshold mechanical force sensing, this could provide another explanation for the lowered threshold in the new Fig 3m.

A minor comment is that some of the sentences are a bit difficult to read. A couple of examples:

P. 15, lines 395-398: "such, that" the comma should be removed, and "alike low-threshold afferents" should be "like low-threshold afferents do" (or something similar)

P. 17, line 489: "the existence of distinct two completely different forms" is redundantly worded.

Author Rebuttals to First Revision:

Responses to reviewers by Gangadharan et al.:

Referees' comments:

Referee #1 (Remarks to the Author):

The authors have adequately addressed the reviewers comments

Response:

Thank you very much.

Referee #2 (Remarks to the Author):

The authors have successfully addressed points raised by the reviewer with substantial revisions. Overall, their conclusions have been significantly strengthened by the new experiments, and no major concerns remain. This manuscript will provide important and novel insights into the role of nociceptor reinnervation during SNI-induced neuropathic pain and is recommended for publication.

One suggestion to further enhance exploration of abnormal end-organ innervation would be to investigate nociceptor innervation of Merkel cells in the denervated tibial territory after SNI. A quick way to determine whether this innervation pattern is present would be to stain for Merkel cells (e.g. anti-CK8 antibody) with SNS-mGFP labeled nociceptors. Given the known function of Merkel cells in facilitating low-threshold mechanical force sensing, this could provide another explanation for the lowered threshold in the new Fig 3m.

Response:

We thank the reviewer for the in-depth review and scholarly comments. We fully agree that abnormal end-organ connectivity with Merkel cells is an attractive possibility. We had initially not followed up on this since Merkel cells are relatively sparse in glabrous skin, which was the focus of this study. We have now performed the immunohistochemical analyses suggested by the reviewer and show the data in the figure below.

In sham-treated mice, as expected, we observe that Merkel cells (stained with anti-CK8 antibody) are innervated by large diameter tactile afferents that are NF-200-positive [REDACTED], but not by nociceptors (marked here with CGRP). In denervated tibial territory after nerve injury (SNI), as we reported in other figures, tactile fiber innervation is lost as seen with lack of staining for NF-200 – surprisingly, we also see a loss of Merkel cell staining. That this is not caused by potential artifacts of the tissue from SNI mice is shown by the observation that we can spot Merkel cells, innervated by tactile fibers, in the undamaged sural nerve territory [REDACTED]. Nociceptors are present, as reported in the manuscript, but Merkel cells appear to be missing, thus likely negating the possibility that Merkel cell innervation by nociceptors is underlying lowered thresholds in the denervated tibial territory. However, because these data suggest that Merkel cells themselves undergo degeneration when their innervation is lost, we believe that these analyses are not unequivocal and deserve more refined analysis and specifically targeted electron microscopic verification. This would be currently out of scope of the present manuscript. We therefore suggest not to show these data in the current manuscript.

[REDACTED]

A minor comment is that some of the sentences are a bit difficult to read. A couple of examples:
P. 15, lines 395-398: “such, that” the comma should be removed, and “alike low-threshold afferents” should be “like low-threshold afferents do” (or something similar)
P. 17, line 489: “the existence of distinct two completely different forms” is redundantly worded.

Response: We thank the reviewer for pointing this out and have now revised the corresponding text along the lines suggested by the reviewer.